JCB Journal of Cell Biology

# Interactions with multiple inner kinetochore proteins determine mitotic localization of FACT

Julia Schweighofer[1,2]*, Bhagyashree Mulay[1,2]*, Ingrid Hoffmann[1], Doro Vogt[1], Marion E. Pesenti[1], and Andrea Musacchio[1,2,3]

**The FAcilitates Chromatin Transcription (FACT) complex is a dimeric histone chaperone that operates on chromatin during transcription and replication. FACT also interacts with a specialized centromeric nucleosome containing the histone H3 variant centromere protein A (CENP-A) and with CENP-TW, two subunits of the constitutive centromere–associated network (CCAN), a 16-protein complex associated with CENP-A. The significance of these interactions remains elusive. Here, we show that FACT has multiple additional binding sites on CCAN. The interaction with CCAN is strongly stimulated by casein kinase II phosphorylation of FACT. Mitotic localization of FACT to kinetochores is strictly dependent on specific CCAN subcomplexes. Conversely, CENP-TW requires FACT for stable localization. Unexpectedly, we also find that DNA readily displaces FACT from CCAN, supporting the speculation that FACT becomes recruited through a pool of CCAN that is not stably integrated into chromatin. Collectively, our results point to a potential role of FACT in chaperoning CCAN during transcription or in the stabilization of CCAN at the centromere during the cell cycle.**

## Introduction

Chromosomes are DNA packaging structures that consist of a single molecule of DNA and many different associated proteins. They contain several functionally specialized regions that work in conjunction with transcription, replication, and inheritance. A notable specialized chromatin locus is the centromere. The histone H3 variant centromere protein A (CENP-A) is greatly enriched at centromeres and is considered the crucial epigenetic marker of centromeres. CENP-A seeds the kinetochore, a large protein complex that connects the replicated chromosomes (sister chromatids) to spindle microtubules during mitosis to ensure their equal distribution to the daughter cells (Musacchio and Desai, 2017; Talbert and Henikoff, 2020). Its presence at centromeres recruits specialized machinery that delivers new CENP-A at every cell cycle to compensate for its dilution during DNA replication (Stirpe and Heun, 2022).

The kinetochore is divided into inner and outer layers (Brinkley and Stubblefield, 1966). The outer layer, consisting of 10 proteins collectively referred to as the Knl1 complex, Mis12 complex, and Ndc80 complex (KMN) network and associated proteins, is assembled during mitosis to directly attach to spindle microtubules (Cheeseman et al., 2006). The inner layer, consisting of 16 proteins collectively referred to as the constitutive centromere–associated network (CCAN), bridges the centromeric chromatin and outer kinetochore and localizes to the centromere throughout the cell cycle (Foltz et al., 2006; Perpelescu and Fukagawa, 2011). The CCAN consists of different subunits and subcomplexes, including CENP-C, CENP-HIKM, CENP-LN, CENP-OPQUR, and CENP-TWSX (McAinsh and Meraldi, 2011) (Fig. 1 A).

Two CCAN proteins, CENP-C and CENP-N, decode the centromere by recognizing CENP-A (Song et al., 2002; Trazzi et al., 2009; Carroll et al., 2009, 2010; Klare et al., 2015; Pentakota et al., 2017; Kato et al., 2013). In addition to binding CENP-A, CENP-C interacts directly with other inner kinetochore subunits, including CENP-HIKM and CENP-LN, as well as the outer kinetochore complex MIS12 (Cohen et al., 2008; Screpanti et al., 2011; Klare et al., 2015; McKinley et al., 2015; Nagpal et al., 2015; Walstein et al., 2021). Another subunit, CENP-T, binds stably to CENP-W and connects the CCAN and the outer kinetochore by interacting with Mis12 and Ndc80 complexes (Mis12C and Ndc80C, respectively) through its long disordered N-terminal tail (Rago et al., 2015; Huis In't Veld et al., 2016). CENP-W and the C-terminal region of CENP-T consist of a histone fold domain (HFD). The CENP-TW subcomplex further tetramerizes with two additional HFD-containing proteins, CENP-S and CENP-X. It has been reported that the resulting CENP-TWSX complex is integrated into centromeric chromatin as a nucleosome-like particle (Hori et al., 2008; Nishino et al., 2012). Recent

[1]Department of Mechanistic Cell Biology, Max Planck Institute of Molecular Physiology, Dortmund, Germany;   [2]Centre for Medical Biotechnology, Faculty of Biology, University Duisburg-Essen, Essen, Germany;   [3]Max Planck School Matter to Life, Heidelberg, Germany.

*J. Schweighofer and B. Mulay contributed equally to this paper.   Correspondence to Andrea Musacchio: andrea.musacchio@mpi-dortmund.mpg.de;   Bhagyashree Mulay: bhagyashree.mulay@mpi-dortmund.mpg.de;   Julia Schweighofer: juliamaria.schweighofer@mpi-dortmund.mpg.de.

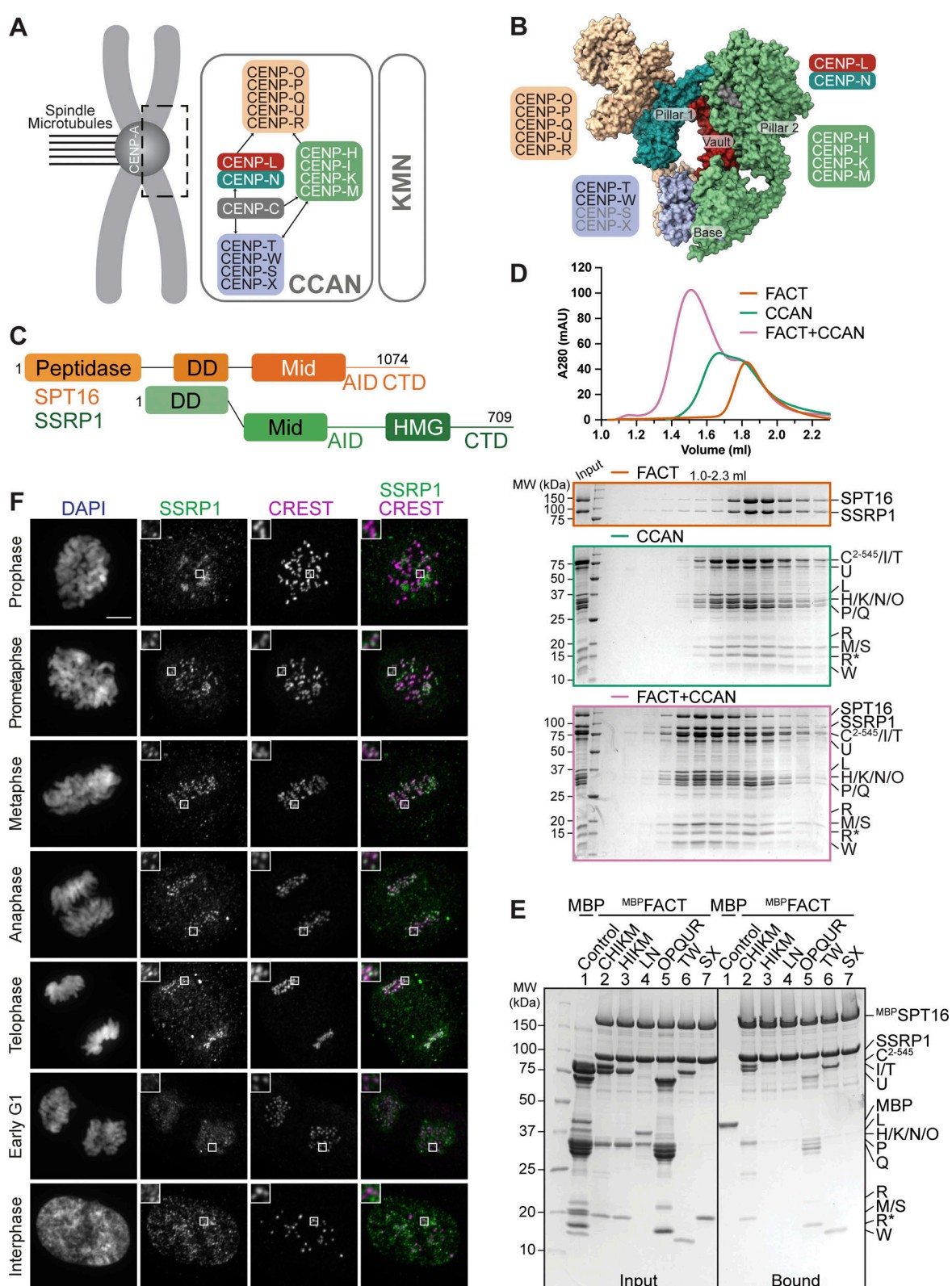

Figure 1. **FACT forms a complex with the CCAN in vitro and localizes to the kinetochores during mitosis. (A)** Scheme of the human kinetochore with a focus on subunits of the CCAN. **(B)** Surface representation of the structure of the human CCAN based on PDB 7QOO with the subcomplexes colored as in A. The structure lacks subunits CENP-SX. **(C)** Scheme of the domain architecture of the FACT complex, SPT16 in orange and SSRP1 in green. Peptidase, peptidase-like domain; DD, dimerization domain; Mid, Mid domain; AID domain, acidic intrinsically disordered domain; CTD, C-terminal domain; HMG, high mobility group. **(D)** Analytical SEC of FACT, CCAN including CENP-C$^{2-545}$, and the resulting 18-subunit complex. Fractions were analyzed by SDS-PAGE and visualized by Coomassie staining. R* is a proteolytic product of CENP-R. **(E)** Amylose-resin pull-down assay with FACT, where SPT16 has an N-terminal MBP-tag, and immobilized on beads and isolated CCAN subcomplexes as preys. **(F)** Representative images of localization of SSRP1 during mitosis. Asynchronous RPE-1 cells

structural work has shown that the CCAN consists of two structural pillars (composed of CENP-HIKM and CENP-OPQUR) flanking a central DNA-binding vault (contributed by CENP-LN) and a base (CENP-TWSX; Fig. 1 B). The central vault enables tight binding of the linker DNA by CCAN. *In vitro* and *in vivo*, CENP-A has been shown to form an octameric nucleosome consisting of a CENP-A/H4 tetramer flanked by two H2A/H2B dimers wrapped by ~150 base pair (bp) DNA (Fukagawa and Earnshaw, 2014). The CENP-A nucleosome has been proposed to neighbor the CCAN structure bound to the linker DNA (Pesenti et al., 2022; Yatskevich et al., 2022; Tian et al., 2022).

The original CENP-A co-precipitation experiments that identified CCAN subunits also identified the FAcilitates Chromatin Transcription (FACT) complex for a specific interaction with CENP-A nucleosomes (Obuse et al., 2004; Foltz et al., 2006; Izuta et al., 2006; Roulland et al., 2016; Seath et al., 2023). FACT is an H2A/H2B chaperone that prevents histone loss while facilitating the assembly and disassembly of nucleosomes during transcription (Orphanides et al., 1998; Belotserkovskaya et al., 2003; Saunders et al., 2003; Hsieh et al., 2013). Additionally, it has been implicated in DNA replication and repair (Schlesinger and Formosa, 2000; Keller and Lu, 2002; Krohn et al., 2003; Tan et al., 2006; Kumari et al., 2009; Xin et al., 2009; Han et al., 2010; Richard et al., 2016; Yang et al., 2016, 2020). FACT is a heterodimer of suppressor of Ty protein 16 (SPT16) and structure-specific recognition protein 1 (SSRP1), both large multidomain proteins with an array of pleckstrin homology domains (Orphanides et al., 1999; Winkler and Luger, 2011; Winkler et al., 2011). SPT16 has an N-terminal peptidase-like domain, which has lost its catalytic activity but interacts with minichromosome maintenance protein complex 2–7 and with the fork protection complex during replication, as well as with the Set3 histone deacetylase complex (Wang et al., 2023; Safaric et al., 2022; Leng et al., 2021). The SPT16 Mid domain binds to histone H3/H4 tetramers. The subsequent acidic intrinsically disordered (AID) segment associates with H2A/H2B dimers (Kemble et al., 2013, 2015; Tsunaka et al., 2016). SSRP1 contains a high mobility group (HMG) domain, which is associated with DNA binding (Yarnell et al., 2001; Štros et al., 2007) (Fig. 1 C).

While the precise significance of its interaction with centromeres remains elusive, FACT is believed to promote CENP-A deposition and to prevent ectopic localization of CENP-A. In chicken DT40 cells, for instance, FACT and CHD1 are targeted to the kinetochore by CENP-HIKM to facilitate CENP-A deposition (Okada et al., 2009). In *Drosophila melanogaster*, FACT assists in transcription-coupled CENP-A deposition by directly binding to the CENP-A assembly factor CAL1 (Chen et al., 2015). In budding yeast, the E3 ubiquitin ligase Psh1 requires binding to FACT to efficiently ubiquitinate CENP-A$^{Cse4}$, targeting it for proteasomal degradation (Deyter and Biggins, 2014). Similarly, in fission yeast, the mutation of FACT leads to the accumulation of overexpressed CENP-A$^{Cnp1}$ at noncentromeric chromatin (Choi et al., 2012). Furthermore, FACT has been implicated in the

maintenance of pericentromeric heterochromatin and the deletion of SSRP1$^{Pob3}$ results in chromosome missegregation (Lejeune et al., 2007). In humans, FACT has been shown to directly interact with CENP-TW HFDs via the AID of SPT16 (Prendergast et al., 2016). In this study, we demonstrate that the interaction of FACT with CCAN is complex, with additional binding sites on CENP-C and CENP-OPQUR. FACT engages in a stable 18-subunit complex with CCAN, whose assembly requires the phosphorylation of FACT by the constitutively active kinase casein kinase II (CK2). Mitotic localization of FACT at the kinetochore is dominated by CENP-HIKM and CENP-TW, and we show that CENP-TW levels are reduced upon FACT depletion. We find that DNA displaces FACT from CCAN, suggesting a potential role of FACT in chaperoning CCAN during transcription or in the deposition of CCAN at the centromere during or after replication.

## Results

### FACT forms a stable complex with CCAN *in vitro*

As CCAN and FACT co-precipitate with CENP-A nucleosomes and FACT has been proposed to bind directly to CCAN subunits, we asked whether a CCAN/FACT complex could be reconstituted *in vitro* using recombinant proteins. Previously, we have reconstituted a 16-subunit CCAN from four stable recombinant subcomplexes, including CENP-CHIKM (assembled with C$^{2–545}$, a fragment of CENP-C encompassing residues 2–545), CENP-LN, CENP-OPQUR, and CENP-TWSX (Pesenti et al., 2018, 2022; Weir et al., 2016). We reconstituted CCAN starting from these subcomplexes (Fig. S1 A). In analytical size-exclusion chromatography (SEC) experiments, FACT and CCAN co-eluted in a single peak and at earlier elution volumes relative to the individual complexes, indicating that CCAN and FACT bind directly in an 18-subunit complex (Fig. 1 D). The addition of excess FACT did not result in a larger shift, and quantification of tryptophan fluorescence of the bands in SDS-PAGE in peak fractions indicated approximately equal amounts of SPT16 and various CCAN subunits, suggesting a 1:1 stoichiometry of FACT and CCAN (Fig. S1, B and C).

To identify CCAN subunits involved in FACT binding, we immobilized FACT on amylose-resin through an N-terminal MBP-tag on SPT16 ($^{MBP}$FACT) and used the various CCAN subcomplexes as preys. In addition to confirming the previously reported interaction with CENP-TW (Prendergast et al., 2016), we observed interactions with CENP-OPQUR and CENP-C$^{2–545}$ HIKM (Fig. 1 E). The latter interaction required CENP-C$^{2–545}$, because the CENP-HIKM complex, which lacks CENP-C, did not bind (Fig. 1 E, lane 3). This extends previous observations suggesting a direct interaction of FACT with CENP-HIKM (Okada et al., 2009). CENP-SX, which contains HFDs similar to CENP-TW (Nishino et al., 2012), did not bind to $^{MBP}$FACT (Fig. 1 E, lane 7). We confirmed the association of FACT with CENP-TW and CENP-OPQUR by analytical SEC, whereas FACT and CENP-

C$^{2-545}$HIKM did not form a stable complex in solution (Fig. S1 D). We conclude that FACT and CCAN bind directly and that the interaction is mediated by multiple binding interfaces (Fig. S1 E).

## Mitotic localization of FACT to the kinetochore depends on the CCAN

FACT localizes to chromatin, especially nucleoli, in interphase, reflecting its role in transcription (Birch et al., 2009; Jeong et al., 2022). FACT was also observed to localize to centromeres during mitosis in chicken DT40 cells and to interphase and mitotic centromeres in *Drosophila* (Okada et al., 2009; Chen et al., 2015). To investigate mitotic FACT localization in human cells, we stained SSRP1 by immunofluorescence in hTERT-immortalized retinal pigment epithelial (RPE-1) cells. FACT localized to the kinetochore in all mitotic phases, exhibiting a more diffuse signal in early and late mitosis and interphase (Fig. 1 F and Fig. S2 A). To dissect how FACT is recruited to kinetochores during mitosis, we exploited a previously described colorectal adenocarcinoma DLD-1 cell line allowing rapid degradation of CENP-C (Fachinetti et al., 2015). In this system, both CENP-C alleles are endogenously tagged with an auxin-inducible degron (Nishimura et al., 2009) and an enhanced yellow fluorescent protein. After treating mitotic cells with the auxin derivative indole acetic acid (IAA) for 4 h, CENP-C was completely depleted from kinetochores, while CENP-HK and CENP-TW remained largely unaffected, as previously observed (Pesenti et al., 2022). SSRP1 localization was also largely unaffected (Fig. 2, A and B; and Fig. S2, B–I), indicating that recruitment of FACT is independent of CENP-C or that FACT remains stably localized after initial depletion of CENP-C. When the treatment with IAA was extended to 24 h, however, the kinetochore levels of FACT were greatly decreased. This correlated with modest-to-strong decreases in CCAN subunit localization (Fig. 2, C–J). Collectively, these observations link kinetochore localization of FACT to the interactions with CCAN observed *in vitro*, although they do not exclude a potential role of centromere transcriptional activity in the recruitment and retention of FACT during mitosis (Dirks and Snaar, 1999; Chan et al., 2012; Rošić et al., 2014; Chen et al., 2015; Liu et al., 2015; Molina et al., 2016; Bobkov et al., 2018).

## Cooperative and anti-cooperative FACT/CCAN binding

To further characterize how individual interactions between CCAN and FACT stabilize their assembly, we titrated CCAN subcomplexes in different combinations in an *in vitro* pull-down assay with $^{MBP}$FACT as a bait (Fig. 3 A). We quantified the results using the band intensities of CENP-M, CENP-U, CENP-L, and CENP-W, which were well resolved in SDS-PAGE gels, as representative of their cognate CCAN subcomplexes (Fig. 3, B–E). As shown above (Fig. 1 E), CENP-C$^{2-545}$HIKM and CENP-OPQUR bound FACT (Fig. 3 A, lanes 2, 3). A CENP-TW complex consisting only of the HFD of these proteins (CENP-T$^{458-C}$/full-length CENP-W, henceforth CENP-TW$^{HFD}$) also bound FACT (Fig. 3 A, lane 4). However, CENP-M, CENP-U, and CENP-W exhibited a markedly lower band intensity when their cognate subcomplex was exposed to FACT without the other CCAN subcomplexes (Fig. 3, A–E, lanes 2–4). When exposed to additional subcomplexes (lanes 5–12), stronger binding was

observed. Notably, the addition of CENP-C$^{2-545}$ to HIKM (instead of isolated CENP-HIKM) enhanced binding when certain subcomplexes were omitted (e.g., CENP-TW in lanes 5, 6; CENP-OPQUR and CENP-LN in lanes 7, 8; and CENP-OPQUR in lanes 9, 10), suggesting that CENP-C$^{2-545}$ stabilizes interactions of incomplete CCAN subcomplexes. Indeed, when the complete CCAN was used as a prey, the absence of CENP-C$^{2-545}$ did not significantly change the level of bound CCAN subunits (lanes 11, 12). Collectively, these results are consistent with the idea that FACT binding involves multiple interaction interfaces of CCAN.

CENP-C binds to CENP-HIKMLN through its proline–glutamic acid–serine–threonine (PEST)–rich region (CENP-C$^{189-400}$, CENP-C$^{PEST}$) (Klare et al., 2015; Cohen et al., 2008; Nagpal et al., 2015; McKinley et al., 2015) (Fig. 4 A). CENP-C$^{PEST}$, however, was neither capable of a direct interaction with FACT when combined with CENP-HIKM, nor did it trigger increased binding of CENP-HIKMLNOPQUR to $^{MBP}$FACT (Fig. 4 B, lanes 3, 6). These observations suggested that CENP-C and FACT bind directly outside the CENP-C$^{PEST}$. To identify regions of CENP-C involved in FACT binding, we divided the sequence of CENP-C into different fragments, expressed them as fusions to MBP, and used them as baits in a pull-down assay. FACT bound CENP-C$^{401-545}$, CENP-C$^{401-600}$, CENP-C$^{546-600}$, CENP-C$^{721-759}$, and CENP-C$^{721-943}$ (Fig. 4 C, lanes 3, 4, 5, 7, 8), which collectively encompass (1) the CENP-C central region, (2) a region adjacent to the central region, (3) the CENP-C conserved motif, and (4) the C-terminal cupin domain involved in dimerization. As both the central region and the CENP-C conserved motif bind specifically to CENP-A nucleosomes (CENP-A nucleosome core particle, CENP-A$^{NCP}$), these observations suggest that FACT stabilizes the CENP-A nucleosome binding region of CENP-C in the absence of nucleosomes (Fig. 4 A) (Trazzi et al., 2009; Carroll et al., 2010; Song et al., 2002; Kato et al., 2013). Confirming this conclusion, inclusion of CENP-A$^{NCP}$ in a pull-down assay where FACT was bound to immobilized $^{MBP}$CENP-C$^{EGFP}$ (a full-length CENP-C construct) caused FACT to dissociate (Fig. 4 D), indicating that binding of FACT and nucleosomes to CENP-C is mutually exclusive.

An unexpected aspect of the CCAN interaction with FACT is that the addition of CENP-OPQUR appeared to reduce the levels of CENP-HIKM and CENP-TW (using CENP-M and CENP-W as readouts, respectively; Fig. 3 A, lanes 11, 12, and quantified in Fig. 3, B and E). Within CCAN, CENP-OPQUR and CENP-TW do not directly bind to each other and require CENP-LN and CENP-HIKM for their interaction (Pesenti et al., 2018, 2022; Yatskevich et al., 2022). As they are both able to bind FACT, however, we anticipated that FACT may bridge these complexes. Contrary to this expectation, CENP-OPQUR and CENP-TW competed for FACT, with CENP-TW showing a higher affinity for FACT (Fig. 4 E, lanes 4, 5).

To investigate this phenomenon further, we tried to shed light on the determinants of the interaction of FACT with CENP-OPQUR. We found the disordered N-terminal tails of CENP-Q and CENP-U, known interaction hubs of the CENP-OPQUR complex (Kang et al., 2006; Amaro et al., 2010; Hua et al., 2011; Pesenti et al., 2018; Singh et al., 2021), to be required for FACT binding, because a truncation of these tails (CENP-OPQ$^{68-C}$U$^{115-}$

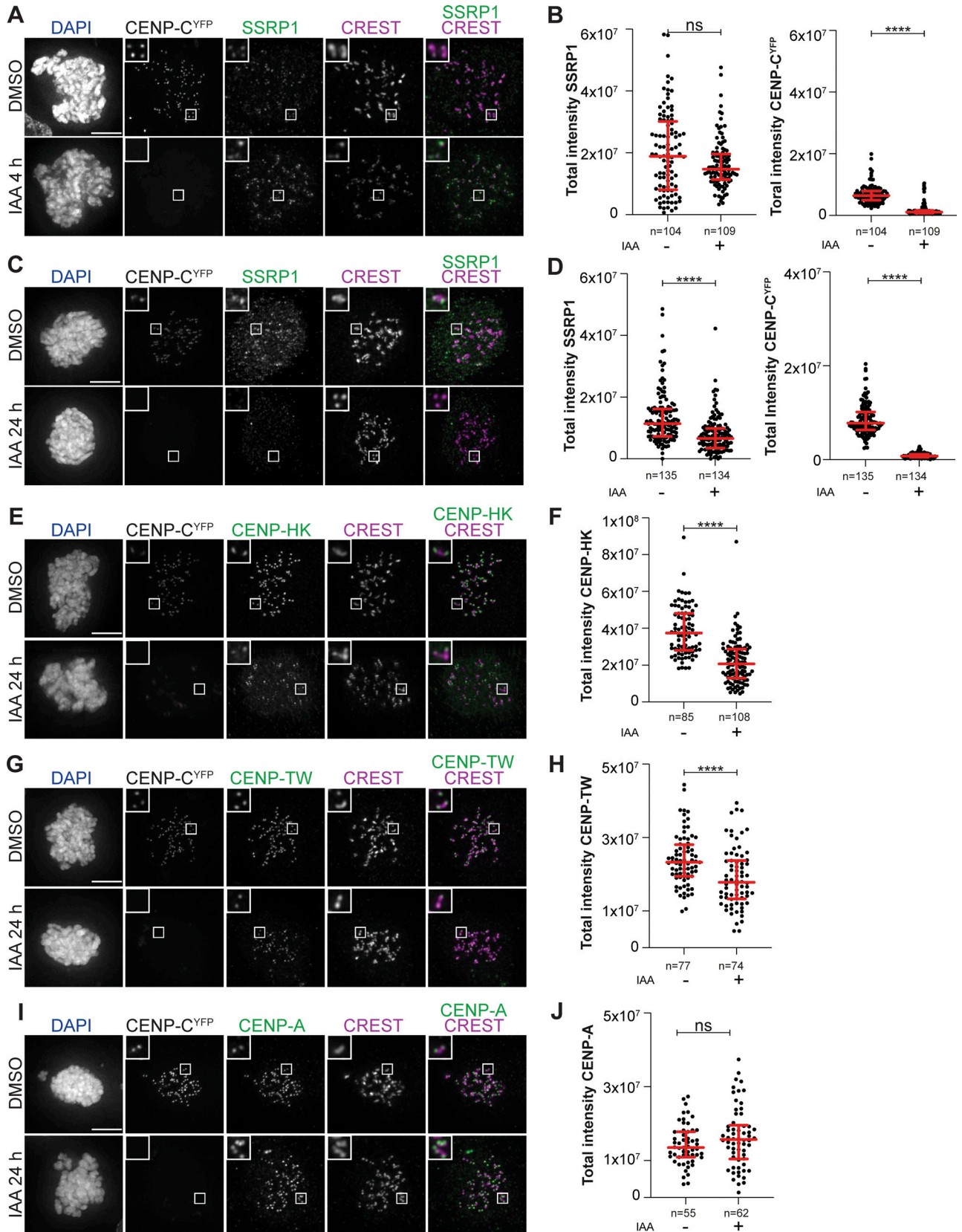

**Figure 2. Mitotic localization of FACT at the kinetochore depends on the CCAN. (A)** Representative images of localization of SSRP1 4 h after the addition of IAA to degrade CENP-C in DLD-1-CENP-C$^{YFP-AID}$ cells. Cells were treated with IAA (500 µM) to degrade endogenous CENP-C and nocodazole (3.3 µM) for 4 h to enrich for mitotic cells. CREST serum was used to visualize kinetochores, and DAPI to stain DNA. Three biological replicates were performed. Scale bar: 5 µm. **(B)** Scatter plots of YFP and SSRP1 levels at kinetochores for the experiment shown in A. *n* is the number of cells. **(C)** Representative images of localization of

SSRP1 24 h after the addition of IAA to degrade CENP-C in DLD-1-CENP-C[YFP-AID] cells. Cells were treated with IAA (500 µM) to degrade endogenous CENP-C for 24 h and nocodazole (3.3 µM) for 4 h to enrich for mitotic cells. CREST serum was used to visualize kinetochores, and DAPI to stain DNA. Three biological replicates were performed. Scale bar: 5 µm. **(D)** Scatter plots of YFP and SSRP1 levels at kinetochores for the experiment shown in C. *n* is the number of cells. **(E)** Representative images of localization of CENP-HK after degradation of CENP-C in DLD-1-CENP-C[YFP-AID] cells for 24 h. Cells were treated with IAA (500 µM) to degrade endogenous CENP-C for 24 h and nocodazole (3.3 µM) for 4 h to get mitotic population of cells. CREST was used to visualize kinetochores, and DAPI to stain DNA. Three biological replicates were performed. Scale bar: 5 µm. **(F)** Scatter plot of CENP-HK levels at kinetochores of the experiment shown in E. *n* is the number of cells. **(G)** Representative images of localization of CENP-TW after degradation of CENP-C in DLD-1-CENP-C[YFP-AID] cells for 24 h. Cells were treated with IAA (500 µM) to degrade endogenous CENP-C for 24 h and nocodazole (3.3 µM) for 4 h to get mitotic population of cells. CREST serum was used to visualize kinetochores, and DAPI to stain DNA. Three biological replicates were performed. Scale bar: 5 µm. **(H)** Scatter plot of CENP-TW levels at kinetochores of the experiment shown in G. *n* is the number of cells. **(I)** Representative images of localization of CENP-A after degradation of CENP-C in DLD-1-CENP-C[YFP-AID] cells for 24 h. Cells were treated with IAA (500 µM) to degrade endogenous CENP-C for 24 h. Nocodazole (3.3 µM) was added for 4 h to enrich for mitotic cells. CREST serum was used to visualize kinetochores, and DAPI to stain DNA. Three biological replicates were performed. Scale bar: 5 µm. **(J)** Scatter plot of CENP-A levels at kinetochores of the experiment shown in I. *n* is the number of cells. Statistical analysis was performed with a nonparametric *t* test comparing two unpaired groups (Mann–Whitney test). Symbols indicate $^{n.s.}P > 0.05$, $*P \le 0.05$, $**P \le 0.01$, $****P \le 0.0001$. Red bars represent the median and interquartile range.

[C]R, herewith indicated as CENP-OPQ[ΔN]U[ΔN]R) completely abolished the association with [MBP]FACT (Fig. 4 F, lane 3). This result was confirmed *in vivo*, where FACT was identified in immunoprecipitates of [EGFP]CENP-U but not of CENP-U[ΔN] (Fig. 4 G). However, CENP-C[2–545]HIKMLNOPQ[ΔN]U[ΔN]R bound [MBP]FACT, probably because CENP-C[2–545] provides sufficient binding affinity for the FACT complex. Even though CENP-OPQ[ΔN]U[ΔN]R does not bind FACT, it continued to oppose binding of FACT to

CENP-TW (Fig. 4 F, lanes 6, 7), possibly through an allosteric mechanism.

**FACT and CCAN interdependence for kinetochore localization**
As the mitotic localization of FACT to the kinetochore requires intact CCAN (Fig. 2, C and D), we wanted to investigate how individual CCAN subcomplexes contribute to FACT localization. RNA interference (RNAi) was used to deplete CCAN

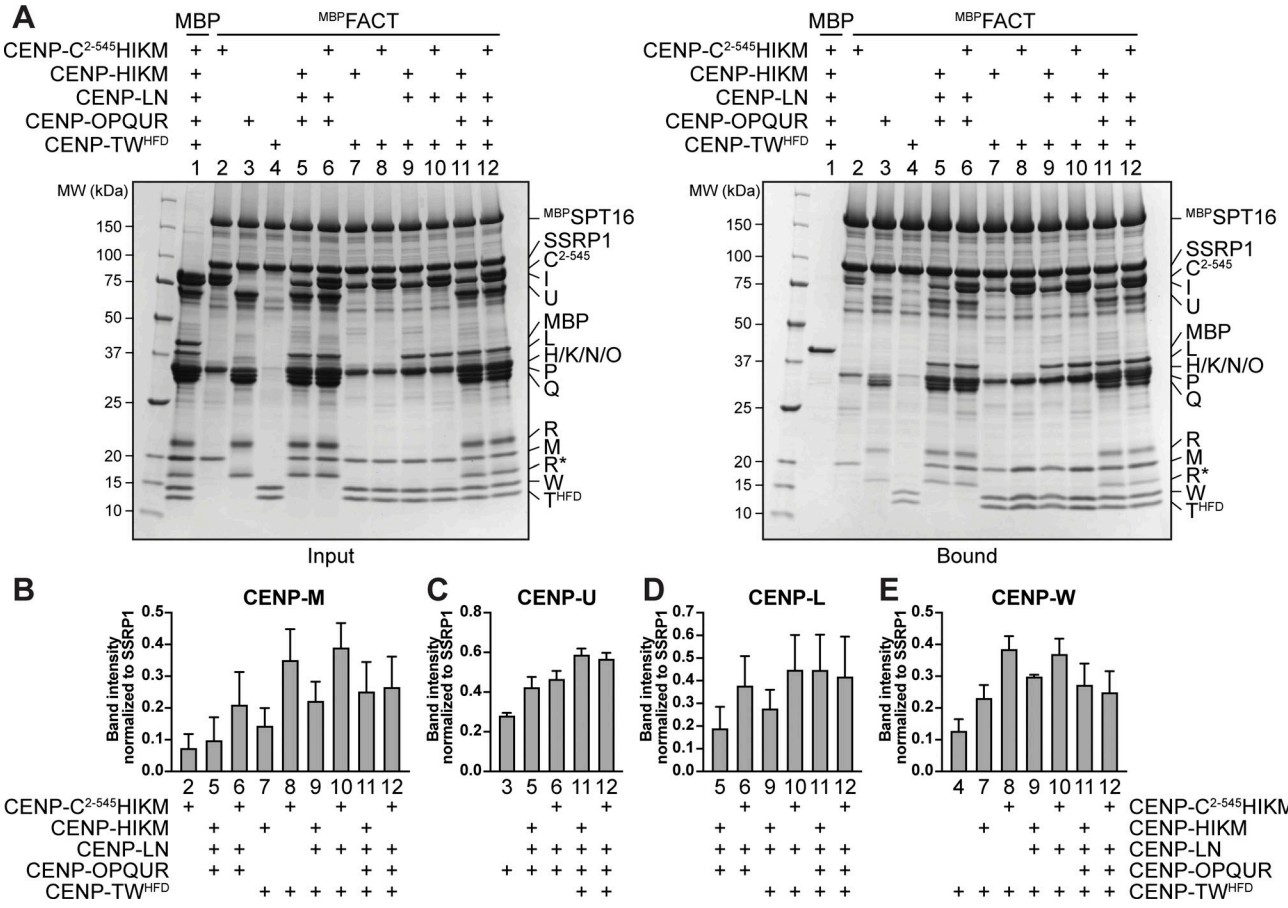

Figure 3. **CCAN binds FACT cooperatively. (A)** Amylose-resin pull-down assay with [MBP]FACT as bait and adding different combinations of CCAN subcomplexes as preys as indicated above the SDS-PAGE gels. MBP was used as a negative control. **(B–E)** Quantifications of the pull-down in A from three repeats. The band intensity of the target protein was normalized to SSRP1. One subunit per subcomplex was quantified: (B) CENP-M, (C) CENP-U, (D) CENP-L, and (E) CENP-W. Bars represent the mean and standard deviation. Source data are available for this figure: SourceData F3.

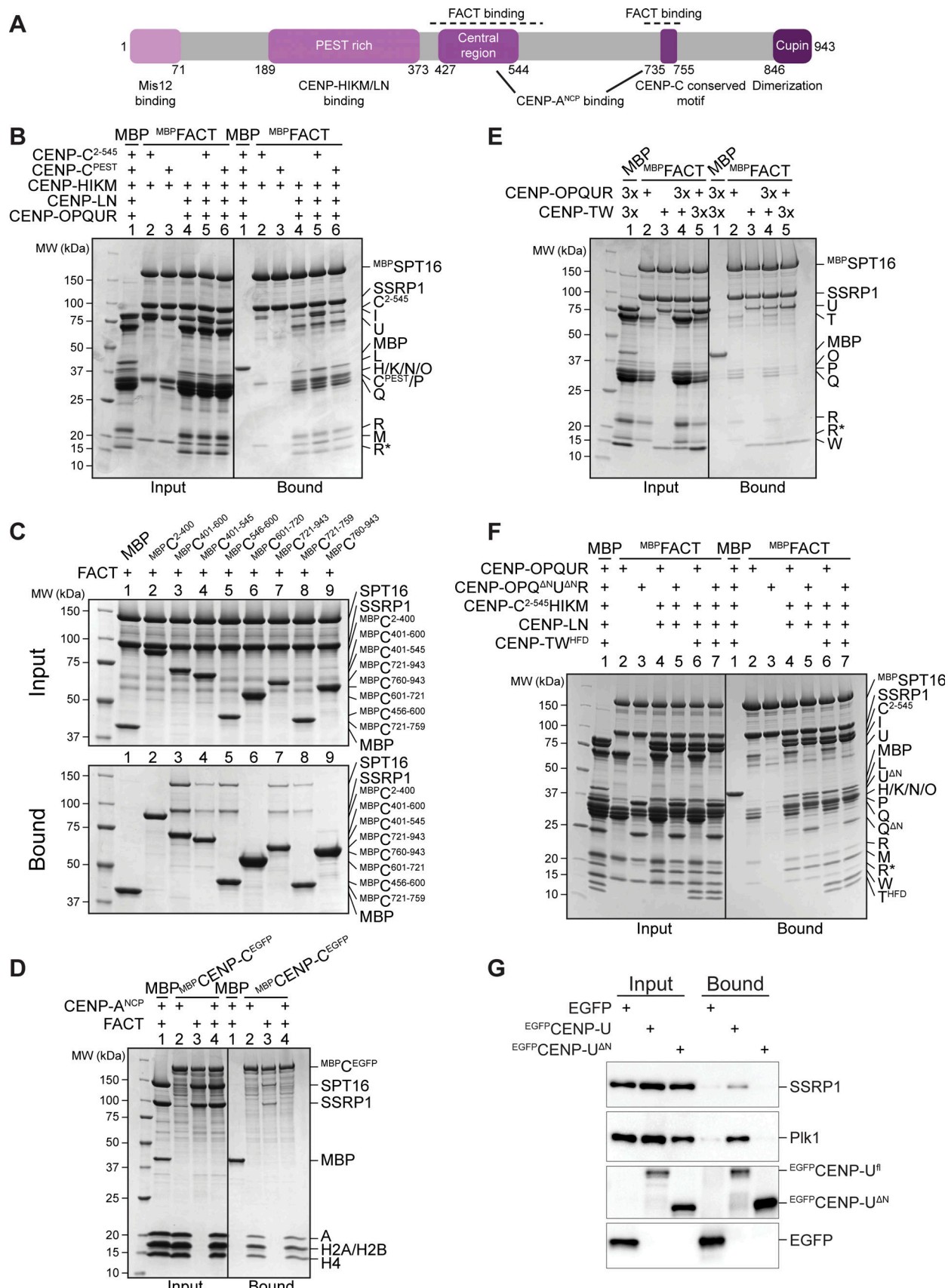

Figure 4. **CENP-C and the N-terminal tails of CENP-QU influence the FACT/CCAN interaction. (A)** Scheme of CENP-C with functional domains and their residue number indicated. Suggested FACT binding sites are indicated above the scheme. **(B)** Amylose-resin pull-down assay with ${}^{MBP}$FACT as a bait to assess

the binding of CENP-HIKM or CENP-HIKMLNOPQUR in the presence of CENP-C$^{2-545}$ or CENP-C$^{PEST}$. **(C)** Amylose-resin pull-down assay with a set of $^{MBP}$CENP-C fusion proteins as baits spanning the entire sequence of CENP-C and FACT as a prey. **(D)** Amylose-resin pull-down assay using immobilized $^{MBP}$CENP-C$^{EGFP}$ on beads and adding FACT and CENP-A$^{NCP}$ as preys. CENP-A$^{NCP}$ is the histone octamer reconstituted on a 145-bp Widom 601 sequence. **(E)** Amylose-resin pull-down assay with $^{MBP}$FACT as a bait and either CENP-OPQUR or CENP-TW added in molar excess. **(F)** Amylose-resin pull-down assay with $^{MBP}$FACT as a bait to analyze the influence of the CENP-QU N-terminal tails on the binding of CENP-OPQUR, CENP-C$^{2-545}$HIKMLNOPQUR, and CENP-C$^{2-545}$HIKMLNOPQURTW$^{HFD}$. **(G)** Lysates prepared from STLC-synchronized DLD-1 cells expressing EGFP alone or $^{EGFP}$CENP-U$^{fl}$ or 115-C were subjected to immunoprecipitation using GFP-trap beads followed by western blotting with antibodies against GFP, SSRP1, and PLK1. PLK1 was used as an internal control. Source data are available for this figure: SourceData F4.

---

subcomplexes in RPE-1 cells, and mitotic cells were immunostained for SSRP1. RPE-1 cells were treated with small interfering RNA (siRNA) against CENP-HIKM for 72 h to deplete the complex from the kinetochore (Fig. S3, A, B, and D). As a result, localization of SSRP1 was severely affected. CENP-OPQUR and CENP-TW localization was also substantially reduced, indicating that depletion of the CENP-HIKM subcomplex destabilizes CCAN (Fig. 5, A–C; and Fig. S3, E and F). Conversely, CENP-A was not perturbed upon CENP-HIKM depletion (Fig. S3, B and C).

A 60-h CENP-T RNAi treatment eliminated CENP-TW from the kinetochore (Fig. S3 G; and Fig. 5, D and F). Also in this case, a concomitant decrease in the kinetochore levels of FACT, CENP-HK, and CENP-O was observed, whereas the levels of CENP-A remained stable (Fig. 5, D and E; and Fig. S3, H–L). As shown above, acute depletion of CENP-C did not affect FACT localization, at least in the short term (Fig. 2, A and B). Despite its persistence at kinetochores upon CENP-HIKM or CENP-T depletion, CENP-C was also insufficient to retain FACT at kinetochores (Fig. 5, A–F). Thus, collectively, the CCAN subcomplexes are interdependent for their localization, in agreement with previous literature (McKinley et al., 2015; Pesenti et al., 2018; Okada et al., 2006; Basilico et al., 2014; Singh et al., 2021). Furthermore, our results demonstrate that FACT localization at the kinetochore during mitosis depends on the CCAN (Fig. 2, C and D; and Fig. 5, A–F).

To assess the potential contribution of CENP-OPQUR to the recruitment of FACT, we endogenously tagged both alleles of CENP-U with FKBP$^{F26V}$ and used the resulting cell line to rapidly degrade CENP-U through the addition of dTAG$^{V}$-1 (Nabet et al., 2018). A 24-h treatment led to the complete loss of CENP-U and CENP-R from the kinetochore, suggesting that the entire CENP-OPQUR complex, not only CENP-U, is removed. In agreement, Polo-like kinase 1 (PLK1) localization, which partially depends on CENP-OPQUR (Singh et al., 2021), decreased (Fig. S4, A–F). On the other hand, localization of CENP-A, CENP-TW, and CENP-HK did not require CENP-U (Fig. 5, G and I; and Fig. S4, G–I). In fact, CENP-TW displayed an increase in its kinetochore levels (Fig. 5 I). This may indicate competition between CENP-TW and CENP-OPQUR within the CCAN, but may also reflect a staining artifact caused by enhanced accessibility of the antigen. Finally, FACT localization was not affected by the depletion of CENP-U (Fig. 5, G and H), indicating that CENP-OPQUR is not necessary for recruiting or retaining FACT at the kinetochore, even if it interacts with FACT *in vivo*, as suggested by co-immunoprecipitation (Fig. 4 G). Alternatively, CENP-OPQUR and FACT may interact in a separate complex outside of CCAN. Thus, collectively, our results demonstrate the importance of

CCAN, even if we cannot point to a single CCAN subunit as a recruiter of FACT. Kinetochore localization of FACT is substantially reduced upon depletion of CENP-HIKM or CENP-TW, a condition that additionally triggers a reduction of CCAN stability. Conversely, CENP-C and CENP-OPQUR are not strictly required for the localization of FACT to kinetochores.

Our data so far indicate that CCAN promotes kinetochore recruitment of FACT. To assess whether FACT is also required to stabilize CCAN at the kinetochore, we used a previously established chronic myeloid leukemia K562 cell line where SSRP1 is endogenously tagged with a dTAG degron for rapid degradation (Žumer et al., 2024). The levels of CCAN subcomplexes were analyzed by immunofluorescence in mitotic cells 8 h after depleting FACT in S-trityl-L-cysteine (STLC)–arrested cells (Fig. 6, A and B). CENP-A, CENP-C, and CENP-HK were not significantly influenced by the rapid depletion of FACT (Fig. 6, E–I). On the contrary, kinetochore levels of CENP-TW were significantly decreased (Fig. 6, C and D). We also observed a minor reduction in the kinetochore levels of CENP-U, possibly caused by the absence of CENP-TW (Fig. S6, F and J). These results suggest a potential role of FACT in stabilizing CENP-TW at the centromere.

### FACT dimerization and Mid-AID domains are required for CCAN binding

The FACT subunits SPT16 and SSRP1 are multidomain distant paralogs with distinct functions (Zhou et al., 2020). To identify binding sites for CCAN, we produced different truncations or isolated domains of FACT (Fig. 7 A) and used them as preys in pull-down assays with CCAN subcomplexes as baits. As already observed, $^{MBP}$CENP-C$^{EGFP}$ and FACT bound directly (Fig. 4 D and Fig. 7 B, lane 1). Additionally, $^{MBP}$CENP-C$^{EGFP}$ pulled down SPT16$^{Mid-AID}$, a minimal SPT16, fragment (lane 3). It also pulled down, with apparently slightly higher affinity, construct 2 (SPT16$^{508-988}$/SSRP1$^{1-514}$, henceforth FACT$^{trunc}$) (Fig. 7, A and B, lane 2). GST-tagged CENP-OPQUR interacted robustly with FACT and FACT$^{trunc}$, but only weakly with SPT16$^{Mid-AID}$ (Fig. 7 C, lanes 1–3). Conversely, MBP-tagged CENP-TW ($^{MBP}$CENP-TW) was sufficient to bind SPT16$^{Mid-AID}$ (Fig. 7 D, lane 3). $^{MBP}$CENP-TW also bound strongly to FACT$^{trunc}$ and with low affinity to SSRP1$^{Mid-AID}$ (Fig. 7 D, lanes 2, 5). Thus, the association of $^{MBP}$CENP-TW with either SPT16 or SSRP1 Mid domain depended on the presence of an intact AID domain (Fig. 7 D, lanes 3–6). In summary, FACT$^{trunc}$ was sufficient to bind all CCAN subcomplexes, while the N-terminal aminopeptidase-like domain of SPT16 and the C-terminal HMG domain of SSRP1 are dispensable for CCAN binding (Fig. 7, B–D, lanes 7–9). In a reverse pull-down, the apparent strengths of the interaction of CCAN with

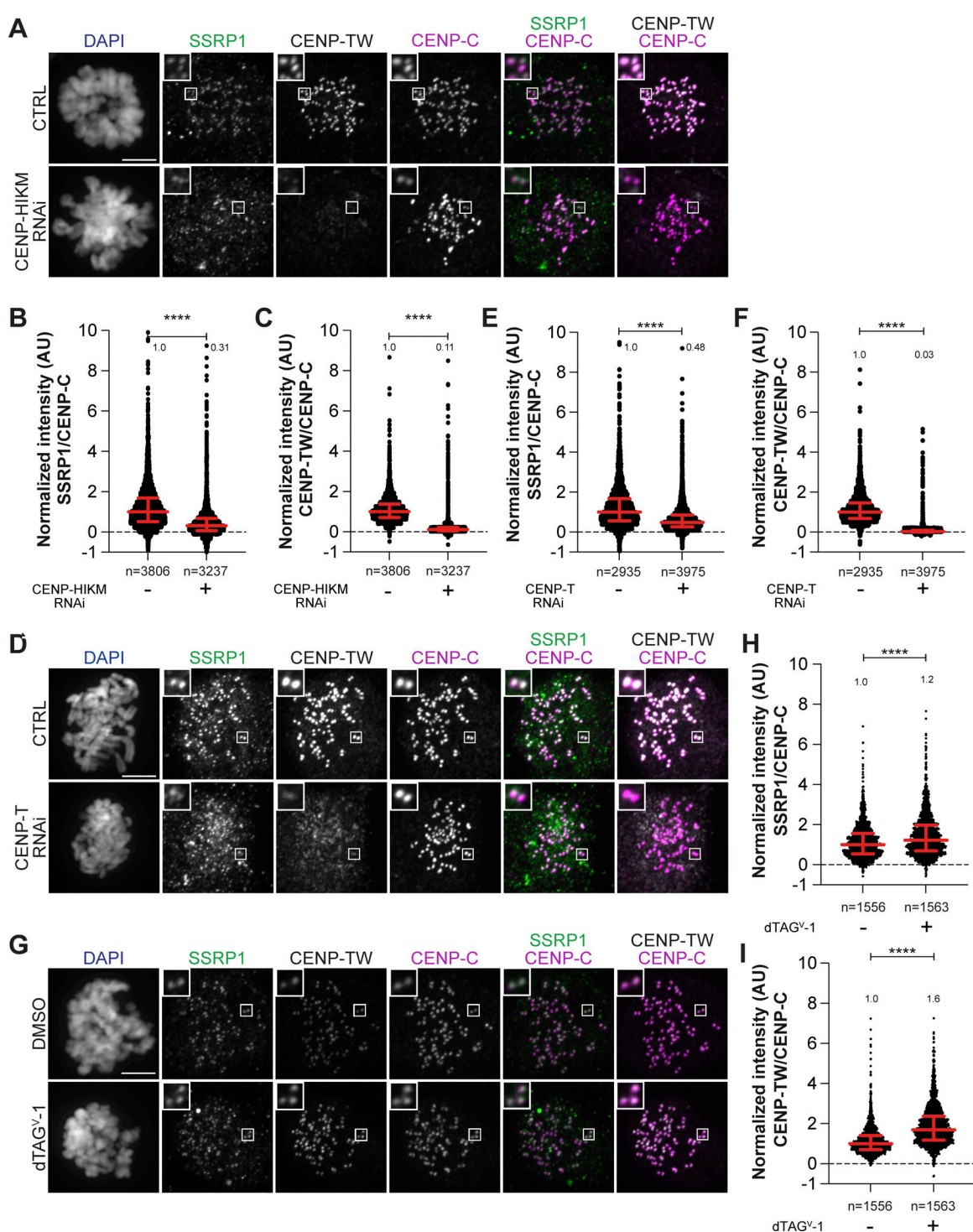

Figure 5. **Mitotic localization of FACT depends on CENP-HIKM and CENP-TW, but not on CENP-OPQUR. (A)** Representative images of localization of SSRP1 and CENP-TW after depletion of the CENP-HIKM complex in RPE-1 cells. CENP-HIKM RNAi was performed for 72 h using silencing oligonucleotides for each subunit at 30 nM concentration. Cells were treated with STLC (5 µM) for 16 h prior to fixation to obtain a mitotic population of cells. CENP-C identifies kinetochores, and DAPI stains DNA. Three biological replicates were performed. Scale bar: 5 µm. **(B)** Scatter plots of SSRP1 levels at kinetochores for the experiment shown in A. *n* refers to individually measured kinetochores. **(C)** Scatter plots show quantification of CENP-TW levels at kinetochores of the experiment shown in A. *n* refers to individually measured kinetochores. **(D)** Representative images of localization of SSRP1 and CENP-TW after depletion of the CENP-T complex in RPE-1 cells. CENP-T RNAi was performed for 60 h using oligos for each subunit at 30 nM concentration. Cells were treated with nocodazole (3.3 µM) for 4 h prior to fixation to obtain mitotic population. CENP-C was used to visualize kinetochores, and DAPI to stain DNA. Three biological replicates were performed. Scale bar: 5 µm. **(E)** Scatter plots show quantification of SSRP1 levels at kinetochores of the experiment shown in D. *n* is the number of individually measured kinetochores. **(F)** Scatter plots show quantification of CENP-TW levels at kinetochores of the experiment shown in D. *n* is the number of individually measured kinetochores. **(G)** Representative images of localization of SSRP1 and CENP-TW after depletion of the CENP-U complex in RPE-1-CENP-U-FKBP^F36V cells. Cells were treated with dTAG^V-1(500 nM) for 24 h to degrade endogenous CENP-U. Cells were treated with nocodazole (3.3 µM) for

4 h prior to fixation to obtain mitotic population. CENP-C identifies kinetochores, and DAPI stains DNA. Three biological replicates were performed. Scale bar: 5 µm. **(H)** Scatter plots of SSRP1 levels at kinetochores for the experiment shown in G. *n* is the number of individually measured kinetochores. **(I)** Scatter plots of CENP-TW levels at kinetochores for the experiment shown in G. *n* is the number of individually measured kinetochores. Statistical analysis was performed with a nonparametric *t* test comparing two unpaired groups (Mann–Whitney test). Symbols indicate [n.s.]P > 0.05, *P ≤ 0.05, **P ≤ 0.01, ****P ≤ 0.0001. Red bars represent the median and interquartile range.

either full-length $^{MBP}$FACT or $^{MBP}$FACT$^{trunc}$ were identical (Fig. 7 E).

**FACT requires phosphorylation by CK2 to interact with CCAN**

FACT is regulated by, and also directly binds to, acidophilic CK2 (Keller and Lu, 2002; Keller et al., 2001; Li et al., 2005; Mayanagi et al., 2019; Rusin et al., 2017). The CK2 holoenzyme is a tetramer composed of the active subunit CK2α or CK2α′ and the regulatory and dimerizing subunit CK2β (Graham and Litchfield, 2000). CK2 is a promiscuous kinase with hundreds of different substrates involved in numerous biological processes and diseases (Borgo et al., 2021). It is characterized as a constitutively active kinase, and its regulation is not defined by a single mechanism, but rather is substrate-specific (Roffey and Litchfield, 2021). Despite its localization to different cellular compartments, CK2 is mostly active in the nucleus (Faust and Montenarh, 2000; Martel et al., 2001), where it has a role in transcription (Schwind et al., 2015; Johnston et al., 2002).

Recombinant FACT purified from insect cells was strongly phosphorylated, but treatment with λ-phosphatase removed phosphorylation (Fig. S5 A, lane 2). The elution volume of FACT was unaffected by changes in its phosphorylation status (Fig. S5 B). Unexpectedly, dephosphorylated FACT (repurified to eliminate λ-phosphatase) failed to bind CCAN in an SEC co-elution assay (Fig. 8 A). The addition of CK2 to the reaction to induce phosphorylation of FACT restored the binding of FACT/CCAN in analytical SEC (Fig. 8 A). The phosphorylation dependency of FACT/CCAN complex formation was corroborated in a solid-phase assay (Fig. 8 B, lanes 11–13). This assay was also used to probe the phosphorylation dependency of the interaction of FACT with specific CCAN subcomplexes. Dephosphorylated $^{MBP}$FACT failed to pull down CENP-C$^{2–545}$HIKM, CENP-OPQUR, and CENP-TW. These interactions were partially restored upon CK2 phosphorylation, although not to the levels observed with the sample before dephosphorylation (Fig. 8 B, lanes 2–10), probably due to incomplete rephosphorylation (Fig. S5 C). Some CCAN subunits, including CENP-C and/or CENP-I, CENP-U, and CENP-T, were also phosphorylated by CK2 (Fig. S5 C, lanes 4, 7, 10). Of note, additional kinases demonstrated an ability to phosphorylate FACT, but they failed to restore the interaction with CCAN, indicating that the effects on CCAN binding are specific to CK2 (Fig. S5 D).

FACT constructs described in Fig. 7 were used to analyze which regions of the complex are phosphorylated by CK2. SPT16 and SSRP1 Mid-AID domains were phosphorylated in an AID-dependent manner (Fig. 8 C, samples 3–6), in agreement with the ability of CK2 to phosphorylate acidic sequences (Kuenzel et al., 1987). Additionally, the C-terminal region of SSRP1 was also phosphorylated by CK2 (Fig. 8 C, sample 9). The phosphorylated FACT complex was subjected to mass spectrometry

analysis to identify target sites. This analysis failed to identify the precise phosphorylation site within the AID sequences, but in combination with sequence-based prediction (Blom et al., 1999, 2004; Obenauer et al., 2003) and published phosphorylation sites (Li et al., 2005; Rusin et al., 2017), we propose more than 20 potential CK2 phosphorylation sites on FACT (Fig. 8 D and Table S1). This large number of phosphorylation sites, expected to render FACT more negatively charged, likely facilitates interactions with positively charged DNA-binding interfaces of CCAN. Due to the constitutive activity of CK2 (Roffey and Litchfield, 2021), cellular conditions leading to phosphorylation of FACT by CK2 remain unclear.

**DNA competes with FACT for CCAN binding**

The CCAN recognizes the centromere by direct binding of CENP-A nucleosomes through CENP-C and CENP-N while also binding to DNA (Trazzi et al., 2009; Carroll et al., 2009, 2010; Klare et al., 2015; Pentakota et al., 2017; Song et al., 2002; Kato et al., 2013; Pesenti et al., 2022), while FACT only binds to nucleosomes that are partially destabilized, e.g., by an actively transcribing RNA polymerase II (Tsunaka et al., 2016; Wang et al., 2018; Farnung et al., 2021; Jeronimo et al., 2021; Žumer et al., 2024). We set out to further dissect FACT's interaction with CCAN on chromatin using biochemical reconstitution. In analytical SEC, the addition of a 145-bp DNA fragment prevented the assembly of the FACT/CCAN complex altogether, as DNA binding to CCAN displaced FACT from the complex (Fig. 9 A). The same effect was observed upon the addition of a 75-bp DNA fragment (Fig. S6 A). This result was confirmed in a solid-phase assay (Fig. 9 B, lanes 8, 9). CCAN binds DNA very tightly, whereas individual subcomplexes bind to DNA with much lower affinity, if at all (Pesenti et al., 2022; Yatskevich et al., 2022). We therefore asked how DNA influenced the interaction of FACT with the CCAN subcomplexes CENP-C$^{2–545}$HIKM, CENP-OPQUR, and CENP-TW. Although a marginal reduction in binding of each complex was detected (Fig. 9 B, lanes 2–7), the effect of DNA was considerably less pronounced than in the presence of the complete CCAN (Fig. 9 B).

Next, we tested whether CCAN binding to FACT is compatible with binding to a CENP-A nucleosome core particle (on a 145-bp Widom 601 sequence). Recombinant FACT did not bind to DNA or intact nucleosomes in analytical SEC (Fig. S6 B). When mixed with CCAN and a CENP-A$^{NCP}$, however, a tripartite complex was formed (Fig. 9 C). These observations were corroborated using a pull-down assay, where CENP-A$^{NCP}$ was seen to interact with $^{MBP}$FACT through CCAN (Fig. 9 D, lane 4). The association of CCAN subcomplexes was also evaluated. CENP-OPQUR and CENP-TW did not bind to nucleosomes, and the interaction with FACT was essentially unaltered (Fig. 9 D, lanes 7–10). Binding of CENP-C$^{2–545}$HIKM was marginally reduced

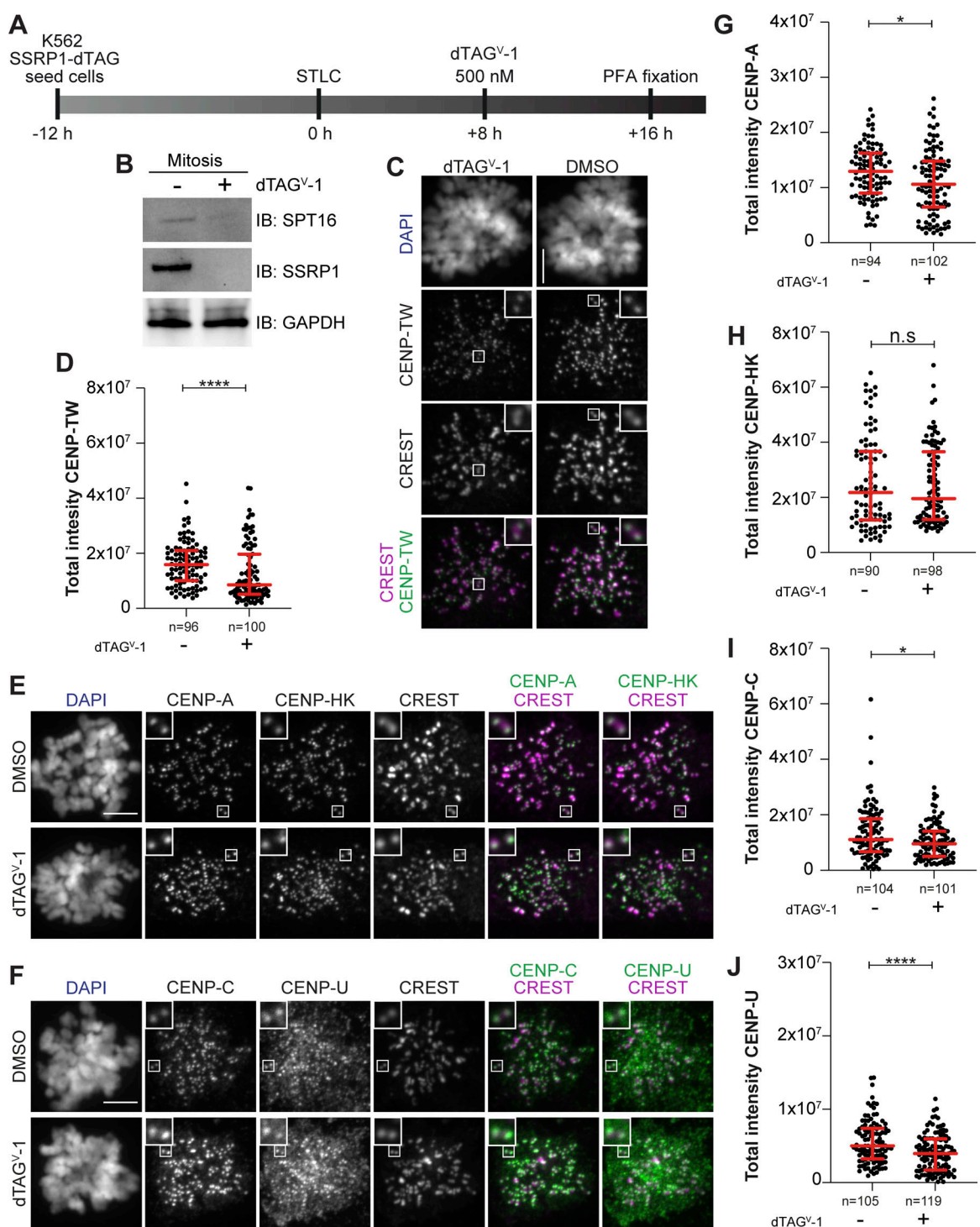

Figure 6. **Mitotic degradation of FACT affects CENP-TW stability. (A)** Schematic representation of experimental scheme used for mitotic SSRP1 dTAG^V-1 treatment. **(B)** Western blot analysis of K562 cells treated with dTAG^V-1 demonstrating the degradation of SSRP1 and a concomitant decrease in SPT16. GAPDH was used as a loading control. **(C)** Representative images of localization of CENP-TW after SSRP1 degradation in K562-SSRP1-dTAG cells. Cells arrested in prometaphase by STLC were treated with dTAG^V-1 (500 nM) for 8 h prior to fixation. CREST serum was used to visualize kinetochores, and DAPI to stain DNA. Three biological replicates were performed. Scale bar: 5 μM. **(D)** Scatter plot of CENP-TW levels for the experiment shown in C. *n* is the number of cells. **(E)** Representative images of localization of CENP-A and CENP-HK after depletion of SSRP1 in K562-SSRP1-dTAG cells. Cells were first arrested in prometaphase by STLC followed by treatment with dTAG^V-1 (500 nM) for 8 h prior to fixation. CREST serum was used to visualize kinetochores, and DAPI to stain DNA. Three biological replicates were performed. Scale bar: 5 μm. **(F)** Representative images of localization of CENP-C and CENP-U after depletion of SSRP1 in K562-SSRP1-dTAG cells. Cells were first arrested in prometaphase by STLC followed by treatment with dTAG^V-1 (500 nM) for 8 h prior to fixation. CREST serum was used to visualize kinetochores, and DAPI to stain DNA. Three biological replicates were performed. Scale bar: 5 μm. **(G)** Scatter plot of CENP-A levels at kinetochores for the experiment shown in E. *n* is the number of cells. **(H)** Scatter plot of CENP-HK levels at kinetochores for the experiment shown in E. *n* is the number of cells. **(I)** Scatter plot of CENP-C levels at kinetochores for the experiment shown in F. *n* is the number of cells. **(J)** Scatter plot of CENP-U levels at

kinetochores for the experiment shown in F. *n* is the number of cells. Statistical analysis was performed with a nonparametric *t* test comparing two unpaired groups (Mann–Whitney test). Symbols indicate n.s.P > 0.05, *P ≤ 0.05, **P ≤ 0.01, ****P ≤ 0.0001. Red bars represent the median and interquartile range. Source data are available for this figure: SourceData F6.

upon the addition of CENP-A$^{NCP}$ (Fig. 9 D, lanes 5, 6). We ligated a naked DNA sequence to a CENP-A nucleosome built on α-satellite DNA, creating a 348-bp sequence of which roughly half was embedded in a nucleosome. The overhanging DNA acted comparably to free DNA, effectively displacing FACT from a CCAN/CENP-A$^{NCP}$ complex (Fig. S6, C and D). Collectively, these results suggest that FACT may recognize a form of CCAN that is not directly bound to DNA at centromeres, despite substantial biochemical and structural information indicating that CCAN binds DNA tightly through the CENP-LN vault and potentially through various neighboring DNA-binding structures (Pesenti et al., 2022; Yatskevich et al., 2022; Tian et al., 2022; Dendooven et al., 2023). Whether a form of CCAN devoid of DNA is present at kinetochores during mitosis, however, remains unclear.

### FACT cannot bind centromeric histones and CCAN simultaneously

As a histone chaperone, FACT is known to bind to both H2A/H2B dimers and H3/H4 tetramers (Kemble et al., 2015; Tsunaka et al., 2016). H2A/H2B and CENP-TW compete for the same binding site on FACT, and FACT has a binding preference for the former (Prendergast et al., 2016). We wanted to broaden our analysis of the FACT/CCAN complex in relation to centromeric histones. A trimeric complex of histones CENP-A/H4 with the first 80 residues of its chaperone HJURP was added to a pull-down assay with $^{MBP}$FACT as a bait and CCAN as a prey. CENP-A/H4/HJURP$^{1–80}$ outcompeted CCAN in FACT binding (Fig. 9 E, lane 3). This result was confirmed in an orthogonal assay, where $^{MBP}$HJURP$^{1–80}$ in complex with CENP-A/H4 as a bait bound FACT efficiently, but CCAN was excluded from the complex (Fig. S6 E, lane 4). $^{MBP}$HJURP$^{1–80}$ in the absence of CENP-A/H4, used as a control, did not bind FACT nor CCAN (Fig. S6 E, lane 1), indicating that competition with CCAN for FACT binding is caused by the CENP-A/H4 dimer rather than the chaperone. Binding of CCAN subcomplexes to FACT was also disrupted upon the addition of CENP-A/H4/HJURP$^{1–80}$ (Fig. 9 E, lanes 4–9), suggesting that CENP-A/H4 and CCAN may share the same binding site on FACT. This was validated by testing the binding of different FACT constructs to CENP-A/H4/$^{MBP}$HJURP$^{1–80}$. Indeed, Mid domains and AID segments of SPT16, but also SSRP1, were important for binding to centromeric histones (Fig. S6 F). These data suggest that FACT is not able to chaperone CENP-A/H4 and CCAN simultaneously.

## Discussion

FACT has established roles in transcription, replication, and DNA repair. On the other hand, the functional significance of its enrichment at human kinetochores remains unclear. We identified novel interactions of FACT with CCAN subunits CENP-C and CENP-OPQUR in addition to the previously reported binding

to the HFD-containing complex CENP-TW (Prendergast et al., 2016). These interactions act cooperatively to form a stable 18-subunit complex (Fig. 10). During mitosis, FACT localizes to the kinetochore in a CCAN-dependent manner, and the CENP-HIKM and CENP-TW complexes are especially important for this localization. Furthermore, FACT depletion results in a significant reduction of CENP-TW at the kinetochore during mitosis. The direct and specific interaction of FACT with the kinetochore suggests a role linked to kinetochore assembly and the regulation of kinetochore interactions with centromeric chromatin. The exact function of kinetochore FACT will have to be elucidated, but our study paves the way for more detailed functional analyses.

*In vitro*, FACT was displaced from the CCAN by DNA, suggesting that CCAN is not stably anchored to chromatin, while FACT is bound (Fig. 10). This is unexpected, as FACT localizes to the kinetochore in a CCAN-dependent manner during mitosis (Fig. 2, C and D), a time when we expect a tight connection between kinetochores and centromeric DNA. Our results may suggest that a subset of CCAN complexes is engaged with FACT rather than with DNA. There is only limited information on the mechanism and timing of recruitment of CCAN to the centromere during or after DNA replication. In line with its chaperone activity, FACT may stabilize the CCAN or CCAN subunits in solution and help in the deposition and assembly of the CCAN at the centromere. We observed that while individual interaction partners had a lower affinity for FACT than the entire CCAN, CENP-C$^{2–545}$HIKMLNTW$^{HFD}$ was the strongest binder, while CENP-TW and CENP-OPQUR were moderately anti-cooperative and competing for FACT binding (Fig. 3 and Fig. 4 E).

Our observations also indicate that individual interactions between FACT and CCAN ultimately cooperate to enhance overall binding (Fig. 10). For instance, we suspect that the interaction of FACT with CENP-OPQUR may undergo a rearrangement inside the CCAN in comparison with the isolated FACT/CENP-OPQUR complex. Thus, FACT may preferentially bind to CCAN complexes that are not fully assembled or properly incorporated into centromeric chromatin. This possibility may also partly explain some discrepancies between the results of binding assays *in vitro* and the analysis of FACT localization after displacement of CCAN subunits *in vivo*. For instance, the CENP-HIKM complex appeared to have a disproportionate effect on FACT localization if gauged against the apparently low binding affinity for FACT *in vitro*. Given the position of CENP-HIKM in the CCAN hierarchy, which is upstream compared with other subcomplexes, it is reasonable to assume that its depletion would lead to a more significant effect on FACT recruitment to the kinetochore. We surmise that depletion of CENP-HIKM may indirectly affect the interaction of CENP-TW with DNA, causing FACT displacement indirectly.

Upon re-entry into interphase, FACT localizes to the entire chromosome, more pronouncedly around nucleoli (Birch et al.,

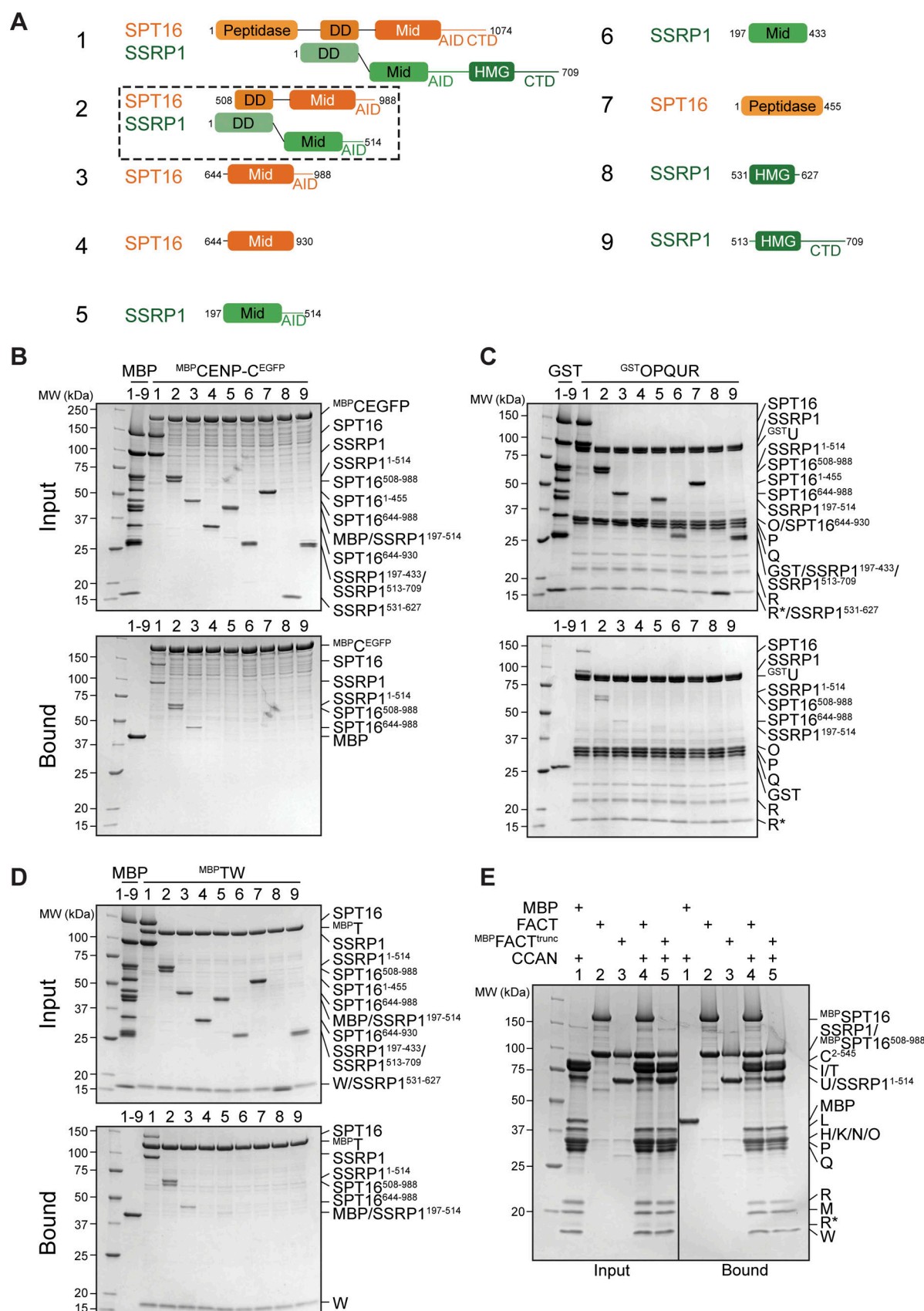

**Figure 7.** **CCAN binding requires FACT dimerization and Mid-AID domains. (A)** Scheme of FACT domains or truncations used as a prey in the following pull-down assays. **(B)** Amylose-resin pull-down assay with $^{MBP}$CENP-C$^{EGFP}$ to analyze binding of the FACT constructs in A. **(C)** Glutathione-agarose pull-down

assay using CENP-OPQUR with CENP-U fused to an N-terminal GST as a bait and FACT constructs in A as preys. **(D)** Amylose-resin pull-down assay with $^{MBP}$CENP-T/CENP-W as a bait and FACT constructs in A as preys. **(E)** Amylose-resin pull-down assay comparing CCAN binding to $^{MBP}$FACT and $^{MBP}$FACT$^{trunc}$. $^{MBP}$FACT$^{trunc}$ corresponds to construct 2 in A with an N-terminal MBP fusion on SPT16. Source data are available for this figure: SourceData F7.

2009; Jeong et al., 2022) (Fig. 1 F). We used a pre-extraction strategy to visualize FACT at the kinetochore during mitosis. This treatment removes FACT from the rest of the chromatin and suggests indirectly that FACT is more stably bound to chromatin in interphase. Our inability to visualize FACT at kinetochores outside of mitosis does not necessarily imply depletion of FACT from these structures, as visualizing kinetochore FACT by immunofluorescence during interphase against a more pronounced chromosome signal is technically challenging. As CCAN localizes to the centromere throughout the cell cycle (Foltz et al., 2006), it will be important to establish whether FACT acts there outside of mitosis. Chromatin is considered transcriptionally silent in mitosis (Gottesfeld and Forbes, 1997), but it has been suggested that centromeric transcription is also active during mitosis (Dirks and Snaar, 1999; Chan et al., 2012; Liu et al., 2015; Molina et al., 2016). Thus, it is possible that the localization of FACT at the kinetochore coincides with active centromeric transcription in mitosis and interphase. CENP-TW localization was reduced by acute depletion of FACT in mitosis. Remarkably, other CCAN subcomplexes were retained at the centromere, aside from a minor decrease in CENP-U localization. As CENP-TWSX is integrated into the specialized centromeric chromatin (Nishino et al., 2012), FACT may stabilize CENP-TW by preventing its loss as RNA polymerase II passes through centromeric chromatin. This would be reminiscent of FACT's known role in preventing histone loss during transcription (Hsieh et al., 2013; Belotserkovskaya et al., 2003). Whether FACT is essential for the stability of the whole CCAN over a longer period of time is currently unclear.

Due to FACT's role in multiple chromatin-related mechanisms, studying its specific role at the kinetochore is challenging. In the future, it will be essential to identify separation-of-function mutants to target specific functions of FACT. The AID of SPT16 and phosphorylation by CK2 are important for other functions in addition to mediating the interaction with CCAN and are therefore not appropriate targets for mutations. Investigation of potential CK2 sites on FACT may ultimately identify sites that are solely important for CCAN binding. Finally, structural information on a FACT/CCAN complex could facilitate the identification of specific interaction interfaces. So far, our efforts to obtain high-resolution structures of the CCAN/FACT complex have been thwarted by the lack of order of the resulting complexes.

The phosphorylation of FACT by CK2 is indispensable for FACT/CCAN complex formation (Fig. 10). *In vitro*, binding of DNA to CCAN leads to the dissociation of FACT, while FACT preferentially binds to centromeric histones, which share binding sites with CCAN. Collectively, these data suggest that FACT impacts the kinetochore directly rather than sharing the same function at the centromere as in other parts of chromatin. FACT is predicted to possess up to 20 or more CK2

phosphorylation sites, especially in the AID sequences (Fig. 8 D). Currently, it remains uncertain which of these sites are crucial for the binding to CCAN. Nevertheless, phosphorylation of FACT is also important for other functions (Keller and Lu, 2002; Keller et al., 2001; Li et al., 2005; Mayanagi et al., 2019; Tsunaka et al., 2009) and it is not clear whether the different CK2 sites on FACT are functionally related. Interestingly, phosphorylation of FACT reduces its DNA-binding activity (Li et al., 2005; Tsunaka et al., 2009). It is possible that FACT changes its exact localization from DNA to histones or the CCAN depending on its phosphorylation state. Alternatively, different pools of FACT may accomplish different functions simultaneously. For instance, one pool may bind CCAN, while another may bind CENP-A/H4 or other histones during transcription and replication. In summary, we provided a characterization of the FACT/CCAN interaction *in vitro* and *in vivo*, and set the basis for future work aiming to dissect this interaction.

## Materials and methods

### Plasmids

Plasmids for the expression of CENP-C$^{2-545}$HIKM, CENP-C$^{2-545}$, CENP-C$^{189-400}$, $^{MBP}$CENP-C$^{721-C}$, CENP-HIKM, CENP-LN, CENP-OPQUR, CENP-OPQ$^{68-C}$U$^{115-C}$R CENP-TWSX, CENP-TW, CENP-T$^{458-C}$W, CENP-SX, $^{MBP}$CENP-T/W, CENP-A/H4, H2A/H2B, CDK1, cyclin B, CKS1, PLK1, and Aurora B$^{45-344}$/INCENP$^{835-903}$ and for the production of DNA sequences were generated as previously described (Klare et al., 2015; Pentakota et al., 2017; Walstein et al., 2021; Pesenti et al., 2022, 2018; Weir et al., 2016; Basilico et al., 2014; Singh et al., 2021; Huis In't Veld et al., 2022; Girdler et al., 2008). Plasmids expressing human CK2α$^{1-335}$ and CK2β$^{1-193}$ were a kind gift of K. Niefind (University of Cologne, Cologne, Germany). SPT16 and SSRP1 with an N-terminal His-tag and a TEV cleavage site were cloned into a pFL-derived MultiBac vector (Fitzgerald et al., 2006). Sequences of $^{MBP-TEV}$SPT16, $^{His-TEV}$SSRP1, $^{His-TEV}$SPT16$^{644-988}$ (Mid-AID), $^{His-TEV}$SPT16$^{644-930}$ (Mid), $^{His-TEV}$SSRP1$^{197-514}$ (Mid-AID), $^{His-TEV}$SSRP1$^{197-433}$ (Mid), $^{His-TEV}$SPT16$^{508-988}$, $^{MBP-TEV}$SPT16$^{508-988}$, SSRP1$^{1-514}$, $^{His-TEV}$SSRP1$^{1-514}$, $^{His-MBP}$CENP-C$^{EGFP}$, and $^{GST}$CENP-U were inserted into pLIB vectors. These were used to combine $^{MBP-TEV}$SPT16+$^{His-TEV}$SSRP1 ($^{MBP}$FACT), $^{His-TEV}$SPT16$^{508-988}$+ SSRP1$^{1-514}$ (FACT$^{trunc}$), $^{MBP-TEV}$SPT16$^{508-988}$+ $^{His-TEV}$SSRP1$^{1-514}$ ($^{MBP}$FACT$^{trunc}$), and $^{GST}$CENP-U with previously described CENP-O/P/Q/R and $^{His-TEV}$CENP-Q$^{68-C}$ and CENP-U$^{115-C}$ with CENP-O/P/R (Pesenti et al., 2018) in pBIG1a vectors for baculovirus-based multigene expression (Weissmann et al., 2016). $^{His-PreSc}$SPT16$^{1-455}$ (peptidase-like domain), $^{His-PreSc}$SSRP1$^{531-627}$ (HMG domain), $^{His-PreSc}$SSRP1$^{513-C}$, $^{MBP}$CENP-C$^{2-400-His}$, $^{MBP}$CENP-C$^{401-600-His}$, $^{MBP}$CENP-C$^{401-545-His}$, $^{MBP}$CENP-C$^{546-600-His}$, $^{MBP}$CENP-C$^{601-720-His}$, $^{MBP}$CENP-C$^{721-759-His}$, $^{His-MBP-TEV}$CENP-C$^{760-C}$, $^{MBP-TEV}$HJURP$^{1-80-His}$, and $^{His-PreSc}$CENP-A co-expressed with H4 and $^{MBP-TEV}$HJURP$^{1-80}$ were cloned into a pETDuet

Figure 8. **The FACT/CCAN interaction requires phosphorylation of FACT by CK2. (A)** Analytical SEC comparing CCAN binding of untreated FACT, dephosphorylated FACT, repurified FACT, and dephosphorylated FACT, where the sample is treated with CK2. The experiment was performed with samples in Fig. 1 D. The first three samples are the same as Fig. 1 D. **(B)** Amylose-resin pull-down assay using either untreated MBPFACT or dephosphorylated MBPFACT as a bait and CCAN subcomplexes or CCAN as preys. λ-PP indicates that MBPFACT had been dephosphorylated and λ-PP had been removed by SEC before the experiment. CK2 indicates that the sample was treated with CK2. **(C)** FACT constructs (Fig. 7 A) were treated with CK2, and the phosphorylation state was assessed by Pro-Q Diamond staining. Coomassie staining of the corresponding SDS-PAGE gels is shown below. **(D)** Scheme of FACT with sequences of potential CK2 phosphorylation sites indicated. CK2 sites were based on mass spectrometry, and predictions or previous publications (Li et al., 2005). SSRP1 S437 and S444 were identified in our mass spectrometric analysis (Table S1) and were also reported in a previous publication (Rusin et al., 2017). Source data are available for this figure: SourceData F8.

vector using the Gibson cloning (Gibson et al., 2009). For the mammalian expression of N-terminally tagged EGFP-SSRP1, the SSRP1 sequence was obtained by PCR and subcloned in-frame with the sequence encoding the EGFP-tag in pCDNA5/FRT/TO-EGFP-IRES, a previously modified version (Krenn et al., 2012) of the pCDNA5/FRT/TO vector (Invitrogen).

**Purification of DNA fragments**

Generation of the 145-bp Widom 601 (5′-ATCAGAATCCCGGTG CCGAGGCCGCTCAATTG-3′, 5′-GTCGTAGACAGCTCTAGCACC GCTTAAACGCACGTACGCGCTGTCCCCCGCGTTTTAACCGCC AAGGGGATTACTCCCTAGTCTCCAGGCACGTGTCAGATATAT ACATCGAT-3′), and the 75-bp (5′-ATCCGTGGTAGAATAGGA

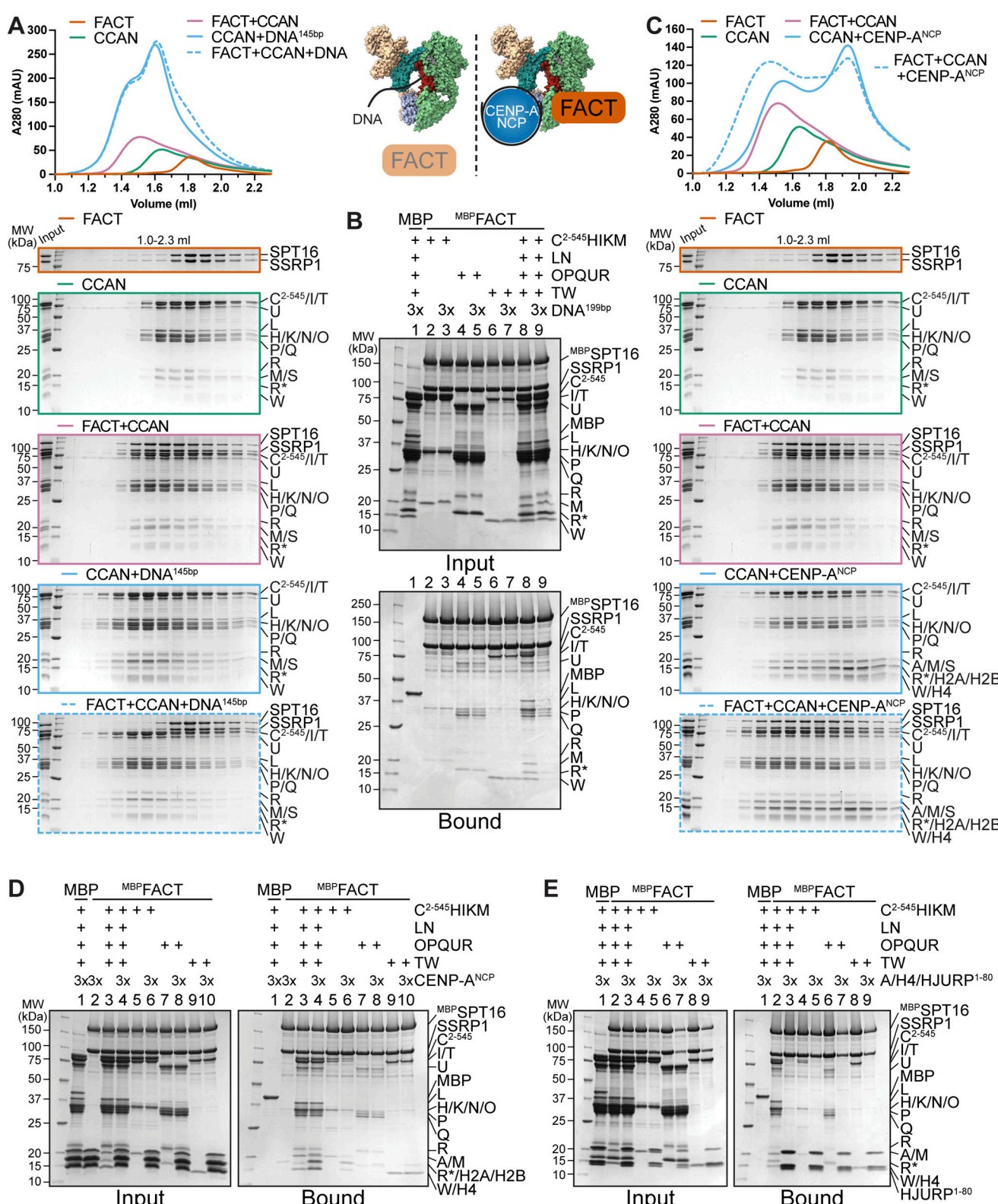

**Figure 9. FACT competes with DNA for CCAN binding, and CCAN competes with centromeric histones for FACT binding. (A)** Analytical SEC to test the effect of DNA on the FACT/CCAN complex. A 145-bp Widom 601 sequence was used. The left part of the scheme illustrates the result. **(B)** Amylose-resin pull-down assay with MBPFACT as a bait and CCAN complexes or full CCAN and DNA as preys. A 199-bp CEN1-like DNA sequence was used. **(C)** Analytical SEC to assess reconstitution of a FACT/CCAN/CENP-ANCP complex. The experiment is part of a larger experiment that includes the experiment in A. The CENP-A/H4/H2A/H2B histone octamer was reconstituted on a 145-bp Widom 601 sequence. The SDS-PAGE gels of FACT, CCAN, and FACT+CCAN are duplicates of those shown in A. The chromatograms are also the same but displayed on a different scale. The right part of the scheme in A illustrates the result. **(D)** Amylose-resin pull-down assay of MBPFACT and CCAN subcomplexes or CCAN with the addition of an excess of CENP-ANCP. **(E)** Amylose-resin pull-down assay of MBPFACT and CCAN subcomplexes or CCAN with the addition of an excess of CENP-A/H4/HJURP[1–80]. Source data are available for this figure: SourceData F9.

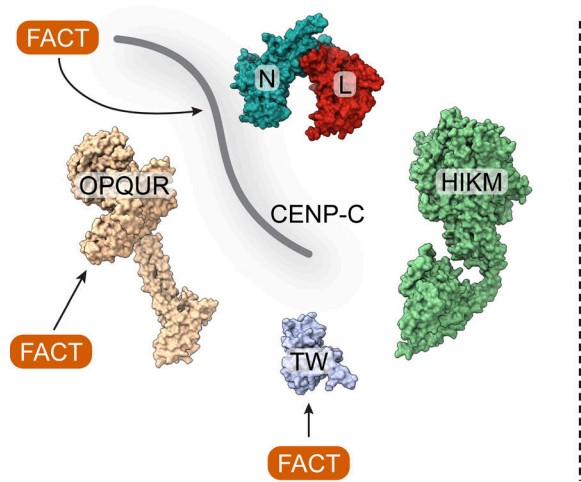
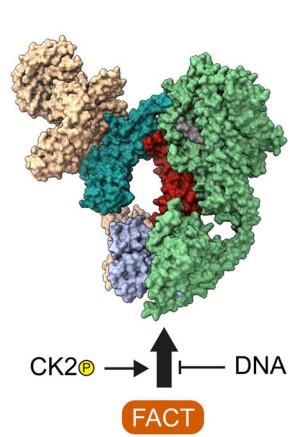

Figure 10. **Determinants of the interaction between FACT and CCAN.** FACT interacts with CENP-C, CENP-OPQUR, and CENP-TW individually. The binding is strengthened upon the assembly of the CCAN. The phosphorylation of FACT by CK2 is indispensable for the interaction, while DNA competes with FACT for CCAN binding.

AAT-3′, 5′-ATCTTCCTATAGAAACTAGACAGAATGATTCTCAGA AACTCCTTTGTGATGGAT-3′), 165-bp (5′-GTGGTAGAATAGGA AATATCTTCCTATAGAAACTAGACAGAATGATTCT-3′, 5′-CAG AAACTCCTTTGTGATGTGTGCGTTCAACTCACAGAGTTTAAC CTTTCTTTTTCATAGAGCAGTTAGGAAACACTCTGTTTGTAAT GTCTGCAAGTGGATATTCAGACGCCCTTG-3′), 183-bp (5′-AGG CCTTCGTTGGAAACGGGATTTCTTCATATTCTGCTAGA-3′, 5′-C AGAAGAATTCTCAGTAACTTCCTTGTGTTGTGTGTATTCAACTC ACAGAGTTGAACGATCCTTTACACAGAGCAGACTTGAAACAC TCTTTTTTGTGGAATTTGCAGGCCTAGATTTCAGCCGCTTTGA GGTCAATCACCCC-3′), and 199-bp (5′-ATCGCCCTTGAG-3′, 5′-GCCTTCGTTGGAAACGGGATTTCTTCATATTCTGCTAGACAG AAGAATTCTCAGTAACTTCCTTGTGTTGTGTGTATTCAACTC ACAGAGTTGAACGATCCTTTACACAGAGCAGACTTGAAACAC TCTTTTTTGTGGAATTTGCAGGCCTAGATTTCAGCCGCTTTGA GGTCAATCACCCCGTGGAT-3′) centromere 1 (CEN1)-like sequence was performed as previously described (Walstein et al., 2021; Pesenti et al., 2022).

## Reconstitution of nucleosomes

CENP-A nucleosome core particles on 145-bp 601 or CEN1-like 183-bp DNA were produced as previously reported (Guse et al., 2012).

To generate CENP-A nucleosomes on 348-bp DNA, 165-bp DNA was ligated to the front of 183-bp DNA on pre-reconstituted CENP-A nucleosome. The two species were mixed in equimolar amounts based on the concentration of the DNA fragments in a buffer consisting of 10 mM Tris, pH 7.4, 100 mM NaCl, and 1 mM EDTA. Two times the amount of [MBP]T4 DNA ligase[His] (produced in-house) relative to the DNA fragment was added with 10X T4 DNA ligase buffer, and the reaction was incubated for ~16 h at 4°C. The reaction was passed through two consecutive 1-ml HisTrap FF columns (Cytiva), equilibrated in 10 mM Tris, pH 7.4, and 100 mM NaCl to remove His-tagged T4 DNA ligase. The flow-through and wash fractions were collected, and EDTA was added to a final concentration of 2 mM.

## Protein expression and purification

CENP-C[2–545]HIKM, CENP-C[2–545], CENP-C[189–400], CENP-HIKM, CENP-LN, CENP-OPQUR, CENP-TWSX, CENP-TW, CENP-T[458-]

[C]W, CENP-SX, [MBP]CENP-T/W, CDK1/cyclin B/CKS1, PLK1, and Aurora B[45–344]/INCENP[835–903] were expressed and purified as previously reported (Klare et al., 2015; Pentakota et al., 2017; Walstein et al., 2021; Pesenti et al., 2022, 2018; Weir et al., 2016; Basilico et al., 2014; Singh et al., 2021; Huis In't Veld et al., 2022; Girdler et al., 2008). [GST]CENP-OPQUR and CENP-OPQ[68-C]U[115-C]R were purified identically to the wild type (Pesenti et al., 2018) by using either glutathione affinity or nickel affinity as a first step.

FACT, [MBP]FACT, FACT[trunc], [MBP]FACT[trunc], SPT16[Mid-AID], Mid domain, SSRP1[Mid-AID], and Mid domain, were expressed by infecting Tnao38 cells with a virus:culture ratio of 1:20 and incubating the cells at 27°C for 72 h. For the expression of [MBP]CENP-C[EGFP], a virus:culture ratio of 1:40 was used. SPT16 peptidase-like domain, SSRP1[HMG], SSRP1[513–709], [MBP]HJURP[1–80-His], and [His]CENP-A/H4/[MBP]HJURP[1–80] were expressed in *E. coli* BL21(DE3)-Codon-plus-RIL cells by growing transformed cells to an $OD_{600}$ of 0.7 in TB medium supplemented with ampicillin and chloramphenicol at 25°C. Expression was induced by adding 0.1 mM IPTG, and cells were cultured for 16 h at 18°C.

All purification steps were performed at 4°C, or samples were kept on ice. If not otherwise indicated, cells were resuspended in lysis buffer supplemented with Protease Inhibitor Mix HP Plus (Serva), 1 mM PMSF, and 10 µg/ml DNase I and lysed by sonication. The lysate was subsequently clarified by centrifugation for 45 min at 100,000 *g* at 4°C and filtration. After the final purification step, proteins of interest (POI) were concentrated, flash-frozen in liquid nitrogen, and stored at –72 or –80°C.

Cells expressing FACT, [MBP]FACT, FACT[trunc], [MBP]FACT[trunc], SPT16[Mid-AID], Mid domain, SSRP1[Mid-AID], Mid domain, and SPT16 peptidase-like domain were resuspended in a buffer containing 20 mM Tris-HCl, pH 8.0, 300 mM NaCl, 5% glycerol, and 1 mM TCEP (Buffer A). The lysate was applied to a 5-ml HisTrap FF column (Cytiva). The column was first extensively washed with Buffer A and then with Buffer A including 30 mM imidazole. Full-length FACT was eluted by a linear gradient to 400 mM imidazole, and others were eluted in Buffer A with 250 mM imidazole. The fractions containing protein were pooled and diluted 1:4 with Buffer A containing 150 mM NaCl. This was loaded on two sequential 1-ml HiTrap Q HP (Cytiva) anion exchange columns. The columns were washed, and the

protein was eluted by a gradient to 1 M NaCl. Peak fractions were analyzed in SDS-PAGE, and fractions containing protein or a stoichiometric complex were pooled and concentrated. To obtain the dephosphorylated protein, FACT and $^{MBP}$FACT were treated with λ-phosphatase (produced in-house) at 4°C in the presence of 1 mM MnCl$_2$ for ~16 h. Full-length FACT, $^{MBP}$FACT, FACT$^{trunc}$, and $^{MBP}$FACT$^{trunc}$ were finally applied to a HiLoad 16/600 Superose 6-pg column, and the others were purified on a HiLoad 16/600 Superdex 200-pg column (Cytiva).

Expressions of SSRP1$^{HMG}$ and SSRP1$^{513–709}$ were resuspended in a buffer composed of 20 mM Hepes, pH 6.8, 300 mM NaCl, 5% glycerol, 10 mM imidazole, and 1 mM TCEP. Nickel affinity purification was performed as explained above. The protein was diluted 1:4 in the same buffer with 100 mM NaCl and loaded on a 5-ml HiTrap Heparin HP column (Cytiva), and the protein was eluted in a gradient to 1 M NaCl. Fractions containing the protein were pooled and concentrated to be applied to a HiLoad 16/600 Superdex 75-pg column (Cytiva).

A pellet of $^{MBP}$CENP-C$^{EGFP}$–expressing Tnap38 cells was resuspended in ~10 vol of TALON buffer (50 mM Hepes, pH 7.0, 500 mM NaCl, 5 mM MgCl$_2$, 5% glycerol, 5 mM imidazole, 2 mM TCEP) supplemented with 2 mM PMSF and DNase I. Affinity purification was performed on a 5-ml HisTALON cartridge prepacked with TALON Superflow Resin (Cytiva), and the column was washed with 10 CV buffer. The protein was eluted in TALON buffer A with 200 mM imidazole and subsequently diluted to 300 mM NaCl in heparin buffer (20 mM Hepes, pH 7.0, 5% glycerol, 2 mM TCEP). A 5-ml HiTrap Heparin HP column (Cytiva) was equilibrated in heparin buffer including 300 mM NaCl, and the diluted protein was bound to it. The column was washed with heparin buffer with 300 mM NaCl, and the protein was eluted in a linear gradient to 1 M NaCl in 150 ml. Peak fractions containing the POI were concentrated and subjected to SEC on a HiLoad 16/600 Superose 6-pg column (Cytiva) in SEC buffer (20 mM Hepes, pH 7.0, 500 mM NaCl, 5% glycerol, 1 mM TCEP).

$^{MBP}$CENP-C$^{2–400}$ and $^{MBP}$CENP-C$^{721-C}$ were purified in a buffer consisting of 50 mM Hepes, pH 7.5, 500 mM NaCl, 10% glycerol, and 1 mM TCEP. Proteins were purified on a 5-ml HiTrap FF column (Cytiva) as indicated above and eluted in 250 mM imidazole. The eluate was diluted three times in heparin buffer (20 mM Hepes, pH 7.5, 150 mM NaCl, 5% glycerol, 1 mM TCEP), bound to a 5-ml HiTrap Heparin HP column (Cytiva), and eluted by a linear gradient to 1 M NaCl. Subsequently to SDS-PAGE, relevant fractions of $^{MBP}$CENP-C$^{2–400}$ were further purified by SEC on a HiLoad 16/600 Superdex 200-pg column, while $^{MBP}$CENP-C$^{721-C}$, which dimerizes, was applied to a HiLoad 16/600 Superose 6-pg column (Cytiva).

$^{MBP}$CENP-C$^{401–600}$, $^{MBP}$CENP-C$^{401–545}$, $^{MBP}$CENP-C$^{546–600}$, $^{MBP}$CENP-C$^{721–759}$, and $^{MBP}$CENP-C$^{760-C}$ were obtained by nickel affinity purification in 20 mM Hepes, pH 7.5, 500 mM NaCl, 10% glycerol, 10 mM imidazole, and 1 mM TCEP. Proteins were eluted in 250 mM imidazole, concentrated, and applied to a HiLoad 16/600 Superdex 200-pg column (Cytiva) in 20 mM Hepes, pH 7.5, 300 mM NaCl, 5% glycerol, and 1 mM TCEP.

$^{MBP}$CENP-C$^{601–720}$ was purified on a 5-ml HisTrap FF column (Cytiva) in 20 mM Tris, pH 8.0, 300 mM NaCl, 5% glycerol, and 1 mM TCEP and eluted in 250 mM imidazole. The eluate was diluted five times in 20 mM Tris, pH 8.0, 200 mM NaCl, 5% glycerol, and 1 mM TCEP and applied to two consecutive 1-ml HiTrap Q HP (Cytiva) columns. The POI was collected in the flow-through, while DNA was bound to the column. The flow-through was concentrated and purified on a HiLoad 16/600 Superdex 200-pg column (Cytiva).

Cells expressing $^{His}$CENP-A/H4/$^{MBP}$HJURP$^{1–80}$ were resuspended in a buffer consisting of 20 mM Tris pH 8.0, 1 M NaCl, and 1 mM TCEP. The lysate was applied to a 5-ml HisTrap FF column (Cytiva), which was first washed with buffer and subsequently with buffer including 10 mM imidazole. The protein was eluted in 250 mM imidazole and diluted 1:4 with IEX buffer (20 mM Hepes, pH 6.8, 600 mM NaCl, 1 mM TCEP) and loaded on a 1-ml HiTrap SP HP (Cytiva). The protein was eluted by a linear gradient to 2 M NaCl and afterward concentrated and purified on a HiLoad 16/600 Superdex 200-pg column (Cytiva) in the initial buffer including 1 M NaCl. To obtain untagged protein, the eluate of nickel affinity purification was treated with PreScission and TEV protease for ~16 h at 4°C while it was dialyzed to a buffer containing 750 mM NaCl. Purification on a 1-ml HiTrap SP HP and SEC on a HiLoad 16/600 Superdex 200-pg column (Cytiva) was performed as explained above.

$^{MBP}$HJURP$^{1–80}$ expressing *E. coli* were resuspended in a buffer consisting of 20 mM Hepes, pH 7.5, 300 mM NaCl, 5% glycerol, and 1 mM TCEP. The cells were lysed by high pressure in a microfluidizer and afterward clarified by centrifugation. Nickel affinity purification was performed as explained above, and the POI was eluted in a linear gradient to 400 mM imidazole. Pure fractions were pooled and concentrated and applied to a HiLoad 16/600 Superdex 200-pg column (Cytiva) in a buffer with 2.5% glycerol.

Generation of the CK2 holoenzyme was loosely based on previous literature (Raaf et al., 2008; Werner et al., 2022). The CK2α$^{1–335}$ expression plasmid carried a resistance against kanamycin, while the CK2β$^{1–193}$ expression plasmid was resistant against ampicillin. The expression was performed in *E. coli* BL21(DE3)-Codon-plus-RIL in TB medium with the specific antibiotic and additional chloramphenicol. The cells were grown to an OD$_{600}$ of 0.6 at 37°C. The expressions were induced by the addition of 0.5 mM IPTG and incubated at 30°C for 4 h. The two isolated cultures were harvested and mixed together. The cells were resuspended in lysis buffer (50 mM Tris, pH 8.5, 500 mM NaCl, 30 mM imidazole). The centrifuged lysate was incubated at 4°C for 16 h to ensure the efficient formation of the holoenzyme. The next day, the lysate was filtered and purified on a 5-ml HisTrap FF column (Cytiva). The column was washed after the application of the lysate, and protein was eluted in lysis buffer with 250 mM imidazole. The eluate was concentrated and applied to SEC on a HiLoad 16/600 Superdex 200-pg column (Cytiva) in SEC buffer (25 mM Tris, pH 8.5, 500 mM NaCl).

### Analytical SEC

Proteins were mixed in SEC buffer (20 mM Hepes, pH 6.8, 300 mM NaCl, 2.5% glycerol, 1 mM TCEP), diluted to 5 µM in 55 µl. Complexes were incubated for at least 1 h at 4°C. Samples were centrifuged, and 5 µl of sample was taken for SDS-PAGE

analysis prior to SEC on a Superose 6 Increase 5/150 GL (Cytiva) on an ÄKTA micro system (Cytiva). All samples were eluted under isocratic conditions at 4°C in SEC buffer at a flow rate of 0.2 ml/min. Fractions of 100 µl were collected and analyzed by SDS-PAGE and Coomassie blue staining.

### Pull-down assays

The proteins were mixed at 3 µM in binding buffer (20 mM Hepes, pH 6.8, 300 mM NaCl, 2.5% glycerol, 1 mM TCEP, 0.01% Tween) to a total volume of 50 µl. The samples were incubated at 4°C for at least 1 h. Afterward, they were centrifuged for 15 min at 16,000 $g$ prior to mixing them with 25 µl of amylose beads (New England Biolabs) or glutathione beads (Serva). Then, 20 µl was taken as an input sample for SDS-PAGE. The rest of the solution was incubated at 4°C for an additional hour on an orbital shaker set to 1,000 rpm (IKA VXR basic Vibrax). The samples were centrifuged at 800 $g$ for 3 min at 4°C. The unbound protein in the supernatant was removed, and the beads were washed four times with 500 µl of binding buffer. At the last step, the maximum amount of buffer was carefully removed and beads were taken up in 20 µl of SDS-PAGE sample loading buffer. The samples were boiled for 5 min at 96°C and analyzed by SDS-PAGE and Coomassie staining.

### Gel densitometry

SDS-PAGE gels were imaged in a ChemiDoc MP imaging system (Bio-Rad). A subsequent densitometric analysis of protein bands was performed using ImageLab software (Bio-Rad). The band intensity of Coomassie-stained proteins was determined to quantify binding in pull-down assays. The band intensity was normalized to the bait to account for differential loading. Differential molecular weight and staining by Coomassie were not accounted for. Therefore, different subunits cannot be compared.

To determine subunit stoichiometry, SDS-PAGE gels were supplemented with 2,2,2-trichloroethanol (TCE) and proteins were visualized by fluorescence upon UV irradiation. The band intensity was normalized to the number of tryptophan residues to establish the relative quantity of proteins (Ladner et al., 2004; Holzmüller and Kulozik, 2016).

### *In vitro* phosphorylation

Proteins were diluted to 2.5 µM in 20 mM Hepes, pH 6.8, 300 mM NaCl, 2.5% glycerol, and 1 mM TCEP. Otherwise, concentrations and buffer were used according to analytical SEC or pull-down assays. 1 mM sodium orthovanadate and 5 µM okadaic acid were added to inhibit residual lambda protein phosphatase (λ-PP) activity, and 10 mM $MgCl_2$ and 2 mM ATP were added for kinase activity. The samples were incubated at 25°C for 90 min with CK2 at a 1:20 ratio. Pro-Q Diamond phosphoprotein stain (Invitrogen) was performed according to the manufacturer's manual.

### Identification of phosphorylation sites by mass spectrometry

Liquid chromatography coupled to mass spectrometry was used to analyze phosphorylation sites on FACT after phosphorylation in vitro by CK2 as described above. We compared phosphorylated FACT with dephosphorylated FACT and untreated FACT, each expressed in insect cells. Samples were reduced, alkylated, and digested with LysC/trypsin and prepared for mass spectrometry as previously described (Rappsilber et al., 2007). The obtained peptides were subjected to a desalting cartridge in water with 0.1% formic acid for 5 min. Subsequently, they were separated on a U3000 nano-HPLC system (Thermo Fisher Scientific) using a gradient from 5% to 30% acetonitrile in 9 µl with 0.1% formic acid on a PepMap C18 nano-HPLC column (Thermo Fisher Scientific). The samples were directly introduced via a nano-electrospray source into a quadrupole–Orbitrap mass spectrometer (Q Exactive Plus; Thermo Fisher Scientific). The Q Exactive was operated in a data-dependent mode acquiring one survey scan followed by up to 10 MS/MS scans. To identify phospho-sites, the resulting raw files were processed with MaxQuant (version 2.2.0.0), searching against the sequences of SPT16 and SSRP1 and a contamination database including N-terminal acetylation, oxidation (M), and phosphorylation (STY) as variable modifications and carbamidomethylation (C) as fixed modification. A false discovery rate cutoff of 1% was applied at the peptide and protein levels and on the site decoy fraction (Cox and Mann, 2008).

### In silico prediction of phosphorylation sites

NetPhos 3.1 with a score higher than 0.5 and Scansite 4.0 at Medium setting searches were performed to predict CK2 phosphorylation sites on SPT16 and SSRP1 (Blom et al., 1999, 2004; Obenauer et al., 2003). Only sites that were predicted by both algorithms are displayed in Fig. 5 D.

### Cell culture

All cells were grown at 37°C in the presence of 5% $CO_2$. Parental Flp-In T-Rex DLD-1 osTIR1 and DLD-1[YFP-AID] cells were a kind gift from D. C. Cleveland (University of California, San Diego, CA, USA). DLD-1 cells were grown in Dulbecco's modified Eagle's medium (DMEM; PAN Biotech), and parental Flp-In T-Rex hTERT RPE-1 cells were a kind gift from J. Pines (Institute of Cancer Research: London, UK) and hTERT RPE-1 cells expressing endogenously tagged CENP-U-FKBP[F36V] were grown in DMEM/F12 media (PAN Biotech); all media were supplemented with 10% tetracycline-free fetal bovine serum (Sigma-Aldrich) and L-glutamine (PAN Biotech). K562-SSRP1-dtag cells were a kind gift from P. Cramer, M. Oudelaar, and K. Žumer (Max Planck Institute for Multidisciplinary Sciences, Göttingen, Germany) and were grown in RPMI (Gibco) media supplemented with 1× GlutaMAX (Gibco).

### Cell synchronization and drug treatments

Degradation of the endogenous CENP-C[YFP-AID] was achieved through the addition of 500 µM IAA (Sigma-Aldrich), and degradation of endogenous CENP-U-FKBP[F36V] was achieved by the addition of 500 nM dTAG[V]-1 (Tocris). Cells were synchronized using STLC at 5 µM for 16 h or nocodazole 3.3 µM for 4 h. Mitotic degradation of FACT was achieved by treating STLC-arrested K562 cells with 500 nM dTAG[V]-1 (Tocris).

## RNA interference

Depletion of endogenous proteins was achieved through transfection of siRNA with RNAiMAX (Invitrogen) according to the manufacturer's instructions. The following siRNA treatments were performed in this study: 30 nM of each oligo, siCENP-T (Dharmacon, 5′-GACGAUAGCCAGAGGGCGU-3′, 5′-AAGUAG AGCCCUUACACGA-3′) for 60 h (Basilico et al., 2014), siCENP-H (Sigma-Aldrich, 5′-CUAGUGUGCUCAUGGAUAA-3′) (Weir et al., 2016), siCENP-I (Sigma-Aldrich, 5′-AAGCAACTCGAAGAACAT CTC-3′) (Liu et al., 2003), siCENP-K (Dharmacon, On-TARGETplus SMARTpool-XX) (Okada et al., 2006), siCENP-M (Sigma-Aldrich, 5′-ACAAAAGGUCUGUGGCUAA-3′, 5′-UUA AGCAGCUGGCGUGUUA-3′, 5′-GUGCUGACUCCAUAAACAU-3′) (Basilico et al., 2014) for 72 h.

## Transient transfection of Flp-In T-Rex hTERT RPE-1

1 µg of pcDNA5-EGFP-SSRP1-IRES was transiently transfected in Flp-In T-Rex hTERT RPE-1 cells using FuGENE HD Transfection Reagent (Promega) following the standard manufacturer's protocol. Expression was induced with 200 ng/ml of doxycycline 6 h after transfection. Cells were arrested with RO3306 for 16 h and released for 1 h to enrich mitotic population for immunofluorescence analysis.

## Generation of stable cell lines

Stable Flp-In T-Rex DLD-1 osTIR1 cell lines were generated using FRT/Flp recombination. $^{EGFP}$CENP-U$^{fl}$ and $^{EGFP}$CENP-U$^{I15-C}$ constructs were cloned into pcDNA5 plasmids (Singh et al., 2021) and were cotransfected with pOG44 (Invitrogen), encoding the Flp recombinase, into DLD-1 cells using X-tremeGENE (Roche) according to the manufacturer's instructions. After selection for 2 wk in DMEM supplemented with hygromycin B (250 µg/ml; Carl Roth) and blasticidin (4 µg/ml; Thermo Fisher Scientific), single-cell colonies were isolated and expanded and the expression of the transgenes was checked by immunofluorescence microscopy and immunoblotting analysis. The gene expression was induced by the addition of 0.3 µg/ml doxycycline (Sigma-Aldrich).

An hTERT RPE-1 CENP-U-FKBP$^{F36V}$-NeoR knock-in cell line was generated via electroporation of gRNA-Cas9 ribonucleoproteins as previously described (Ghetti et al., 2021). Briefly, $3 \times 10^5$ cells were electroporated using P3 Primary Cell Nucleofector 4D Kit and Nucleofector 4D system (Lonza) with 400 ng donor DNA, 120 pmol of Cas9, 1.5 µl Alt-R-CRISPR-Cas9 crRNA (5′-TTAGAGAAG CTCCTTGACCA [GGG]-3′), 1.5 µl Alt-R CRISPR-Cas9 tracrRNA (100 µM, IDT), and 1.2 µl of Alt-R Cas9 Electroporation Enhancer (100 µM, IDT). After electroporation, cells were seeded into DMEM/F12 media supplemented with 1 µM NU7441 for 48 h before selection with 400 µg/ml of G418 for 2 wk. The pool of cells was subjected to single-cell dilution to obtain monoclonal lines. Genomic DNA was isolated from clones, and in-frame knock-in was confirmed by Sanger sequencing using primers spanning the locus of insertion (Primer fwd: 5′-CATGTGTGTGGTAGTCACAGCATG-3′, Primer rev: 5′-TCTGGGATAATGGCATTGATGATGC-3′).

## Immunofluorescence

Cells were grown on coverslips pre-coated with poly-L-lysine (Sigma-Aldrich). Cells were pre-permeabilized with 0.5% Triton X-100 solution in PHEM (Pipes, Hepes, EGTA, MgCl$_2$) buffer supplemented with 100 nM microcystin for 5 min before fixation with 4% paraformaldehyde in PHEM for 15 min. After blocking with 5% boiled goat serum (BGS) in PHEM buffer for 30 min, cells were incubated for 2 h at room temperature with the following primary antibodies: CENP-C (guinea pig, #PD030; MBL, 1:1,000), CENP-HK, CENP-TW (rabbit, made in-house, 1:800), SSRP1 (mouse, #609702; BioLegend Europe, 1:200), CENP-A (mouse, GTX13939; GeneTex, 1:500), CENP-O (gift from McAinsh Lab, 1:200), PLK1 (mouse, #ab17057; Abcam, 1:500), CENP-R (rabbit, #107431-AP; Proteintech, 1:200), and CREST (anticentromere anti-immune serum) (human, #15-234; Antibodies Inc. [via antibodies-online], 1:200) diluted in 2.5% BGS-PHEM with an exception of CENP-U (Sigma-Aldrich [HPA022048; Atlas, 1:100]), which was diluted in 5% BGS-PHEM and incubated at 37°C for 3 h.

Subsequently, cells were incubated for 1 h at room temperature with the following secondary antibodies: (all 1:200 in 2.5% BGS-PHEM): goat anti-mouse Alexa Fluor 488 (A11001; Invitrogen A), goat anti-mouse Rhodamine Red (115-295-003; Jackson ImmunoResearch), donkey anti-rabbit Alexa Fluor 488 (A21206; Invitrogen), donkey anti-rabbit Rhodamine Red (711-295-152; Jackson ImmunoResearch), goat anti-human Alexa Fluor 647 (109-603-003; Jackson ImmunoResearch), goat anti-guinea pig Alexa Fluor 647 (A-21450; Invitrogen). All washing steps were performed with PHEM supplemented with 0.1% Triton X-100 buffer. DNA was stained with 0.5 µg/ml DAPI (Serva), and Mowiol (Calbiochem) was used as mounting media.

Cells were imaged at room temperature using a spinning disk confocal device on the 3i Marianas system equipped with an Axio Observer Z1 microscope (Zeiss), a CSU-X1 confocal scanner unit (Yokogawa Electric Corporation), 100×/1.4NA oil objectives (Zeiss), and Orca Flash 4.0 V2 sCMOS Camera (Hamamatsu) and Orca Fusion BT sCMOS Camera (Hamamatsu). Images were acquired as z sections at 0.27 µm using Slidebook software 2023.3 and 2024.2 (Intelligent Imaging Innovations). Images were converted into maximum intensity projections and exported as 16-bit TIFF files. Alternatively, cells were imaged using a UPLSAPO 100×/1.4NA oil objective on a DeltaVision deconvolution microscope (GE Healthcare) equipped with an IX71 inverted microscope (Olympus), and a pco.edge sCMOS camera (PCO-TECH Inc.). Images were acquired as z sections at 0.2 µm.

Quantification of kinetochore signals was performed on 16-bit maximum intensity projections using a semi-automatic quantification MACRO in FIJI (Schindelin et al., 2012) with background subtraction. When CREST was used as kinetochore reference, Otsu's thresholding of the DAPI signal was applied for generating a KTS segmentation mask per cell. Integrated intensities per cell were calculated for each fluorescence channel based on the DAPI reference mask. Background-corrected mean fluorescence intensities were multiplied by the ROI area to obtain the total fluorescence signal for POI and reference in each cell. The total fluorescence signal for each channel of interests was plotted using GraphPad Prism 9. When CENP-C was used as a kinetochore reference, the mask was created using CENP-C for individual kinetochore picking. Data were exported to Microsoft Excel for normalization and plotted using GraphPad Prism 9

software. Statistical analysis was performed with a nonparametric $t$ test comparing two unpaired groups (Mann–Whitney test). Symbols indicate $^{n.s.}P > 0.05$, $*P \leq 0.05$, $**P \leq 0.01$, $***P \leq 0.001$, $****P \leq 0.0001$. Images were assembled in Adobe Illustrator 2024.

### Co-immunoprecipitation

For co-immunoprecipitation, DLD-1 cells expressing EGFP, [EGFP]CENP-U[fl], and [EGFP]CENP-U[115-C] were harvested by mitotic shake off and lysed using lysis buffer (75 mM Hepes, pH 7.5, 150 mM KCl, 10% glycerol, 1 mM EGTA, 1.5 mM MgCl$_2$, 1 mM DTT, 0.075% NP-40) supplemented with 1 mM PMSF, Protease Inhibitor Mix HP Plus (Serva), phosphatase inhibitor PhosSTOP (Sigma-Aldrich), and Benzonase (EMD Millipore Corp). Lysates were clarified by centrifugation at 22,000 $g$ for 30 min at 4°C, and the supernatant was collected for immunoprecipitation analysis. Lysates at a concentration of 7 mg/ml were incubated with 20 µl GFP-Trap magnetic agarose (ChromoTek) for 3 h at 4°C in a total volume of 500 µl in lysis buffer. Beads were washed three times with lysis buffer. The dry beads were resuspended in SDS-PAGE sample loading buffer and boiled for 5 min at 95°C. The samples were analyzed by SDS-PAGE and subsequent western blotting analysis. The following antibodies were used: GFP (rabbit, made in-house, 1:3,000), SSRP1 (mouse; BioLegend, 1:1,000), PLK1 (1:1,000), and anti-mouse or anti-rabbit (1:10,000; NXA931 and NA934; Amersham) conjugated to horseradish peroxidase were used. After incubation with ECL western blotting reagent (GE Healthcare), images were acquired with the ChemiDoc MP system (Bio-Rad) using ImageLab 6.0.1 software.

### Online supplemental material

Fig. S1 supports Fig. 1 and includes the previously established reconstitution of the CCAN, additional SEC runs suggesting a 1:1 of FACT and CCAN, and additional SEC runs and summarizing table related to the pull-down in Fig. 1 E. Fig. S2 supports Fig. 1 and Fig. 2 and presents the localization of GFP-SSRP1 and additional IF experiments after a 4-h depletion of CENP-C. Fig. S3 supports Fig. 5, which contains additional information on the CENP-HIKM and CENP-T RNAi experiments. Fig. S4 supports Fig. 5 and presents additional controls for the rapid depletion of CENP-U-FKBP[F36V]. Fig. S5 supports Fig. 8 with additional SEC experiments, a pull-down, and phospho-stainings related to experiments in Fig. 8. Fig. S6 supports Fig. 9 and displays various SEC experiments and pull-downs. Table S1 summarizes the phospho-sites on SPT16 and SSRP1 that were phosphorylated by CK2 *in vitro* and could be detected by mass spectrometry.

### Data availability

All vectors, reagents, and data described in this manuscript are available from Andrea Musacchio upon reasonable request.

## Acknowledgments

We thank Franziska Müller and Petra Janning for help with mass spectrometry experiments, Dongqing Pan for the generation of initial constructs for the expression of recombinant FACT, Karsten Niefind for providing expression constructs for CK2, Duccio Conti for the help in initial experiments, Sabine Wohlgemuth for the purification of CENP-LN, Carolin Körner for the preparation of recombinant kinases, Lia Nitz for the production of a subset of [MBP]CENP-C fusion proteins, Nico Schmidt for help with microscopy experiments and data analysis, and Patrick Cramer, Kristina Žumer, A. Marieke Oudelaar, Daniele Fachinetti, Don C. Cleveland, and Jonathon Pines for sharing cell lines.

A. Musacchio acknowledges funding from the Max Planck Society, the European Research Council Synergy Grant 951430 (BIOMECANET), the DFG's Collaborative Research Centre 1430 "Molecular Mechanisms of Cell State Transitions," and the CANTAR network under the Netzwerke-NRW program. Open access funding provided by the Max Planck Society.

Author contributions: J. Schweighofer: conceptualization, formal analysis, investigation, resources, validation, visualization, and writing—original draft, review, and editing. B. Mulay: conceptualization, formal analysis, investigation, resources, validation, visualization, and writing—review and editing. I. Hoffmann: investigation and writing—review and editing. D. Vogt: resources. M.E. Pesenti: investigation and writing—review and editing. A. Musacchio: conceptualization, funding acquisition, project administration, supervision, validation, visualization, and writing—original draft, review, and editing.

Disclosures: The authors declare no competing interests exist.

Submitted: 5 December 2024

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

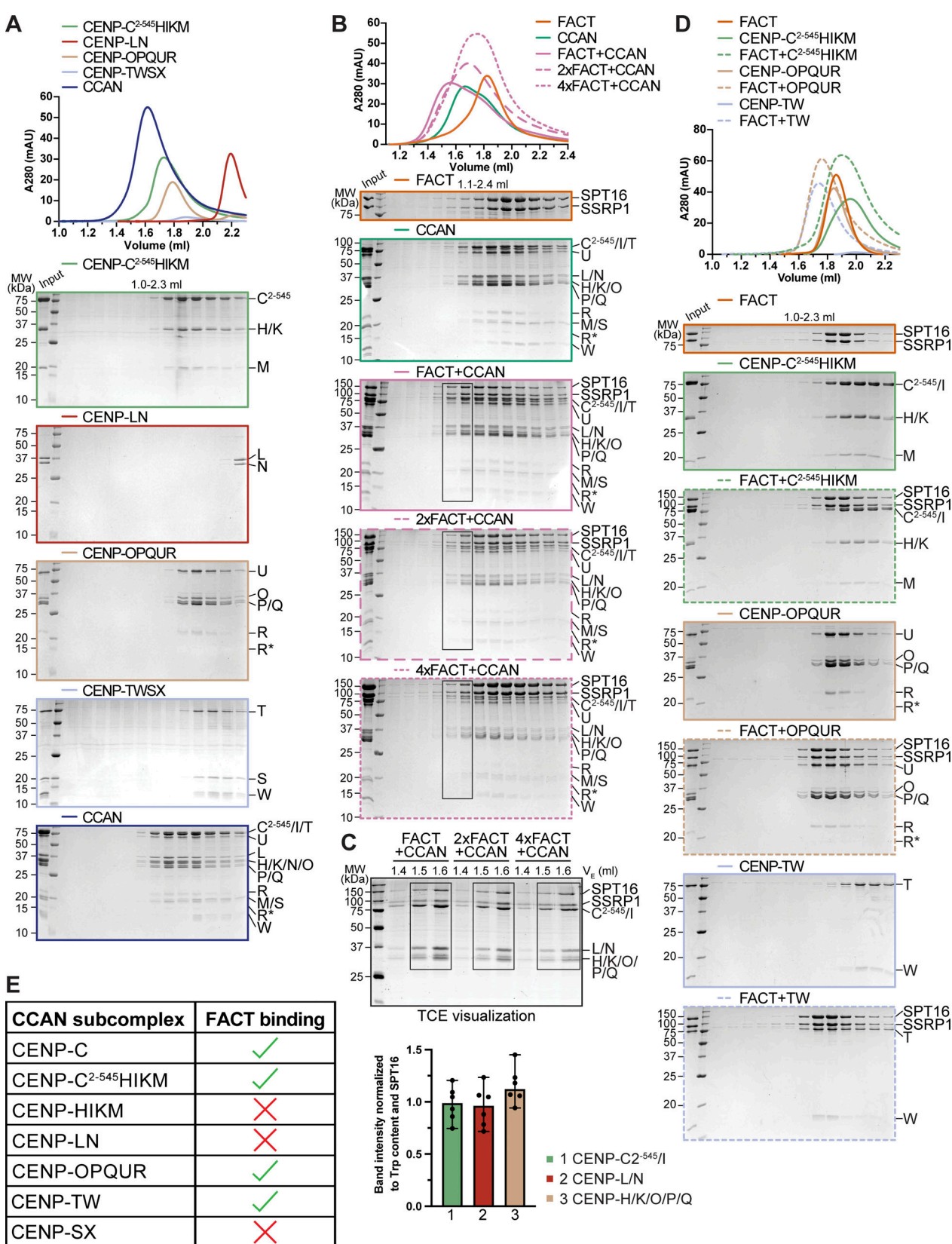

Figure S1. **(Related to Fig. 1). (A)** Analytical SEC of the individual CCAN subcomplexes and its reconstitution. **(B)** Analytical SEC of the CCAN with different amounts of FACT. **(C)** Fractions highlighted with boxes in B were run on an SDS-PAGE containing TCE. The visualized tryptophan was quantified as the band intensity and normalized by the number of tryptophan residues and SPT16 to estimate the relative amount of the protein. Bars represent the median and range. **(D)** Analytical SEC of FACT and CENP-C²⁻⁵⁴⁵HIKM, CENP-OPQUR, and CENP-TW with Coomassie-stained SDS-PAGE gels below. **(E)** Interaction between FACT and the CCAN is summarized in a table. Source data are available for this figure: SourceData FS1.

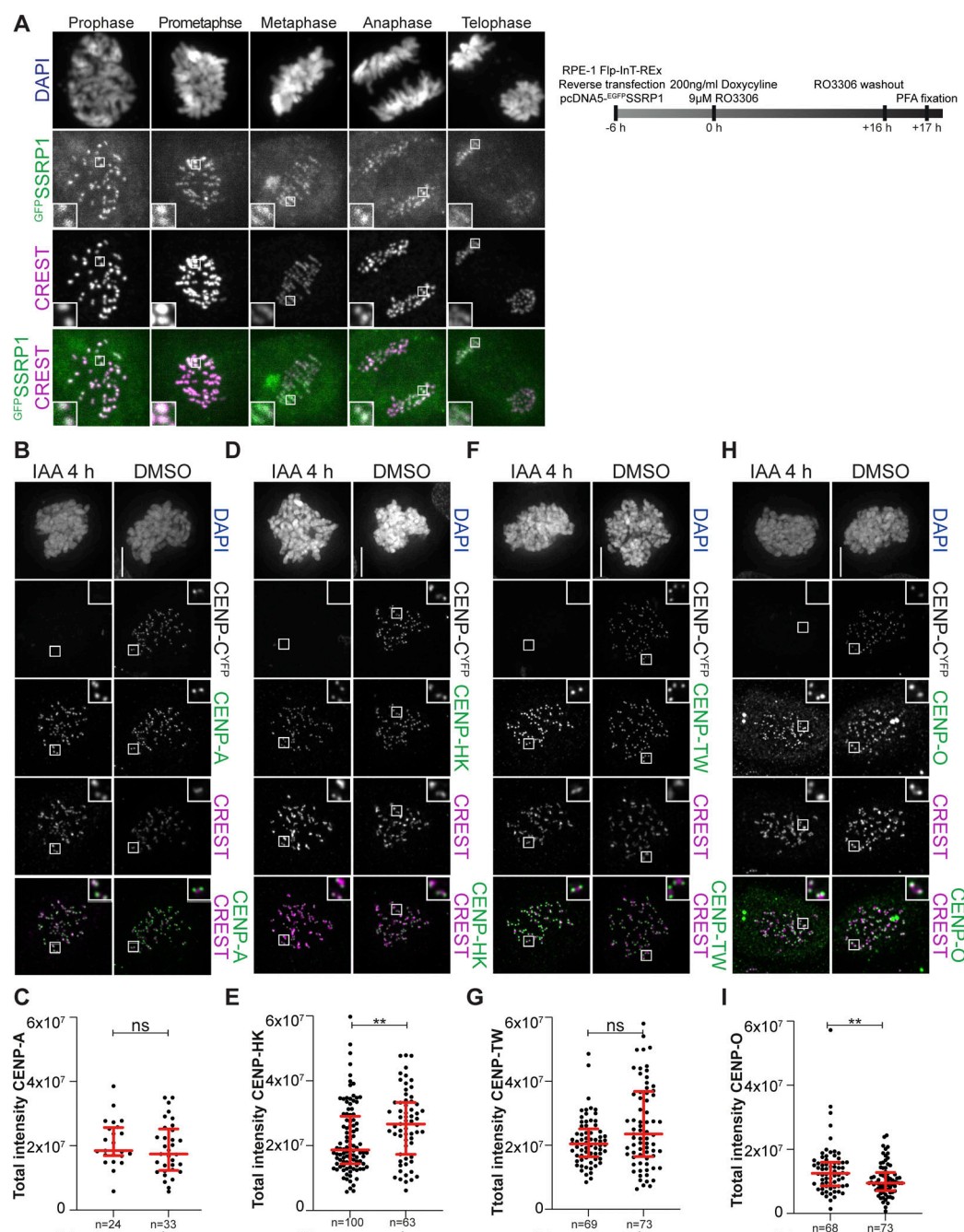

Figure S2. **(Related to** Fig. 1 **and** Fig. 2**). (A)** Representative images of localization of GFPSSRP1 after transient transfection in RPE-1 cells. CREST serum was used to visualize kinetochores, and DAPI to stain the DNA. Scale bar: 5 µm. **(B)** Representative images of localization of CENP-A after degradation of CENP-C in DLD-1-CENP-CYFP-AID cells for 4 h. Cells were treated with IAA (500 µM) to degrade endogenous CENP-C and nocodazole (3.3 µM) to get mitotic population of cells for 4 h. CREST serum was used to visualize kinetochores, and DAPI to stain DNA. Three biological replicates were performed. Scale bar: 5 µm. **(C)** Scatter plot of CENP-A levels at kinetochores for the experiment shown in B. n is the number of cells. **(D)** Representative images of localization of CENP-HK after degradation of CENP-C in DLD-1-CENP-CYFP-AID cells for 4 h. Cells were treated with IAA (500 µM) to degrade endogenous CENP-C and nocodazole (3.3 µM) to get mitotic population of cells for 4 h. CREST serum was used to visualize kinetochores, and DAPI to stain DNA. Three biological replicates were performed. Scale bar: 5 µm. **(E)** Scatter plot of CENP-HK levels at kinetochores for the experiment shown in D. n refers to the number of cells. **(F)** Representative images of localization of CENP-TW after degradation of CENP-C in DLD-1-CENP-CYFP-AID cells for 4 h. Cells were treated with IAA (500 µM) to degrade endogenous CENP-C and with nocodazole (3.3 µM) for 4 h to enrich for mitotic cells. CREST serum was used to visualize kinetochores, and DAPI to stain DNA. Three biological replicates were performed. Scale bar: 5 µm. **(G)** Scatter plot of CENP-TW levels at kinetochores for the experiment shown in F. n refers to the number of cells. **(H)** Representative images of localization of CENP-O after degradation of CENP-C in DLD-1-CENP-CYFP-AID cells for 4 h. Cells were treated with IAA (500 µM) to degrade endogenous CENP-C and nocodazole (3.3 µM) to get mitotic population of cells for 4 h. CREST was used to visualize kinetochores, and DAPI to stain DNA. Three biological replicates were performed. Scale bar: 5 µm. **(I)** Scatter plot of CENP-O levels at kinetochores for the experiment in H. n is the number of cells. Statistical analysis was performed with a nonparametric t test comparing two unpaired groups (Mann–Whitney test). Symbols indicate n.s.P > 0.05, *P ≤ 0.05, **P ≤ 0.01, ****P ≤ 0.0001. Red bars represent the median and interquartile range.

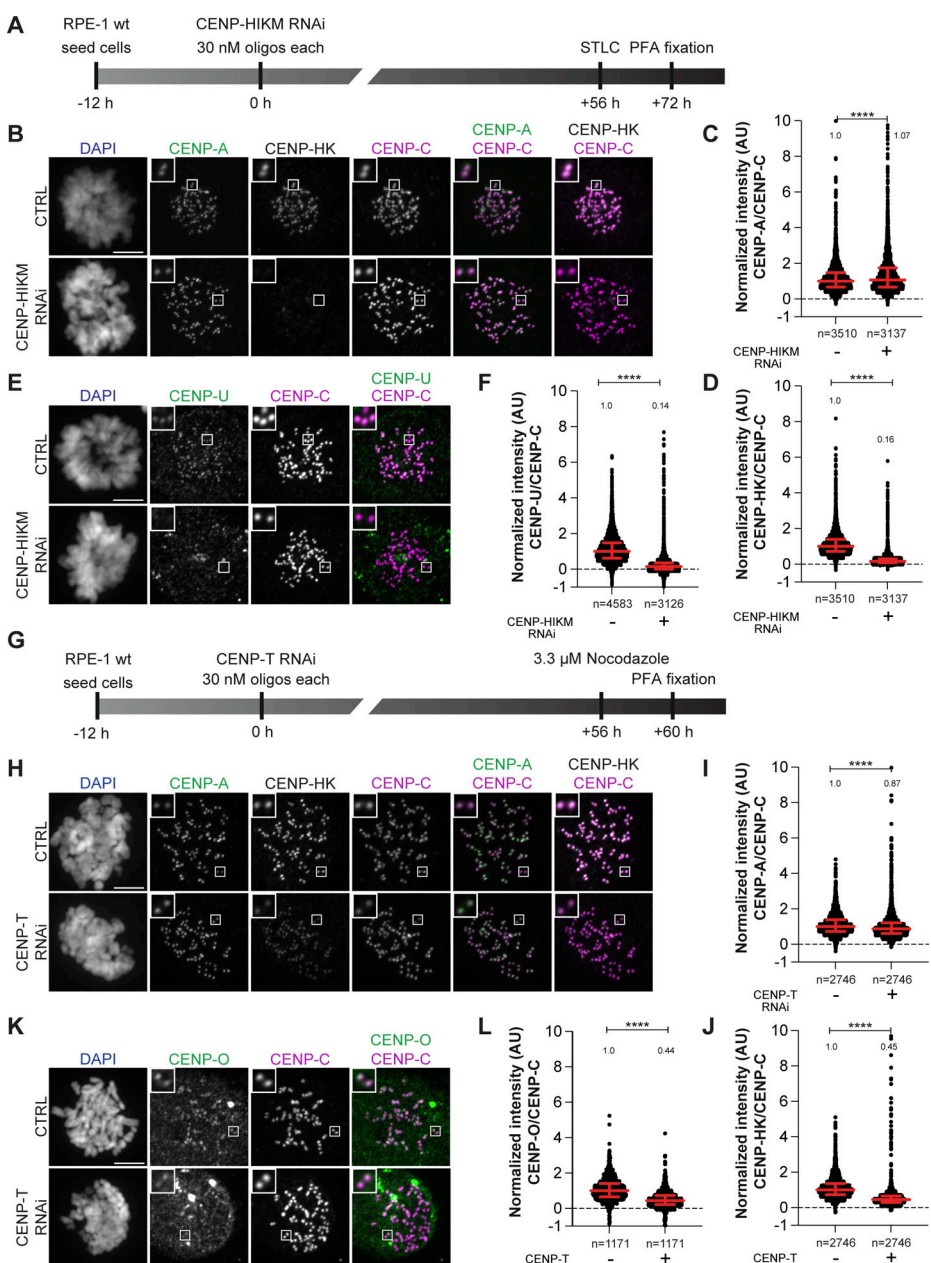

Figure S3. **(Related to** Fig. 5**). (A)** Schematic representation of experimental scheme used for CENP-HIKM RNAi. **(B)** Representative images of localization of CENP-A and CENP-HK after depletion of the CENP-HIKM complex in RPE-1 cells. CENP-HIKM RNAi was performed for 72 h using oligos for each subunit at 30 nM concentration. Cells were treated with STLC (5 µM) for 16 h to obtain a mitotic population before fixation. CENP-C was used to visualize kinetochores, and DAPI to stain DNA. Three biological replicates were performed. Scale bar: 5 µm. **(C)** Scatter plot of CENP-A levels at kinetochores for the experiment shown in B. *n* is the number of individually measured kinetochores. **(D)** Scatter plot of CENP-HK levels at kinetochores for the experiment in B. *n* is the number of individually measured kinetochores. **(E)** Representative images of localization of CENP-U after depletion of the CENP-HIKM complex in RPE-1 cells. CENP-HIKM RNAi was performed for 72 h using oligos for each subunit at 30 nM concentration. To obtain mitotic cells, cells were treated with STLC (5 µM) for 16 h before fixation. CENP-C visualizes kinetochores, and DAPI stains DNA. Three biological replicates were performed. Scale bar: 5 µm. **(F)** Scatter plot of CENP-U levels at kinetochores of the experiment shown in E. *n* refers to individually measured kinetochores. **(G)** Schematic representation of the experimental scheme used for CENP-T RNAi. **(H)** Representative images of localization of CENP-A and CENP-HK after depletion of the CENP-T complex in RPE-1 cells. CENP-T RNAi was performed for 60 h using oligos for each subunit at 30 nM concentration. Cells were treated with nocodazole (3.3 µM) for 4 h before fixation to enrich for mitotic cells. CENP-C visualizes kinetochores, and DAPI stains DNA. Three biological replicates were performed. Scale bar: 5 µm. **(I)** Scatter plot of CENP-A levels at kinetochores for the experiment in H. *n* is the number of individually measured kinetochores. **(J)** Scatter plot of CENP-HK levels at kinetochores for the experiment in H. *n* is the number of individually measured kinetochores. **(K)** Representative images of localization of CENP-O after depletion of the CENP-T complex in RPE-1 cells. CENP-T RNAi was performed for 60 h using oligos for each subunit at 30 nM concentration. Cells were treated with nocodazole (3.3 µM) for 4 h before fixation to obtain mitotic population. CENP-C visualizes kinetochores, and DAPI stains DNA. Three biological replicates were performed. Scale bar: 5 µm. **(L)** Scatter plot of CENP-O levels at kinetochores for the experiment in K. *n* is the number of individually measured kinetochores. Statistical analysis was performed with a nonparametric *t* test comparing two unpaired groups (Mann–Whitney test). Symbols indicate [n.s.]$P > 0.05$, *$P \leq 0.05$, **$P \leq 0.01$, ****$P \leq 0.0001$. Red bars represent the median and interquartile range.

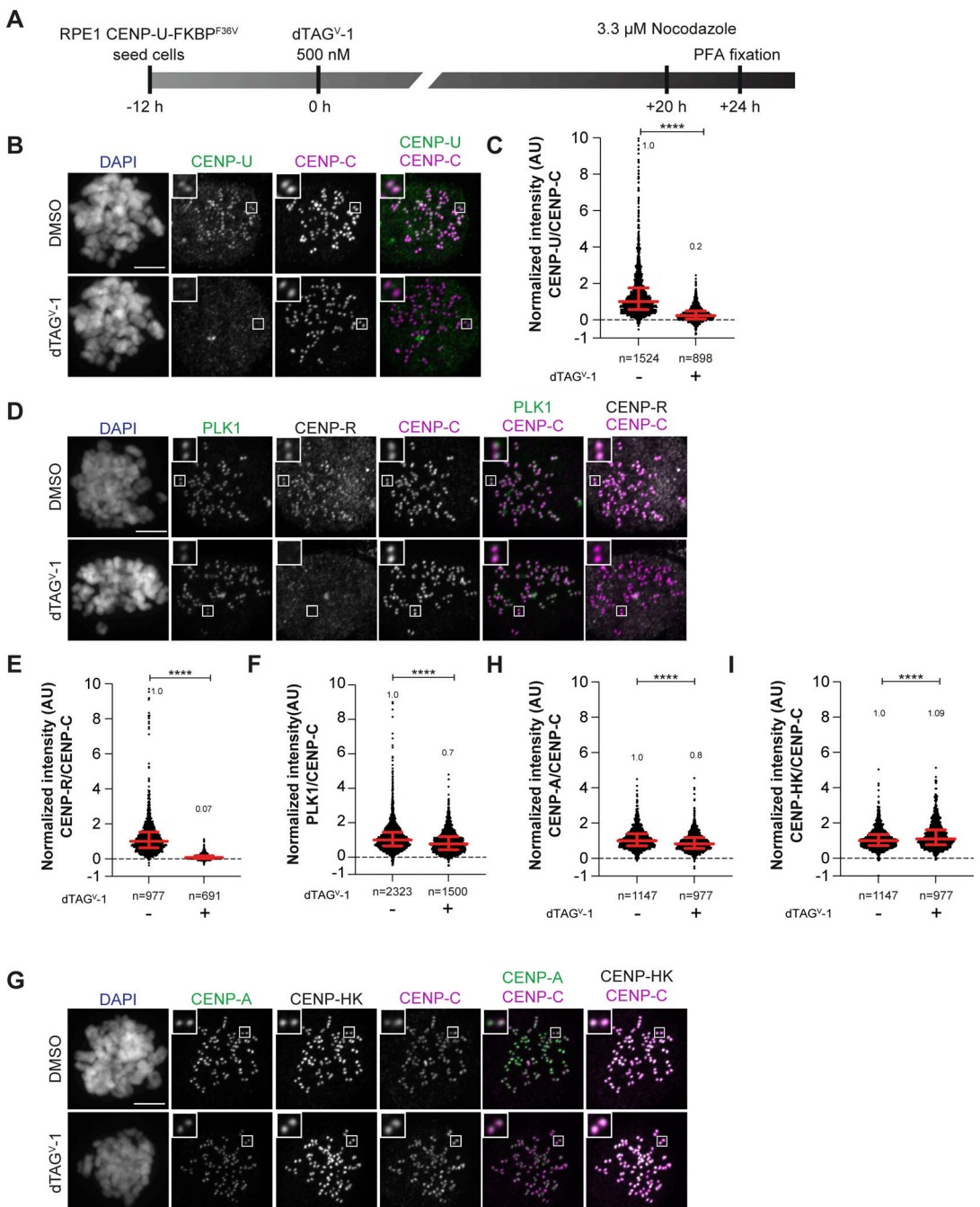

Figure S4. **(Related to Fig. 5). (A)** Schematic representation of the experimental scheme used for CENP-U dTAG^V-1 treatment. **(B)** Representative images of localization of CENP-U after depletion of the CENP-U complex in RPE-1-CENP-U-FKBP^F36V cells. Cells were treated with dTAG^V-1 (500 nM) for 24 h to degrade the endogenous CENP-U complex. Cells were treated with nocodazole (3.3 μM) for 4 h to obtain a mitotic cell population prior to fixation. CENP-C was used to visualize kinetochores, and DAPI to stain DNA. Three biological replicates were performed. Scale bar: 5 μm. **(C)** Scatter plot of CENP-U levels at kinetochores for the experiment in B. *n* is the number of individually measured kinetochores. **(D)** Representative images of localization of CENP-R and PLK1 after depletion of the CENP-U complex in RPE-1-CENP-U-FKBP^F36V cells. Cells were treated with dTAG^V-1 (500 nM) for 24 h to degrade the endogenous CENP-U complex. Cells were treated with nocodazole (3.3 μM) for 4 h prior to fixation to obtain mitotic population. CENP-C was used to visualize kinetochores, and DAPI to stain DNA. Three biological replicates were performed. Scale bar: 5 μm. **(E)** Scatter plot of CENP-R levels at kinetochores for the experiment in D. *n* is the number of individually measured kinetochores. **(F)** Scatter plot of PLK1 levels at kinetochores for the experiment in D. *n* is the number of individually measured kinetochores. **(G)** Representative images of localization of CENP-A and CENP-HK after depletion of the CENP-U complex in RPE-1-CENP-U-FKBP^F36V cells. Cells were treated with dTAG^V-1 (500 nM) for 24 h to degrade the endogenous CENP-U complex. Cells were treated with nocodazole (3.3 μM) for 4 h prior to fixation to obtain mitotic population. CENP-C was used to visualize kinetochores, and DAPI to stain DNA. Three biological replicates were performed. Scale bar: 5 μm. **(H)** Scatter plot of CENP-A levels at kinetochores of the experiment shown in G. *n* is the number of individually measured kinetochores. **(I)** Scatter plot of CENP-HK levels at kinetochores of the experiment shown in G. *n* is the number of individually measured kinetochores. Statistical analysis was performed with a nonparametric *t* test comparing two unpaired groups (Mann–Whitney test). Symbols indicate ^n.s.P > 0.05, *P ≤ 0.05, **P ≤ 0.01, ****P ≤ 0.0001. Red bars represent the median and interquartile range.

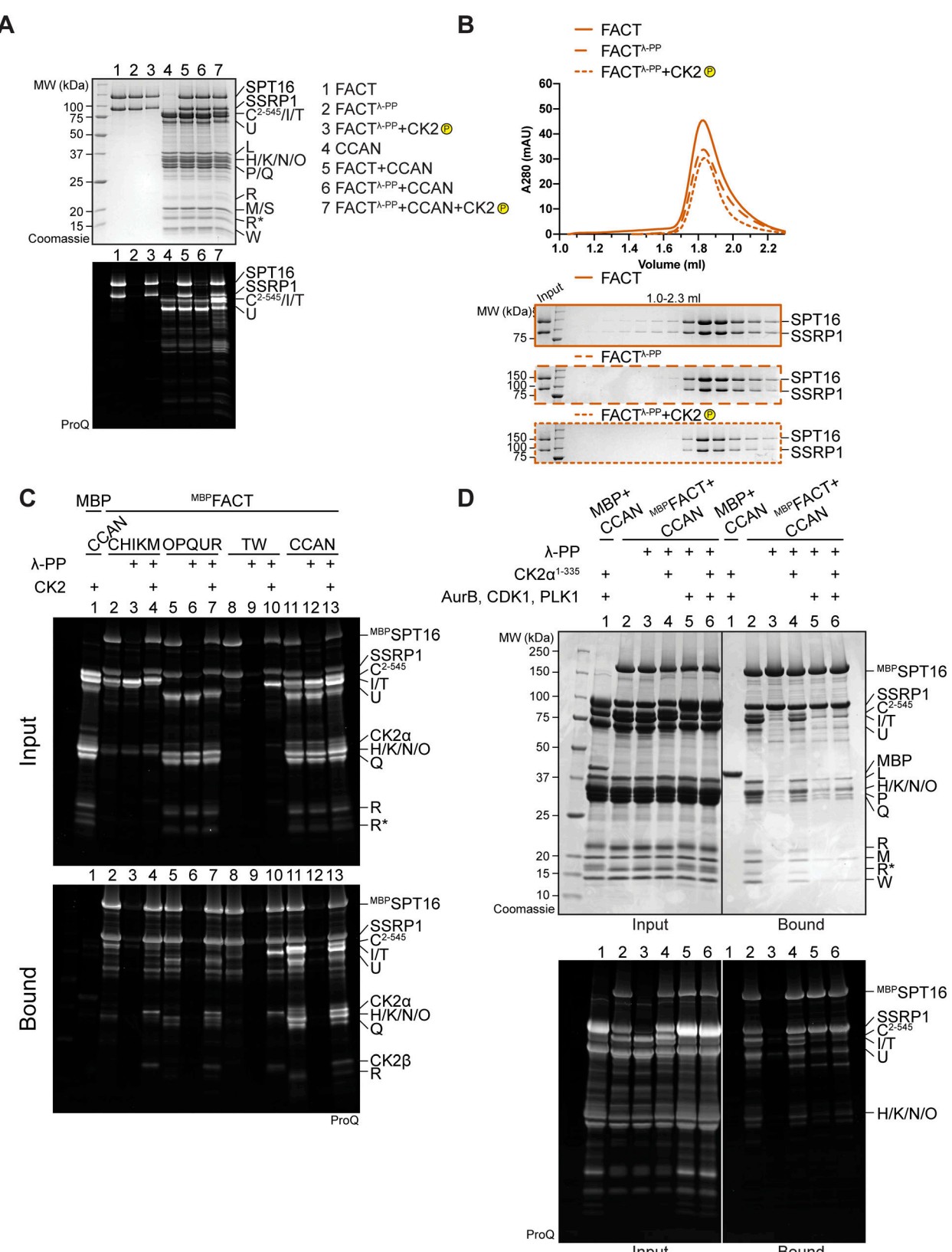

Figure S5. **(Related to** Fig. 8**). (A)** Pro-Q Diamond staining to monitor the phosphorylation state of samples in Fig. 8 A. **(B)** Additional samples of the analytical SEC experiment in Fig. 8 A verify that the phosphorylation state of FACT does not alter its elution volume. **(C)** Pro-Q Diamond staining of the pull-down assay in Fig. 8 B. **(D)** Amylose-resin pull-down assay to analyze CCAN binding to dephosphorylated ᴹᴮᴾFACT upon phosphorylation by CK2α$^{1-335}$ or Aurora B, CDK1, and PLK1, or all of them. CDK1 indicates the use of a complex of CDK1/cyclin B/CKS1. The Pro-Q Diamond staining of the SDS-PAGE is shown below. Source data are available for this figure: SourceData FS5.

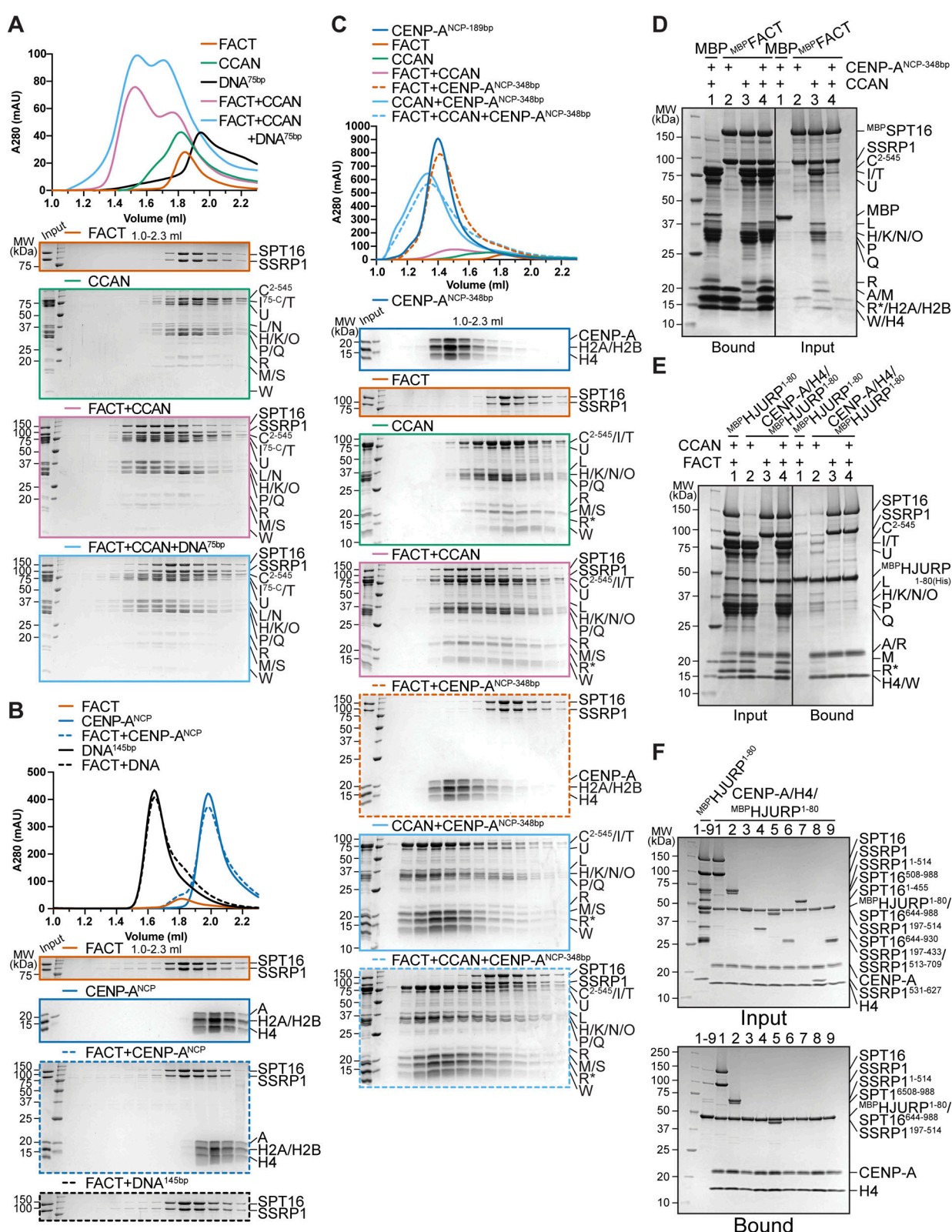

Figure S6. **(Related to** Fig. 9**). (A)** Analytical SEC of FACT and CCAN upon the addition of a 75-bp CEN1-like DNA. **(B)** Analytical SEC to test binding of FACT to CENP-A[NCP] or 145-bp Widom 601 DNA. **(C)** Analytical SEC of FACT, CCAN, and CENP-A[NCP] on a 348-bp DNA. The histone octamer was reconstituted on a 183-bp CEN1-like sequence, and 165-bp CEN1-like sequence was ligated to it. **(D)** Amylose-resin pull-down assay of [MBP]FACT and CCAN upon the addition of CENP-A[NCP-348bp]. **(E)** Amylose-resin pull-down assay using [MBP]HJURP[1–80] in complex with CENP-A/H4 as a bait and CCAN and FACT as preys. [MBP]HJURP[1–80] in the absence of histones was used as a negative control. **(F)** Amylose-resin pull-down assay using [MBP]HJURP[1–80] in complex with CENP-A/H4 as a bait and FACT constructs (Fig. 7 A) as preys. [MBP]HJURP[1–80] in the absence of histones was used as a negative control. Source data are available for this figure: SourceData FS6.

**Provided online is Table S1. Table S1 summarizes the phospho-sites on SPT16 and SSRP1 that were phosphorylated by CK2 in vitro and could be detected by mass spectrometry.**

