## [Peer Review File · The Journal of Cell Biology]

Interactions with multiple inner kinetochore proteins determine mitotic localization of FACT

Julia Schweighofer, Bhagyashree Mulay, Ingrid Hoffmann, Doro Vogt, Marion Pesenti, and Andrea Musacchio

Corresponding Author(s): Andrea Musacchio, Max Planck Society

Review Timeline:

Submission Date:	2024-12-05
Editorial Decision:	2025-01-24
Revision Received:	2025-02-04

Monitoring Editor: William Earnshaw

Scientific Editor: Tim Fessenden

Transaction Report:

DOI: <https://doi.org/10.1083/jcb.202412042>

Revision 0

Review #1

1. Evidence, reproducibility and clarity:

Evidence, reproducibility and clarity (Required)

****Summary:****

The authors investigated molecular interactions between CCAN and FACT complexes. They revealed contact domains in FACT and the cognate subcomplexes of CCAN by in vitro reconstitution from recombinant proteins followed by SEC and pull-down assay.

They also revealed a couple of potential means to control interactions between FACT and the CCAN. They conclude that phosphorylation of FACT by CK2 is essential for binding to the CCAN; and CENP-A nucleosomes or DNA prevent CCAN from interacting with FACT.

****Major comments:****

The authors show that phosphorylation of FACT is essential for interaction with CCAN. They argue that this phosphorylation is partly catalysed by CK2.

My concerns are:

1. The authors assume that the sites phosphorylated in insect cell are also phosphorylated in human cells. However, it is not demonstrated which residues are phosphorylated in human cells and whether they match those from insect cells. Whether phosphorylation of recombinant proteins in insect cells is physiologically relevant to mammalian is uncertain. Kinetochores components are not very well conserved evolutionarily, thus their regulation may be different.
2. They identify several residues which are phosphorylated by CK2 in vitro. However, these are not necessarily the same sites as those phosphorylated in insect cells or more importantly in human cells. The in vitro phosphorylation by CK2 did not restore binding affinity in full, suggesting phosphorylation at other sites may be critical for interaction with CCAN. Further evidence is required to support the claim that those sites are phosphorylated in vivo and important for integrity of kinetochores in mitosis.

****Minor comments:****

Figure 1H

I am confused with 4 stars shown at the top of the right plot. If the 4 stars are meant to show a significant difference, then the statement in the text (line 123) is not correct.

"SSRP1 localization was also largely unaffected ..."

Similar discrepancies are found in Figures 3H (line 212), Figures S2 (line 122), S5I (line 197), and S6I (line 209). Figure S6H is not referred to anywhere.

There is no description for the numbers at the top. Are they mean values? Do red bars represent S.D.?

Figure 1D

There is no description of R* to the right of gels.

Figure S2

A 4 hour nocodazole treatment is too short to drive all cells into mitosis. Is the data taken from mitotic cells only?

2. Significance:

Significance (Required)

The interaction of FACT with kinetochore components has been known for several years. However how FACT contributes to architecture or function of kinetochore is not very well understood. How the FACT complex, which is known for its established role as a histone chaperone, is involved in kinetochore assembly/architecture will attract interest in several fields of basic research including epigenetics, mitosis, structural biology.

Identification of CCAN subunits that interact with FACT is important for future analysis to understand the kinetochore function of FACT. The authors identified OPQRU and CHIKM subcomplex in addition to TW as FACT-interacting domains. These subcomplexes are geographically scattered in a 3D model of CCAN holocomplex. Stoichiometry of CCAN and FACT might be informative whether a single or multiple FACT binds to the multiple sites of CCAN. The authors do not address whether these multiple sites are occupied simultaneously, separately or sequentially.

The statement at the end of Abstract (lines 23-25) is a speculative hypothesis without evidence for "a pool of CCAN that is not stably integrated into chromatin", "chaperoning CCAN", and "stabilisation of CCAN".

3. How much time do you estimate the authors will need to complete the suggested revisions:

Estimated time to Complete Revisions (Required)

(Decision Recommendation)

Between 3 and 6 months

4. Review Commons values the work of reviewers and encourages them to get credit for their work. Select 'Yes' below to register your reviewing activity at Web of Science Reviewer Recognition Service (formerly Publons); note that the content of your review will not be visible on Web of Science.

Yes

Review #2

1. Evidence, reproducibility and clarity:

Evidence, reproducibility and clarity (Required)

FACT is a histone chaperone and is involved in various events on chromatin such as transcription and replication. In addition, FACT interacts with various kinetochore components, suggesting potential functions at the kinetochore. However, it is largely unclear how FACT functions in the kinetochore. Authors of this MS took the biochemical approach to understand roles of FACT in the kinetochore.

Authors demonstrated that FACT forms a complex with the constitutive centromere associated network (CCAN), which contains 16 subunits on centromeric chromatin, using multiple binding sites. They also showed that casein kinase II (CK2) phosphorylated FACT and dephosphorylated FACT did not bind to CCAN. Furthermore, they displayed that DNA addition disrupt the stable FACT-CCAN complex.

Overall, while authors have done solid and high-quality biochemical analyses (these are elegant), it is still unclear how FACT plays its roles in the kinetochore. Simple knockout or knockdown study on FACT might be complicated, because FACT has multi-functions. If authors can identify specific regions of FACT for interaction with CCAN, they would put specific mutations into FACT to analyze phenotype. Although they did not reach a high-resolution structure for the FACT-CCAN complex, they can utilize AlphaFold and test specific interaction regions, biochemically. Then, using such information, significance of FACT-CCAN interaction might be testable in cells. Such a kind of study would be expected. In summary, biochemical parts are beautiful, but the paper did not address significance of FACT-CCAN interaction.

****Specific point****

1. Authors showed nice mitotic localization of FACT. Can they observe this localization by a usual IF? Using GFP fusion, do they observe kinetochore localization like IF experiments?
2. On page 7, authors tested CENP-C binding to FACT and they conclude that C-terminal region of CENP-C preferentially binds to FACT. However, they used N-terminal region of CENP-C (2-545) for CCAN-FACT complex formation in entire MS. therefore, this is complicated, and story on CENP-C N-terminal region can be removed from this MS.
3. On page 9, authors suddenly focus on N-terminal tails of CENP-Q and CENP-U. Why did they focus on this region. They should explain this. If they perform a structural prediction, they should describe this point.
4. I agree the fact that FACT phosphorylation is required for FACT-CCAN interaction. They may explain how the phosphorylation contributes to stable FACT-CCAN interaction.
5. Readers really want to know phenotype, if FACT-CCAN interaction was compromised without

disruption of CCAN assembly in cells. Although I agree that FACT has some functions in the kinetochore, it is still unclear what FACT does in the kinetochore.

2. Significance:

Significance (Required)

As mentioned above, biochemical parts are beautiful, but the paper did not address significance of FACT-CCAN interaction.

3. How much time do you estimate the authors will need to complete the suggested revisions:

Estimated time to Complete Revisions (Required)

(Decision Recommendation)

Cannot tell / Not applicable

4. Review Commons values the work of reviewers and encourages them to get credit for their work. Select 'Yes' below to register your reviewing activity at Web of Science Reviewer Recognition Service (formerly Publons); note that the content of your review will not be visible on Web of Science.

Yes

Review #3

1. Evidence, reproducibility and clarity:

Evidence, reproducibility and clarity (Required)

****Main findings:****

The major findings of this paper are:

- Detailed dissection of CCAN subunit interactions and requirements to bind the FACT complex using in vitro reconstituted components
- Binding of FACT and nucleosomes to CENP-C are mutually exclusive
- FACT phosphorylation by CK2 enhances interaction with CCAN
- FACT localization in mitosis depends on the CCAN
- CCAN binding to FACT is outcompeted by DNA and CENP-A nucleosomes

The claims and conclusions of the paper are supported by the data and do not require additional experiments. All experiments include biological replicates and appropriate controls.

****Minor comments****

Intro:

- Line 81: In humans [...], here it is worth mentioning that in Drosophila, FACT subunits have been shown to interact directly with the CENP-A assembly factor CAL1 (Ref 61). This paper is perfunctorily cited once in the context of its implication of FACT in CENP-A deposition, but it merits more consideration when setting up the foundational context for the present work.

Figure 1:

- 1F: Add insets.
- 1G and all other figures containing IFs: Avoid red/green color scheme (red-green colorblindness is fairly common, affecting about 8% of men).
- 1E: Please add a table summarizing interactions.

Results:

- It's fine to direct readers to previous work in which you reconstituted the CCAN, but the text should mention how proteins are exogenously expressed and purified (as done for FACT in line 247).
- Line 113: FACT has been shown to localize to the mitotic kinetochore also in Drosophila (Ref 61).
- Line 132: The authors should cite work from the Drosophila system as well when they mention centromere transcriptional activity in mitosis (e.g., <https://doi.org/10.1083/jcb.201404097>; <https://doi.org/10.1083/jcb.201611087>; and Ref 61).
- Figure 2F: The authors could use a line to mark the region interacting with FACT and that interacting with CENP-A to help summarize the data in this diagram.
- Figure 4: Highlight constructs n.2 (FACT^{TRUNC}) since these are sufficient for interaction (e.g., use a box around them).
- Line 276: "CCAN decodes CENP-A^{NCP}..." What do the authors mean by "decodes"? This whole sentence would benefit from clearer language.
- Figure 6: There's a lot of information in these experiments that would benefit from two schematics, one showing the mechanism of FACT + CCAN binding with DNA and one with CENP-A nucleosomes.

Discussion:

The authors discuss FACT localization at kinetochores in mitosis. In Drosophila Schneider cells, FACT is observed enriched at the centromeres in both mitosis and interphase (Ref 61). The authors mention their inability to detect FACT in interphase in the discussion, but I did not find this mentioned in the results. The authors state that FACT "redistributes to the entire chromosome" upon entry into interphase. They cite Figure 1F in reference to this statement, but the staining in the early G1 panel is difficult to interpret with the low signal/noise scaling of CENP-C and the lack of zoom insets. Their protocol uses a pre-extraction step with Triton prior to fixation. Apparently, this was not enough to reveal FACT in interphase, but better images and a brief description are warranted.

It is unlikely that FACT would change its localization pattern in mitosis. A more likely possibility is

that in mitosis FACT is not redistributed, but rather more tightly bound (and thus less easily extracted by Triton treatment) at kinetochores, while along the arms FACT is more readily removed by extraction because at this time transcription is repressed and FACT is likely less engaged in transcription-mediated histone destabilization.

Given the well-known function of FACT in transcription and the many studies linking transcription to centromere maintenance, including with the involvement of FACT, the model that "the localization of FACT at the kinetochore coincides with active centromeric transcription in mitosis and interphase" is very tempting. A speculative model would go a long way to help the reader visualize all these complex aspects of FACT's interactions and possible functions.

2. Significance:

Significance (Required)

The strongest aspect of the study is the detailed characterization of protein-protein interactions, as well as competition with DNA and CENP-A nucleosomes. The siRNA experiments in cells complement this largely in vitro study. However, a limitation of the study is that it does not shed light on what FACT might be doing at the centromere. Additionally, it does not sufficiently provide context for these findings in relation to previous studies that have demonstrated the roles of FACT at the centromere in budding yeast, fission yeast, and *Drosophila*. Nonetheless, this study provides valuable insights into the details of FACT interactions at the kinetochore and will be of interest to readers interested in centromeres and kinetochore.

I am a centromere biologist with molecular and cell biology expertise.

3. How much time do you estimate the authors will need to complete the suggested revisions:

Estimated time to Complete Revisions (Required)

(Decision Recommendation)

Less than 1 month

No

We thank all three reviewers for their time and engagement, for their generally supportive comments, and for raising some important concerns. We are pleased to submit a significantly revised manuscript where we tried to accommodate all suggested changes and extensions. Importantly, we have included additional experiments that support the relevance of FACT for the overall stability of the inner kinetochore. Below is a detailed point-to-point response. Changes to the manuscript relative to the original submission have been highlighted at the end of this response.

Reviewer #1 (Evidence, reproducibility and clarity (Required)):

Summary:

The authors investigated molecular interactions between CCAN and FACT complexes. They revealed contact domains in FACT and the cognate subcomplexes of CCAN by *in vitro* reconstitution from recombinant proteins followed by SEC and pull-down assay.

They also revealed a couple of potential means to control interactions between FACT and the CCAN. They conclude that phosphorylation of FACT by CK2 is essential for binding to the CCAN; and CENP-A nucleosomes or DNA prevent CCAN from interacting with FACT.

Major comments:

The authors show that phosphorylation of FACT is essential for interaction with CCAN.

They argue that this phosphorylation is partly catalysed by CK2.

My concerns are:

-1- The authors assume that the sites phosphorylated in insect cell are also phosphorylated in human cells. However, it is not demonstrated which residues are phosphorylated in human cells and whether they match those from insect cells. Whether phosphorylation of recombinant proteins in insect cells is physiologically relevant to mammalian is uncertain. Kinetochore components are not very well conserved evolutionarily, thus their regulation may be different.

We thank the reviewer for these remarks, which we answer together with point 2 below.

-2- They identify several residues which are phosphorylated by CK2 *in vitro*. However, these are not necessarily the same sites as those phosphorylated in insect cells or more importantly in human cells. The *in vitro* phosphorylation by CK2 did not restore binding affinity in full, suggesting phosphorylation at other sites may be critical for interaction with CCAN. Further evidence is required to support the claim that those sites are phosphorylated *in vivo* and important for integrity of kinetochores in mitosis.

Our analysis of FACT phosphorylation represents a relatively small part of a very data-rich paper, and was not meant to be exhaustive. Nonetheless, the reviewer's comments are important and well received. We agree that we have no definitive evidence that the same sites are phosphorylated in insect cells, *in vitro*, and in human cells. However, it is quite remarkable, and supports specificity, that the interaction with FACT, lost after dephosphorylation *in vitro*, is restored with CK2 and not with three additional mitotic kinases (CDK1, Aurora B, and PLK1 – Figure S8D). We also note that S437, S444 and S667 of SSRP1, which were phosphorylated by CK2 *in vitro*, were also detected as phosphorylated sites on recombinant FACT purified from insect cells (Table S1). So collectively, while we agree with the reviewer that the analysis of FACT

phosphorylation is not complete, it does significantly add to the manuscript and more generally to the FACT field.

Minor comments:

Figure 1H

I am confused with 4 stars shown at the top of the right plot. If the 4 stars are meant to show a significant difference, then the statement in the text (line 123) is not correct.

"SSRP1 localization was also largely unaffected ..."

Similar discrepancies are found in Figures 3H (line 212), Figures S2 (line 122), S5I (line 197), and S6I (line 209). Figure S6H is not referred to anywhere.

There is no description for the numbers at the top. Are they mean values? Do red bars represent S.D.?

We thank the reviewer for these comments. In this revised version of the manuscript, we have substantially improved the quantification and statistical analysis. The main problem with the previous automated analysis is that the non-circular shape of the CREST-staining led to inconsistencies with the statistical analysis and the statement. In contrast, the same analysis works well when the CENP-C signal was used for KT identification (e.g. in Figure 3), as CENP-C staining yields well separated circular signals ideally suited for our automated identification of individual KTs and subsequent retrieval of fluorescence intensities. We have therefore modified our analysis macro for all experiments where CREST was used as a reference. We used Otsu-thresholding of the DAPI signal for generating a segmentation mask per each cell. Then, integrated cell intensities were calculated for each fluorescence channel based on the DAPI reference mask. With these adjustments, the statistical analyses (Figures 1, S2, S3) support the claim presented. We have updated the Methods and Results sections to reflect the revised analysis.

The numbers on top of the graphs are median values, bars represent interquartile ranges. We have now included the description in figure legends.

We appreciate your feedback, which prompted us to clarify and enhance the rigor of our approach.

We are now referring to Fig. S6H in the text.

Figure 1D

There is no description of R^* to the right of gels.

We have added a description of R^* to the relevant figure legend.

Figure S2

A 4 hour nocodazole treatment is too short to drive all cells into mitosis. Is the data taken from mitotic cells only?

Yes, the data are taken only from the mitotic population. We have now clarified this in the figure legend.

Reviewer #1 (Significance (Required)):

The interaction of FACT with kinetochore components has been known for several years. However how FACT contributes to architecture or function of kinetochore is not very well understood. How the FACT complex, which is known for its established role as a histone chaperone, is involved in kinetochore assembly/architecture will attract interest in several fields of basic research including epigenetics, mitosis, structural biology.

We are grateful to the reviewer for this supportive statement that recognizes the broad potential interest of the manuscript.

Identification of CCAN subunits that interact with FACT is important for future analysis to understand the kinetochore function of FACT. The authors identified OPQRU and CHIKM subcomplex in addition to TW as FACT-interacting domains. These subcomplexes are geographically scattered in a 3D model of CCAN holocomplex. Stoichiometry of CCAN and FACT might be informative whether a single or multiple FACT binds to the multiple sites of CCAN. The authors do not address whether these multiple sites are occupied simultaneously, separately or sequentially.

We thank the reviewer for raising this point. As mentioned in the discussion, we have not yet been able to perform a structural analysis of the FACT/CCAN complex to determine its stoichiometry. However, we have now added a new experiment (Figure S1B,C) where we quantified in-gel tryptophan fluorescence after analytical size-exclusion chromatography. This strongly suggests that FACT and CCAN form a complex with a 1:1 stoichiometry. Nevertheless, we cannot comment on which sites are occupied.

The statement at the end of Abstract (lines 23-25) is a speculative hypothesis without evidence for "a pool of CCAN that is not stably integrated into chromatin", "chaperoning CCAN", and "stabilisation of CCAN".

We agree with the reviewer that this is speculative, and have therefore modified the Abstract to clearly indicate this point.

Reviewer #2 (Evidence, reproducibility and clarity (Required)):

FACT is a histone chaperone and is involved in various events on chromatin such as transcription and replication. In addition, FACT interacts with various kinetochore components, suggesting potential functions at the kinetochore. However, it is largely unclear how FACT functions in the kinetochore. Authors of this MS took the biochemical approach to understand roles of FACT in the kinetochore.

Authors demonstrated that FACT forms a complex with the constitutive centromere associated network (CCAN), which contains 16 subunits on centromeric chromatin, using multiple binding sites. They also showed that casein kinase II (CK2) phosphorylated FACT and dephosphorylated FACT did not bind to CCAN. Furthermore, they displayed that DNA addition disrupt the stable FACT-CCAN complex.

Overall, while authors have done solid and high-quality biochemical analyses (these are elegant), it is still unclear how FACT plays its roles in the kinetochore. Simple knockout or knockdown study on FACT might be complicated, because FACT has multi-functions. If authors can identify specific regions of FACT for interaction with CCAN, they would put specific mutations into FACT to analyze phenotype. Although they did not reach a high-resolution structure for the FACT-CCAN complex, they can utilize AlphaFold and test specific interaction regions, biochemically. Then, using such information, significance of FACT-CCAN interaction might be

testable in cells. Such a kind of study would be expected. In summary, biochemical parts are beautiful, but the paper did not address significance of FACT-CCAN interaction.

We thank the reviewer for praising the biochemical work in our manuscript. The reviewer, however, also underscored the limits of our functional analysis. The reviewer proposes generating separation-of-function mutants in a minimal kinetochore-binding region. Indeed, we have identified the minimal domain for the interaction of FACT with kinetochores. However, this information is insufficient for a reliable functional analysis at this stage, as the region we identified encompasses the AIDs and the phosphorylation-rich region, both of which have been previously shown to be important for transcription and other functions. Furthermore, any suitable mutant should be tested in cells devoid of endogenous FACT, raising the concern that the resulting phenotype may be indirect.

Nonetheless, as we wanted to provide at least an initial answer to the reviewer's concern, we enriched the manuscript by adding experiments in a recently published cell line (K562-SSRP1-dTAG) where FACT levels can be controlled with a small molecule (Žumer *et al.* Mol Cell., 2024) and that the authors generously shared with us. In this line, which grows in suspension and that we had to adapt to grow on a substrate for imaging, we were able to deplete FACT while cells were arrested in mitosis. We are glad to report that we found a significant reduction in the kinetochore levels of CENP-TW after this treatment, which is consistent with other conclusions from our study. These experiments add an initial functional characterization of the interaction of FACT with kinetochores, and extend the significance of the manuscript. We refer to these results again below in response to specific point 5.

Specific point

1. Authors showed nice mitotic localization of FACT. Can they observe this localization by a usual IF? Using GFP fusion, do they observe kinetochore localization like IF experiments?

The localization of FACT was observed using pre-extraction and fixation followed by antibody staining. We have now added a panel demonstrating mitotic localization of GFP-SSRP1 at the kinetochore in transiently transfected RPE-1 cells (Fig. S2A).

2. On page 7, authors tested CENP-C binding to FACT and they conclude that C-terminal region of CENP-C preferentially binds to FACT. However, they used N-terminal region of CENP-C (2-545) for CCAN-FACT complex formation in entire MS. therefore, this is complicated, and story on CENP-C N-terminal region can be removed from this MS.

We were only able to purify full-length CENP-C with tags at the N- and C-terminus, including an MBP tag with a stabilizing effect. At the time of our first successful purification of full-length CENP-C, we had already established the solid phase assay using ^{MBP}FACT as a bait on amylose beads and CENP-C²⁻⁵⁴⁵HIKM as one of the preys. As we cannot obtain stable full-length CENP-C without MBP, this form of CENP-C is incompatible with our pull-down assay. Nevertheless, CENP-C²⁻⁵⁴⁵ still has low affinity for FACT, influencing the FACT/CCAN interaction independent of the PEST-rich region. We, therefore, opted for keeping this information in the manuscript.

3. On page 9, authors suddenly focus on N-terminal tails of CENP-Q and CENP-U. Why did they focus on this region. They should explain this. If they perform a structural prediction, they should describe this point.

Thanks for raising this point. We focused on the N-terminal tails of CENP-QU because they are known interaction hubs. We have now added a sentence to introduce this concept and citing the appropriate literature.

4. I agree the fact that FACT phosphorylation is required for FACT-CCAN interaction. They may explain how the phosphorylation contributes to stable FACT-CCAN interaction.

We have added a sentence explaining that FACT is known to mimic DNA, and negative charges due to phosphorylation could drive this effect. A more detailed mechanistic understanding will require identifying specific phosphorylation sites required for the interaction.

5. Readers really want to know phenotype, if FACT-CCAN interaction was compromised without disruption of CCAN assembly in cells. Although I agree that FACT has some functions in the kinetochore, it is still unclear what FACT does in the kinetochore.

We wholeheartedly agree with the reviewer. As depletion of FACT by RNAi required 48 h, an unreasonably long time for this multifunctional protein. We therefore turned to engineering RPE-1 cells for rapid degradation of SSRP1. While these attempts were unsuccessful, earlier this year, Žumer *et al.* Mol Cell, 2024 reported generating a K562-SSRP1-dTAG cell line growing in suspension. As already reported, this cell line now allowed to demonstrate a significant effect on the kinetochore stability of CENP-TW upon mitotic depletion of FACT.

Reviewer #2 (Significance (Required)):

As mentioned above, biochemical parts are beautiful, but the paper did not address significance of FACT-CCAN interaction.

We thank the reviewer for this positive assessment. In this revision, we have obtained initial evidence that FACT contributes to kinetochore stability.

Reviewer #3 (Evidence, reproducibility and clarity (Required)):

Main findings:

The major findings of this paper are:

- Detailed dissection of CCAN subunit interactions and requirements to bind the FACT complex using in vitro reconstituted components
- Binding of FACT and nucleosomes to CENP-C are mutually exclusive
- FACT phosphorylation by CK2 enhances interaction with CCAN
- FACT localization in mitosis depends on the CCAN
- CCAN binding to FACT is outcompeted by DNA and CENP-A nucleosomes

The claims and conclusions of the paper are supported by the data and do not require additional experiments. All experiments include biological replicates and appropriate controls.

We are thankful to the reviewer for this very positive assessment of our work.

Minor comments

Intro:

- Line 81: In humans [...], here it is worth mentioning that in *Drosophila*, FACT subunits have been shown to interact directly with the CENP-A assembly factor CAL1 (Ref 61). This paper is perfunctorily cited once in the context of its implication of FACT in CENP-A deposition, but it merits more consideration when setting up the foundational context for the present work.

We have extended the Introduction and discuss the specified paper more thoroughly.

Figure 1:

- 1F: Add insets.

Done.

- 1G and all other figures containing IFs: Avoid red/green color scheme (red-green colorblindness is fairly common, affecting about 8% of men).

Done.

- 1E: Please add a table summarizing interactions.

We have included this table as Fig. S1E.

Results:

- It's fine to direct readers to previous work in which you reconstituted the CCAN, but the text should mention how proteins are exogenously expressed and purified (as done for FACT in line 247).

Done.

- Line 113: FACT has been shown to localize to the mitotic kinetochore also in *Drosophila* (Ref 61).

We have included this information now.

- Line 132: The authors should cite work from the *Drosophila* system as well when they mention centromere transcriptional activity in mitosis (e.g. <https://doi.org/10.1083/jcb.201404097>; <https://doi.org/10.1083/jcb.201611087>; and Ref 61).

We have added these citations.

- Figure 2F: The authors could use a line to mark the region interacting with FACT and that interacting with CENP-A to help summarize the data in this diagram.

Done.

- Figure 4: Highlight constructs n.2 (FACT^{TRUNC}) since these are sufficient for interaction (e.g, use a box around them).

Done.

- Line 276: "CCAN decodes CENP-A^{NCP}..." What do the authors mean by "decodes"? This whole sentence would benefit from clearer language.

We thank the reviewer for this suggestion and have aimed for clearer language.

- Figure 6: There's a lot of information in these experiments that would benefit from two schematics, one showing the mechanism of FACT + CCAN binding with DNA and one with CENP-A nucleosomes.

Done.

Discussion:

The authors discuss FACT localization at kinetochores in mitosis. In *Drosophila* Schneider cells, FACT is observed enriched at the centromeres in both mitosis and interphase (Ref 61). The authors mention their inability to detect FACT in interphase in the discussion, but I did not find this mentioned in the results. The authors state that FACT "redistributes to the entire chromosome" upon entry into interphase. They cite Figure 1F in reference to this statement, but the staining in the early G1 panel is difficult to interpret with the low signal/noise scaling of CENP-C and the lack of zoom insets. Their protocol uses a pre-extraction step with Triton prior to fixation. Apparently, this was not enough to reveal FACT in interphase, but better images and a brief description are warranted.

We have now added a staining of SSRP1 in interphase in the panel.

It is unlikely that FACT would change its localization pattern in mitosis. A more likely possibility is that in mitosis FACT is not redistributed, but rather more tightly bound (and thus less easily extracted by Triton treatment) at kinetochores, while along the arms FACT is more readily removed by extraction because at this time transcription is repressed and FACT is likely less engaged in transcription-mediated histone destabilization.

We thank the reviewer for these remarks and have updated the Discussion.

Given the well-known function of FACT in transcription and the many studies linking transcription to centromere maintenance, including with the involvement of FACT, the model that "the localization of FACT at the kinetochore coincides with active centromeric transcription in mitosis and interphase" is very tempting. A speculative model would go a long way to help the reader visualize all these complex aspects of FACT's interactions and possible functions.

We agree with the reviewer that such a model is tempting. However, we also feel that it would be rather speculative at this stage and we feel that the manuscript does not provide sufficient data to support the model.

Reviewer #3 (Significance (Required)):

The strongest aspect of the study is the detailed characterization of protein-protein interactions, as well as competition with DNA and CENP-A nucleosomes. The siRNA experiments in cells complement this largely in vitro study. However, a limitation of the study is that it does not shed light on what FACT might be doing at the centromere. Additionally, it does not sufficiently provide context for these findings in relation to previous studies that have demonstrated the

roles of FACT at the centromere in budding yeast, fission yeast, and *Drosophila*. Nonetheless, this study provides valuable insights into the details of FACT interactions at the kinetochore and will be of interest to readers interested in centromeres and kinetochore. I am a centromere biologist with molecular and cell biology expertise.

We are very grateful to the reviewer for his/her support.

Interactions with multiple inner kinetochore proteins determine mitotic localization of FACT

Short title: The CCAN binds FACT

Julia Schweighofer^{1,2,*†}, Bhagyashree Mulay^{1,2,*†}, Ingrid Hoffmann¹, Doro Vogt¹, Marion E. Pesenti¹ & Andrea Musacchio^{1,2,3*}

¹Department of Mechanistic Cell Biology, Max Planck Institute of Molecular Physiology, Otto-Hahn-Straße 11, 44227 Dortmund, Germany.

²Centre for Medical Biotechnology, Faculty of Biology, University Duisburg-Essen, Essen, Germany.

³Max Planck School Matter to Life, Jahnstrasse 29, 69120 Heidelberg, Germany.

† Equal contribution

*Corresponding authors: juliamaria.schweighofer@mpi-dortmund.mpg.de; bhagyashree.mulay@mpi-dortmund.mpg.de; andrea.musacchio@mpi-dortmund.mpg.de

Abstract

The FAcilitates Chromatin Transcription (FACT) complex is a dimeric histone chaperone that operates on chromatin during transcription and replication. FACT also interacts with a specialized centromeric nucleosome containing the histone H3 variant CENP-A and with CENP-TW, two subunits of CCAN, a 16-protein complex associated with CENP-A. The significance of these interactions remains elusive. Here, we show that FACT has multiple additional binding sites on CCAN. The interaction with CCAN is strongly stimulated by casein kinase II (CK2) phosphorylation of FACT. Mitotic localization of FACT to kinetochores is strictly dependent on specific CCAN subcomplexes. Conversely, CENP-TW requires FACT for stable localization. Unexpectedly, we also find that DNA readily displaces FACT from CCAN, suggesting supporting the speculation that FACT becomes recruited through a pool of CCAN that is not stably integrated into chromatin. Collectively, our results point to a potential role of FACT in chaperoning CCAN during transcription or in the stabilization of CCAN at the centromere during the cell cycle.

Teaser

DNA-sensitive, direct interactions with multiple inner kinetochore subunits deliver FACT to the kinetochore.

MAIN TEXT

Introduction

Chromosomes are DNA packaging structures that consist of a single molecule of DNA and many different associated proteins. They contain several functionally specialized regions that work in conjunction with transcription, replication, and inheritance. A notable specialized chromatin locus is the centromere. The histone H3 variant centromere protein A (CENP-A) is greatly enriched at centromeres and is considered the crucial epigenetic marker of centromeres. CENP-A seeds the kinetochore, a large protein complex that connects the replicated chromosomes (sister chromatids) to spindle microtubules during mitosis to ensure their equal distribution to the daughter cells (1, 2). Its presence at centromeres recruits specialized machinery that delivers new CENP-A at every cell cycle to compensate for its dilution during DNA replication (3).

The kinetochore is divided into inner and outer layers (4). The outer layer, consisting of 10 proteins collectively referred to as the Knl1 complex, Mis12 complex, Ndc80 complex (KMN) network and associated proteins is assembled during mitosis to directly attach to spindle microtubules (5). The inner layer, consisting of 16 proteins collectively referred to as the constitutive centromere associated network (CCAN), bridges the centromeric chromatin and outer kinetochore and localizes to the centromere throughout the cell cycle (6, 7). The CCAN consists of different subunits and subcomplexes, including CENP-C, CENP-HIKM, CENP-LN, CENP-OPQUR, and CENP-TWSX (8) (Fig 1A).

Two CCAN proteins, CENP-C and CENP-N, decode the centromere by recognizing CENP-A (9–15). In addition to binding CENP-A, CENP-C interacts directly with other inner kinetochore subunits, including CENP-HIKM and CENP-LN as well as the outer kinetochore complex MIS12 (13, 16–20). A second subunit, CENP-T, binds stably to CENP-W and connects the CCAN and the outer kinetochore by interacting with Mis12 and Ndc80 complexes (Mis12C and Ndc80C, respectively) through its long disordered N-terminal tail (21, 22). CENP-W and the C-terminal region of CENP-T consist of a histone fold domain (HFD). The CENP-TW subcomplex further tetramerizes with two additional HFD-containing proteins, CENP-S and CENP-X. It has been reported that the resulting CENP-TWSX complex is integrated into centromeric chromatin as a nucleosome-like particle (23, 24). Recent structural work has shown

that CCAN consists of two structural pillars (composed of CENP-HIKM and CENP-OPQUR) flanking a central DNA-binding vault (contributed by CENP-LN) and a base (CENP-TWSX; Fig. 1B). The central vault enables tight binding of linker DNA by CCAN. *In vitro* and *in vivo*, CENP-A has been shown to form an octameric nucleosome consisting of a CENP-A/H4 tetramer flanked by two H2A/H2B dimers wrapped by ~150 base pairs (bp) DNA (25). The CENP-A nucleosome has been proposed to neighbour the CCAN structure bound to linker DNA (26–28).

The original CENP-A co-precipitation experiments that identified CCAN subunits also identified the FACilitates Chromatin Transcription (FACT) complex for a specific interaction with CENP-A nucleosomes (6, 29–32). FACT is a H2A/H2B chaperone that prevents histone loss whilst facilitating assembly and disassembly of nucleosomes during transcription (33–36). Additionally, it has been implicated in DNA replication and repair (37–46). FACT is a heterodimer of Suppressor of Ty protein 16 (SPT16) and Structure-specific recognition protein 1 (SSRP1), both large multi-domain proteins with an array of Plekstrin homology (PH) domains (47–49). SPT16 has an N-terminal peptidase-like domain, which has lost its catalytic activity but interacts with Mini Chromosome Maintenance protein complex (MCM) 2-7 and with the fork protection complex during replication, as well as with the Set3 histone deacetylase complex (50–52). SPT16 Mid domain binds to histone H3/H4 tetramers. The subsequent acidic intrinsically disordered (AID) segment associates with H2A/H2B dimers (53–55). SSRP1 contains a high mobility group (HMG) domain, which is associated with DNA binding (56, 57) (Fig. 1C).

~~The~~ While the precise significance of ~~theirs~~ interaction ~~of FACT~~ with centromeres remains elusive, ~~FACT is believed to promote CENP-A deposition and to prevent ectopic localization of CENP-A.~~ In chicken DT40 cells, for instance, FACT and CHD1 are targeted to the kinetochore by CENP-HIKM to facilitate CENP-A deposition (58); In *Drosophila melanogaster*, FACT assists in transcription-coupled CENP-A deposition by directly binding to the CENP-A assembly factor CAL1 (59); In budding yeast, the E3 ubiquitin ligase Psh1 requires binding to FACT to efficiently ubiquitinate CENP-A^{Cse4}, targeting it for proteasomal degradation (60) ~~and~~. Similarly, in fission yeast, ~~the mutation of FACT leads to the accumulation of over-expressed CENP-A^{Cnp1} at non-centromeric chromatin (61).~~ Furthermore, ~~FACT has been implicated in CENP-A deposition and in preventing ectopic localization of CENP-A the maintenance of pericentromeric heterochromatin and the deletion of SSRP1^{Pob3} results in chromosome missegregation (62).~~ In humans, FACT has been shown to directly interact with CENP-TW HFDs via the AID of SPT16 (63). In this study, we demonstrate that the interaction of FACT with CCAN is complex, with additional binding sites on CENP-C and CENP-OPQUR. FACT

engages in a stable 18-subunit complex with CCAN, whose assembly requires the phosphorylation of FACT by the constitutively active kinase casein kinase II (CK2). Mitotic localization of FACT at the kinetochore is dominated by CENP-HIKM and CENP-TW, and we show that CENP-TW levels are reduced upon FACT depletion. We find that DNA displaces FACT from CCAN, suggesting a potential role of FACT in chaperoning CCAN during transcription or in the deposition of CCAN at the centromere during or after replication.

Results

FACT forms a stable complex with CCAN *in vitro*

As CCAN and FACT co-precipitate with CENP-A nucleosomes and FACT has been proposed to bind directly to CCAN subunits, we asked if a CCAN/FACT complex could be reconstituted *in vitro* using recombinant proteins. Previously, we have reconstituted a 16-subunit CCAN from four stable recombinant subcomplexes, including CENP-CHIKM (assembled with C²⁻⁵⁴⁵, i.e. with a fragment of CENP-C encompassing residues 2-545), CENP-LN, CENP-OPQUR and CENP-TWSX (26, 64, 65). We reconstituted CCAN starting from these subcomplexes (Fig. S1A). In analytical size-exclusion chromatography (SEC) experiments, FACT and CCAN co-eluted in a single peak and at earlier elution volumes relative to the individual complexes, indicating that CCAN and FACT bind directly in an 18-subunit complex (Fig. 1D). Addition of excess FACT did not result in a larger shift, and quantification of tryptophan fluorescence of the bands in SDS-PAGE in peak fractions indicated approximately equal amounts of SPT16 and various CCAN subunits, suggesting a 1:1 stoichiometry of FACT and CCAN (Fig. S1B,C).

To identify CCAN subunits involved in FACT binding, we immobilized FACT on amylose-resin through an N-terminal MBP-tag on SPT16 (^{MBP}FACT), and used the various CCAN subcomplexes as preys. In addition to confirming the previously reported interaction with CENP-TW (63), we observed interactions with CENP-OPQUR and CENP-C²⁻⁵⁴⁵HIKM (Fig. 1E). The latter interaction required CENP-C²⁻⁵⁴⁵, because the CENP-HIKM complex, which lacks CENP-C, did not bind (Fig. 1E, lane 3), as opposed to previously published experiments with avian CCAN, which suggest). This extends previous observations suggesting a direct interaction of FACT with CENP-HIKM (58). CENP-SX, which contains HFDs similar to CENP-TW (24), did not bind to ^{MBP}FACT (Fig. 1E, lane 7). We confirmed the association of FACT with CENP-TW and CENP-OPQUR by analytical SEC, whereas FACT and CENP-C²⁻⁵⁴⁵HIKM did not form a stable complex in solution (Fig. S4BS1D). We conclude that FACT

and CCAN bind directly, and that the interaction is mediated by multiple binding interfaces- (Fig. S1E).

Mitotic localization of FACT to the kinetochore depends on the CCAN

FACT localizes to chromatin, especially nucleoli, in interphase, reflecting its role in transcription (66, 67). ~~In chicken DT40 cells,~~ FACT was also observed to localize at centromeres during mitosis in chicken DT40 cells and to interphase and mitotic centromeres in *Drosophila* (58, 59). To investigate mitotic FACT localization in human cells, we stained SSRP1 by immunofluorescence in hTERT-immortalized retinal pigment epithelial (RPE-1) cells. FACT localized to the kinetochore in all mitotic phases, exhibiting a more diffuse signal in early and late mitosis and interphase (Fig. 1F, S2A). To dissect how FACT is recruited to kinetochores during mitosis, we exploited a previously described colorectal adeno-carcinoma DLD-1 cell ~~for line~~ allowing rapid degradation of CENP-C (68). In this system, both CENP-C alleles are endogenously tagged with an auxin-inducible degron (AID) (69) and an enhanced yellow fluorescent protein (EYFP). After ~~a 4-hour treatment of treating mitotic cells arrested in mitosis by nocodazole treatment~~ with the auxin derivative indole acetic acid (IAA), for 4 hours, CENP-C was completely depleted from kinetochores. ~~Instead,~~ while CENP-HK and CENP-TW localization was remained largely unaffected, as previously observed (26). SSRP1 localization was also largely unaffected (Fig. 1G,H and Fig. S2S2B-I), indicating that recruitment of FACT is independent of CENP-C or that FACT remains stably localized after initial depletion of CENP-C. When the treatment with IAA was extended to 24 hours, however, the kinetochore levels of FACT were greatly decreased. This correlated with modest to strong decreases in CCAN subunit localization (Fig. 1I,J and Fig. S3). ~~Of note, these experiments also revealed that the course of depletion of CCAN subunits after CENP-C degradation in mitotically arrested cells does not completely recapitulate the pattern of depletion seen in cycling cells (Fig. S2 and Fig. S3).~~ Collectively, these observations link kinetochore localization of FACT to the interactions with CCAN observed *in vitro*, although they do not exclude a potential role of centromere transcriptional activity in the recruitment and retention of FACT during mitosis (59, 70–75).

Cooperative and anti-cooperative FACT/CCAN binding

To further characterize how individual interactions of CCAN and FACT stabilize their assembly, we titrated CCAN subcomplexes in different combinations in an *in vitro* pull-down assay with ^{MBP}FACT as bait (Fig. 2A). We quantified the results using the band intensities of CENP-M,

CENP-U, CENP-L and CENP-W, which were well resolved in SDS-PAGEs, as representative of their cognate CCAN subcomplexes (Fig. 2B-E). As shown above (Fig. 1E), CENP-C²⁻⁵⁴⁵HIKM and CENP-OPQUR bound FACT (Fig. 2A). A CENP-TW complex consisting only of the histone fold domain (HFD) of these proteins (CENP-T^{458-C}/full-length CENP-W, henceforth CENP-TW^{HFD}) also bound FACT (Fig. 2A, lane 4). However, CENP-M, CENP-U and CENP-W exhibited a markedly lower band intensity when their cognate subcomplex was exposed to FACT without the other CCAN subcomplexes (Fig. 2A-E, lanes 2-4). When exposed to additional subcomplexes (lanes 5-12), stronger binding was observed. Notably, addition of CENP-C²⁻⁵⁴⁵ to HIKM (instead of isolated CENP-HIKM) enhanced binding when certain subcomplexes were omitted (e.g. CENP-TW in lanes 5, 6; CENP-OPQUR and CENP-LN in lanes 7, 8; and CENP-OPQUR in lanes 9, 10). ~~However,~~ suggesting that CENP-C²⁻⁵⁴⁵ stabilizes interactions of incomplete CCAN subcomplexes. Indeed, when the complete CCAN was used as prey, absence of CENP-C²⁻⁵⁴⁵ did not significantly change the level of bound CCAN subunits (lanes 11,12). Collectively, these results are consistent with the idea that FACT binding involves multiple interaction interfaces of CCAN.

CENP-C binds to CENP-HIKMLN through its PEST (proline-glutamic acid-serine-threonine)-rich region (CENP-C¹⁸⁹⁻⁴⁰⁰, CENP-C^{PEST}) (13, 16, 18, 19) (Fig. 2F). CENP-C^{PEST}, however, was neither capable of a direct interaction with FACT when combined with CENP-HIKM, nor did it trigger increased binding of CENP-HIKMLNOPQR to ^{MBP}FACT (Fig. 2G). These observations suggested that CENP-C and FACT ~~may~~ bind directly outside ~~of~~ the CENP-C^{PEST}. To identify regions of CENP-C involved in FACT binding, we divided the sequence of CENP-C in different fragments, expressed them as fusions to MBP, and used them as baits in a pull-down assay. FACT bound CENP-C⁴⁰¹⁻⁵⁴⁵, CENP-C⁴⁰¹⁻⁶⁰⁰, CENP-C⁵⁴⁶⁻⁶⁰⁰, CENP-C⁷²¹⁻⁷⁵⁹, and CENP-C⁷²¹⁻⁹⁴³ (Fig. S4A), which collectively encompass 1) the CENP-C central region, 2) a region adjacent to the central region, 3) the CENP-C conserved motif, and 4) the C-terminal cupin domain involved in dimerization. As both the central region and the CENP-C conserved motif bind specifically to CENP-A nucleosomes (CENP-A nucleosome core particle, CENP-A^{NCP}), these observations suggest that FACT ~~may stabilize~~ stabilizes the CENP-A nucleosome binding region of CENP-C in the absence of nucleosomes (Fig. 2F) (9, 10, 12, 15). Confirming this conclusion, inclusion of CENP-A^{NCP} in a pull-down assay where FACT was bound to immobilized ^{MBP}CENP-C^{EGFP} (a full-length CENP-C construct) caused FACT to dissociate (Fig. S4B), indicating that binding of FACT and nucleosomes to CENP-C is mutually exclusive.

An unexpected aspect of the CCAN interaction with FACT is that the addition of CENP-OPQUR appeared to reduce the levels of CENP-HIKM and CENP-TW (using CENP-M and

CENP-W as readouts, respectively; Fig. 2A lanes 11, 12 and quantified in Fig. 2B and E). Within CCAN, CENP-OPQUR and CENP-TW do not directly bind to each other and require CENP-LN and CENP-HIKM for their interaction (26, 27, 64). As they are both able to bind FACT, however, we anticipated that FACT may bridge these complexes. Contrary to this expectation, CENP-OPQUR and CENP-TW competed for FACT, with CENP-TW showing a higher affinity for FACT (Fig. 2H).

To investigate this phenomenon further, we tried to shed light on the determinants of the interaction of FACT with CENP-OPQUR. We found the disordered N-terminal tails of CENP-Q and CENP-U, known interaction hubs of the CENP-OPQUR complex (64, 76–79), to be required for ~~CENP-OPQUR~~/FACT binding, because a truncation of these tails (CENP-OPQ^{68-C}U^{115-C}R, herewith indicated as CENP-OPQ^{ΔN}U^{ΔN}R) completely abolished the association with MBPFACT (Fig. S4C). This result was confirmed *in vivo*, where FACT was identified in immunoprecipitates (IPs) of ^{EGFP}CENP-U but not of CENP-U^{115-C} (Fig. S4D). However, CENP-C²⁻⁵⁴⁵HIKMLNOPQ^{ΔN}U^{ΔN}R bound MBPFACT, probably because CENP-C²⁻⁵⁴⁵ provides sufficient binding affinity for the FACT complex. Even though CENP-OPQ^{ΔN}U^{ΔN}R does not bind FACT, it continued to oppose binding of FACT to CENP-TW (Fig. S4C), possibly through an allosteric mechanism.

~~FACT localization to the and CCAN interdependence for kinetochore requires CENP-HIKM and TW localization~~

As the mitotic localization of FACT to the kinetochore requires intact CCAN (Fig. 11-J), we wanted to investigate how individual CCAN subcomplexes contribute to FACT localization. RNA interference (RNAi) was used to deplete CCAN subcomplexes in RPE-1 cells and mitotic cells were immunostained for SSRP1. RPE-1 cells were treated with siRNA against CENP-HIKM for 72 hours to deplete the complex from the kinetochore (Fig. S5A-B,D). As a result, localization of SSRP1 was severely affected. CENP-OPQUR and CENP-TW localization was also substantially reduced, indicating that depletion of the CENP-HIKM subcomplex destabilizes CCAN (Fig. 3A-C, Fig. S5E,F). Conversely, CENP-A was not perturbed upon CENP-HIKM depletion (Fig. S5B,C).

A 60-hour CENP-T RNAi treatment eliminated CENP-TW from the kinetochore (Fig. S5G, Fig. 3D,F). Also in this case, a concomitant decrease in the kinetochore levels of FACT, CENP-HK, and CENP-O was observed, whereas the levels of CENP-A remained stable (Fig. 3D,E; Fig. S5H-I). ~~While rapidAs shown above, acute~~ depletion of CENP-C ~~showed that did not affect~~

FACT localization ~~does not directly depend on CENP-C, at least in the short term~~ (Fig. 1G-H). ~~Despite its persistence at kinetochores upon CENP-T depletion,~~ CENP-C was also ~~not sufficient~~ to retain FACT at ~~the kinetochore~~ kinetochores (Fig. 3A-F). Thus, collectively, the CCAN subcomplexes are inter-dependent for their localization, in agreement with previous literature (18, 64, 79–81). Furthermore, our results demonstrate that FACT localization at the kinetochore during mitosis depends on the CCAN (Fig. 1I-J, Fig. 3A-F).

To assess a potential contribution of CENP-OPQUR to the recruitment of FACT, we endogenously tagged both alleles of CENP-U with FKBP^{F26V} and used the resulting cell line to rapidly degrade CENP-U through addition of dTAG^V-1 (82). A 24-hour treatment led to the complete loss of CENP-U and CENP-R from the kinetochore, suggesting that the entire CENP-OPQUR complex, not only CENP-U, are removed. ~~Furthermore~~ In agreement, Polo-like kinase 1 (PLK1) localization, which partially depends on CENP-OPQUR (79), decreased (Fig. S6A-F). On the other hand, localization of CENP-A, CENP-TW and CENP-HK did not require CENP-U (Fig. 3G,I; Fig. S6G-J). In fact, CENP-TW displayed an increase in its kinetochore levels (Fig. 3I). This may indicate competition between CENP-TW and CENP-OPQUR within the CCAN, but may also reflect a staining artifact caused by enhanced accessibility of the antigen. Finally, FACT localization was not affected by the depletion of CENP-U (Fig. 3G-H), indicating that CENP-OPQUR is not necessary for recruiting or retaining FACT at the kinetochore, even if it interacts with FACT *in vivo*, as suggested by co-immunoprecipitation (Fig. S4D). Alternatively, CENP-OPQUR and FACT may interact in a separate complex outside of CCAN. Thus, collectively, our results demonstrate the importance of CCAN, even if we cannot point to a single CCAN subunit as a recruiter of FACT. Kinetochore localization of FACT is substantially reduced upon depletion of CENP-HIKM or CENP-TW, a condition that additionally triggers a reduction of CCAN stability. Conversely, CENP-C and CENP-OPQUR are not strictly required for the localization of FACT to kinetochores.

Our data so far indicate that CCAN promotes kinetochore recruitment of FACT. To assess if FACT is also required to stabilize CCAN at the kinetochore, we used a previously established chronic myeloid leukemia K562 cell line where SSRP1 is endogenously tagged with a dTAG degron for rapid degradation (83)

Figure 1. CCAN binding requires FACT dimerization and Mid-AID domains. (A) Scheme of FACT domains or truncations used as prey in the following pull-down assays. (B) Amylose-resin pull-down assay with ^{MBP}CENP-C^{EGFP} to analyze binding of the FACT constructs in (A). (C) Glutathione-agarose pull down assay using CENP-OPQUR with CENP-U fused to an N-terminal GST as bait and FACT constructs in (A) as preys. (D) Amylose-resin pull-down assay with ^{MBP}CENP-T/CENP-W as bait and FACT constructs in (A) as preys. (E) Amylose-resin pull-down assay comparing CCAN binding to ^{MBP}FACT and ^{MBP}FACT^{trunc}.

. The levels of CCAN subcomplexes were analyzed by immunofluorescence in mitotic cells 8 hours after depleting FACT in STLC-arrested cells (Fig. S7A,B). CENP-A, CENP-C and CENP-HK were not significantly influenced by the rapid depletion of FACT (Fig. S7C-G). On the contrary, kinetochore levels of CENP-TW were significantly decreased (Fig. 3J,K). We also observed a minor reduction in the kinetochore levels of CENP-U, possibly caused by absence of CENP-TW (Fig. S7F,H). These results suggest a potential role for FACT in stabilizing CENP-TW at the centromere.

FACT dimerization and Mid-AID domains are required for CCAN binding

The FACT subunits SPT16 and SSRP1 are multi-domain distant paralogs with distinct functions (84). To identify binding sites for CCAN, we produced different truncations or isolated domains of FACT (Fig. 4A) and used them as preys in pull-down assays with CCAN subcomplexes as baits. As already observed, ^{MBP}CENP-C^{EGFP} and FACT bound directly (Fig. S4B, Fig. 4B). Additionally, ^{MBP}CENP-C^{EGFP} pulled down SPT16^{Mid-AID}, a minimal SPT16, fragment. It also pulled down, with apparently slightly higher affinity, construct 2 (SPT16⁵⁰⁸⁻⁹⁸⁸/SSRP1¹⁻⁵¹⁴, henceforth FACT^{trunc}) (Fig. 4A,B). GST-tagged CENP-OPQUR interacted robustly with FACT and FACT^{trunc}, but only weakly with SPT16^{Mid-AID} (Fig. 4C). Conversely, MBP-tagged CENP-TW (^{MBP}CENP-TW) was sufficient to bind SPT16^{Mid-AID} (Fig. 4D). ^{MBP}CENP-TW also bound strongly to FACT^{trunc} and with low affinity to SSRP1^{Mid-AID} (Fig. 4D). Thus, association of ^{MBP}CENP-TW with either SPT16 and SSRP1 Mid-domain depended on the presence of an intact AID domain (Fig. 4D, lanes 3-6). In summary, FACT^{trunc} was sufficient to bind all CCAN subcomplexes, while the N-terminal aminopeptidase-like domain of SPT16 and the C-terminal HMG-domain of SSRP1 are dispensable for CCAN-binding (Fig. 4B-D). In a reverse pull-down, the apparent strengths of the interaction of CCAN with either full-length ^{MBP}FACT or ^{MBP}FACT^{trunc} were identical (Fig. 4E).

FACT requires phosphorylation by CK2 to interact with CCAN

FACT is regulated by, and also directly binds to, acidophilic casein kinase II (CK2) (38, 85–88). The CK2 holoenzyme is a tetramer composed of the active subunit CK2 α or CK2 α' and the

regulatory and dimerizing subunit CK2 β (89). CK2 is a promiscuous kinase with hundreds of different substrates involved in numerous biological processes and diseases (90). It is characterized as a constitutively active kinase and its regulation is not defined by a single mechanism, but rather is substrate specific (91). Despite its localization to different cellular compartments, CK2 is mostly active in the nucleus (92, 93), where it has a role in transcription (86, 87, 94, 95).

Field Code Changed

Recombinant FACT purified from insect cells was strongly phosphorylated, but treatment with λ -phosphatase removed phosphorylation (Fig. S7AS8A). The elution volume of FACT was unaffected by changes in its phosphorylation status (Fig. S7BS8B). Unexpectedly, dephosphorylated FACT (repurified to eliminate λ -phosphatase) failed to bind CCAN in a SEC co-elution assay (Fig. 5A). Addition of CK2 to the reaction to induce phosphorylation of FACT restored the binding of FACT/CCAN in analytical SEC (Fig. 5A). The phosphorylation dependency of FACT/CCAN complex formation was corroborated in a solid-phase assay (Fig. 5B, lanes 11-13). This assay was also used to probe the phosphorylation dependency of the interaction of FACT with specific CCAN subcomplexes. Dephosphorylated ^{MBP}FACT failed to pull down CENP-C²⁻⁵⁴⁵HIKM, CENP-OPQR and CENP-TW. These interactions were partially restored upon CK2 phosphorylation, although not to the levels observed with the sample before dephosphorylation (Fig. 5B), probably due to incomplete rephosphorylation (Fig. S7CS8C). Some CCAN subunits, including CENP-C and/or CENP-I, CENP-U and CENP-T, were also phosphorylated by CK2 (Fig. S7CS8C). Of note, additional kinases demonstrated an ability to phosphorylate FACT, but they ~~could not~~ failed to restore the interaction with CCAN, indicating that the effects on CCAN binding are specific to CK2 (Fig. S7DS8D).

FACT constructs described in Figure 4 were used to ~~analyse~~ analyze which regions of the complex are phosphorylated by CK2. SPT16 and SSRP1 Mid-AID domains were phosphorylated in an AID-dependent manner (Fig. 5C), in agreement with the ability of CK2 to phosphorylate acidic sequences (96). Additionally, the C-terminal region of SSRP1 was also phosphorylated by CK2 (Fig. 5C). Phosphorylated FACT complex was subjected to mass spectrometry analysis to identify target sites. This analysis failed to identify the precise phosphorylation site within the AID sequences, but in combination with sequence-based prediction (97-99) and published phosphorylation sites (86, 88), we propose more than ~~thirty~~ twenty potential CK2 phosphorylation sites on FACT (Fig. 5D, Table S1). ~~In conclusion, by dephosphorylating insect cell-expressed~~ This large number of phosphorylation sites, expected to ~~render~~ FACT, we revealed that the interaction with more negatively charged, likely facilitates interactions with positively charged DNA-binding interfaces of CCAN is phosphorylation-

~~dependent. We identified the required kinase as CK2, but we could not determine specific phosphorylation sites necessary for binding.~~ Due to the constitutive activity of CK2 (91), cellular conditions leading to phosphorylation of FACT by CK2 remain unclear.

DNA competes with FACT for CCAN binding

~~The CCAN de~~~~codes~~~~recognizes the centromere by direct binding of~~ CENP-A^{NCP} A nucleosomes through CENP-C and CENP-N ~~and binds while also binding to~~ DNA ~~cooperatively~~ (9–15, 26), while FACT only binds to nucleosomes that are partially destabilized, e.g. by an actively transcribing RNA polymerase II (55, 83, 100–102). We set out to further dissect FACT's interaction with CCAN on chromatin using biochemical reconstitution. In analytical SEC, addition of a 145-bp DNA fragment prevented assembly of the FACT/CCAN complex altogether, as DNA binding to CCAN displaced FACT from the complex (Fig. 6A). The same effect was observed upon addition of a 75-bp DNA fragment (Fig. S8A-S9A). This result was confirmed in a solid-phase assay (Fig. 6B, lanes 8,9). CCAN binds DNA very tightly, whereas individual subcomplexes bind to DNA with much lower affinity, if at all (26, 27). We therefore asked how DNA influenced the interaction of FACT with the CCAN subcomplexes CENP-C²⁻⁵⁴⁵HIKM, CENP-OPQUR and CENP-TW. Although a marginal reduction in binding of each complex was detected (Fig. 6B, lanes 2-7), the effect of DNA was considerably less pronounced than in presence of the complete CCAN (Fig. 6B).

Next, we tested whether CCAN binding to FACT is compatible with binding to a CENP-A nucleosome core particle (NCP, on a 145 bp Widom 601 sequence). Recombinant FACT did not bind to DNA or intact nucleosomes in analytical SEC (Fig. S8B-S9B). When mixed with CCAN and a CENP-A^{NCP}, however, a tripartite complex formed (Fig. 6C). These observations were corroborated using a pull-down assay, where CENP-A^{NCP} was seen to interact with ^{MBP}FACT through CCAN (Fig. 6D). The association of CCAN subcomplexes was also evaluated. CENP-OPQUR and CENP-TW did not bind to nucleosomes and the interaction with FACT was essentially unaltered. Binding of CENP-C²⁻⁵⁴⁵HIKM was marginally reduced upon the addition of CENP-A^{NCP} (Fig. 6D). We ligated a naked DNA sequence to a CENP-A nucleosome built on α -satellite DNA, creating a 348-bp sequence of which roughly half was embedded in a nucleosome. The overhanging DNA acted comparably to free DNA, effectively displacing FACT from a CCAN/CENP-A^{NCP} complex (Fig. S8C-S9C,D). Collectively, these results suggest that FACT may recognize a form of CCAN that is not directly bound to DNA at centromeres, despite substantial biochemical and structural information indicating that CCAN binds DNA

tightly through the CENP-LN vault and potentially through various neighbouring DNA-binding structures (26–28, 103). Whether a form of CCAN devoid of DNA is present at kinetochores during mitosis, however, remains unclear.

FACT cannot bind centromeric histones and CCAN simultaneously

As a histone chaperone, FACT is known to bind to both H2A/H2B dimers and H3/H4 tetramers (54, 55). H2A/H2B and CENP-TW compete for the same binding site on FACT, and FACT has a binding preference for the former (63). We wanted to broaden our analysis of the FACT/CCAN complex in relation to centromeric histones. A trimeric complex of histones CENP-A/H4 with the first 80 residues of its chaperone HJURP was added to a pull-down assay with ^{MBP}FACT as bait and CCAN as prey. CENP-A/H4/HJURP¹⁻⁸⁰ outcompeted CCAN in FACT binding (Fig. 6E). This result was confirmed in an orthogonal assay, where ^{MBP}HJURP¹⁻⁸⁰ in complex with CENP-A/H4 as bait bound FACT efficiently, but CCAN was excluded from the complex (Fig. S9AS9E). ^{MBP}HJURP¹⁻⁸⁰ in absence of CENP-A/H4, used as control, did not bind FACT nor CCAN (Fig. S9AS9E), indicating that competition with CCAN for FACT binding is caused by the CENP-A/H4 dimer rather than the chaperone. Binding of CCAN subcomplexes to FACT was also disrupted upon addition of CENP-A/H4/HJURP¹⁻⁸⁰ (Fig. 6E), suggesting that CENP-A/H4 and CCAN may share the same binding site on FACT. This was validated by testing the binding of different FACT constructs to CENP-A/H4/^{MBP}HJURP¹⁻⁸⁰. Indeed, Mid-domains and AID segments of SPT16, but also SSRP1, were important for binding to centromeric histones (Fig. S9BS9E). These data suggest that FACT is not able to chaperone CENP-A/H4 and CCAN simultaneously.

Discussion

FACT has established roles in transcription, replication, and DNA repair. On the other hand, the functional significance of its enrichment at human kinetochores remains unclear. We identified novel interactions of FACT with CCAN subunits CENP-C and CENP-OPQUR in addition to the previously reported binding to the HFD-containing complex CENP-TW (63). These interactions act cooperatively to form a stable 18-subunit complex. During mitosis, FACT localizes to the kinetochore in a CCAN-dependent manner, and the CENP-HIKM and CENP-TW complexes are especially important for this localization. Furthermore, FACT depletion results in a significant reduction of CENP-TW at the kinetochore during mitosis. The direct and

specific interaction of FACT with the kinetochore suggests a role linked to kinetochore assembly and the regulation of kinetochore interactions with centromeric chromatin. The exact function of kinetochore FACT will have to be elucidated, but our study paves the way for more detailed functional analysisanalyses.

In vitro, FACT was displaced from the CCAN by DNA, suggesting that CCAN is not stably anchored to chromatin while FACT is bound. This is unexpected, as FACT localizes to the kinetochore in a CCAN-dependent manner during mitosis (Fig. 1I-J), a time when we expect a tight connection between kinetochores and centromeric DNA. Our results may suggest that a subset of CCANs complexes is engaged with FACT rather than with DNA. There is only limited information on the mechanism and timing of recruitment of CCAN to the centromere during or after DNA replication. In line with its chaperone activity, FACT may stabilize the CCAN or CCAN subunits in solution and help in the deposition and assembly of the CCAN at the centromere. We observed that while individual interaction partners had a lower affinity for FACT than the entire CCAN, CENP-C²⁻⁵⁴⁵HIKMLNTW^{HPD} was the strongest binder, while CENP-TW and -OPQUR were moderately anti-cooperative and competing for FACT binding (Fig. 2A-E, H).

Our observations also indicate that individual interactions of FACT and CCAN ultimately cooperate to enhance overall binding. For instance, we suspect that the interaction of FACT with CENP-OPQUR may undergo a rearrangement inside the CCAN in comparison to the isolated FACT/CENP-OPQUR complex. Thus, FACT may preferentially bind to CCAN complexes that are not fully assembled or properly incorporated into centromeric chromatin. This possibility may also partly explain some discrepancies between the results of binding assays *in vitro* and the analysis of FACT localization after displacement of CCAN subunits *in vivo*. For instance, the CENP-HIKM complex appeared to have a disproportionate effect on FACT localization if gauged against the apparently low binding affinity for FACT *in vitro*. Given the position of CENP-HIKM in the CCAN hierarchy, which is upstream compared to other sub-complexes, it is reasonable to assume that its depletion would lead to a more significant effect on FACT recruitment to the kinetochore. We surmise that depletion of CENP-HIKM may indirectly affect the interaction of CENP-TW with DNA, causing FACT displacement indirectly.

Upon re-entry into interphase, FACT redistributeslocalizes to the entire chromosome, more pronouncedly around nucleoli (66, 67) ~~(Fig. 1F)~~. (Fig. 1F). We used a pre-extraction strategy to visualize FACT at the kinetochore during mitosis. This treatment removes FACT from the rest of the chromatin, and suggests indirectly that FACT is more stably bound to chromatin in

interphase. Our inability to visualize FACT at kinetochores outside of mitosis does not necessarily imply depletion of FACT from these structures, as visualizing kinetochore FACT by immunofluorescence during interphase against a more pronounced chromosome signal is technically challenging. As CCAN localizes to the centromere throughout the cell cycle (6), it will be important to establish if FACT acts there outside of mitosis. Chromatin is considered transcriptionally silent in mitosis (104), but it has been suggested that centromeric transcription is also active during mitosis (70, 71, 73, 74). Thus, it is possible that the localization of FACT at the kinetochore coincides with active centromeric transcription in mitosis and interphase.

CENP-TW localization was reduced by acute depletion of FACT in mitosis. Remarkably, other CCAN subcomplexes were retained at the centromere, aside from a minor decrease in CENP-U localization. As CENP-TWSX is integrated into the specialized centromeric chromatin (24) For instance, FACT may prevent loss of CCAN, FACT may stabilize CENP-TW by preventing its loss as RNA polymerase II passes through centromeric chromatin. This would be reminiscent of FACT's known role in preventing histone loss during transcription (36, 105). Whether FACT is essential for the stability of the whole CCAN over longer period of times is currently unclear.

Due to FACT's role in multiple chromatin related mechanisms, studying its specific role at the kinetochore is challenging. ~~As RNAi based depletion of FACT is slow and likely causative of pleiotropic effects, targeted acute degradation of FACT specifically in mitosis~~ In the future, it will be ~~required to overcome this challenge and focus on its mitosis specific role. Another option would be~~ essential to identify separation-of-function ~~mutant~~ mutants to target specific functions of FACT. The AID of SPT16 as well as phosphorylation by CK2 are important for other functions in addition to mediating the interaction with CCAN and are therefore not appropriate targets for mutations. Investigation of potential CK2 sites on FACT may ultimately identify sites that are solely important for CCAN binding. Finally, structural information on a FACT/CCAN complex could facilitate the identification of specific interaction interfaces. So far, our efforts to obtain high-resolution structures of the CCAN/FACT complex have been thwarted by lack of order of the resulting complexes.

The phosphorylation of FACT by CK2 is indispensable for FACT/CCAN complex formation. *In vitro*, binding of DNA to CCAN leads to the dissociation of FACT, while FACT preferentially binds to centromeric histones, which share binding sites with CCAN. Collectively, these data suggest that FACT impacts the kinetochore directly rather than sharing the same function at the centromere as in other parts of chromatin. FACT is predicted to possess up to ~~thirty~~ twenty or more CK2 phosphorylation sites, especially in the AID sequences (Fig. 5D). Currently, it remains uncertain which of these sites are crucial for the binding to CCAN. Nevertheless,

phosphorylation of FACT is also important for other functions (38, 85–87, 106) and it is not clear whether the different CK2 sites on FACT are functionally related. Interestingly, phosphorylation of FACT reduces its DNA binding activity (86, 106). It is possible that FACT changes its exact localization from DNA to histones or the CCAN depending on its phosphorylation state. Alternatively, different pools of FACT may accomplish different functions simultaneously. For instance, one pool may bind CCAN, while another may bind CENP-A/H4 or other histones during transcription and replication. In summary, we provided a characterization of the FACT/CCAN interaction *in vitro* and *in vivo*, and set the basis for future work aiming to dissect this interaction.

Materials and Methods

Plasmids

Plasmids for the expression of CENP-C²⁻⁵⁴⁵HIKM, CENP-C²⁻⁵⁴⁵, CENP-C¹⁸⁹⁻⁴⁰⁰, MBP^{CENP-C⁷²¹⁻⁷²¹}C, CENP-HIKM, CENP-LN, CENP-OPQR, CENP-OPQ^{68-C}U^{115-C}R CENP-TWSX, CENP-TW, CENP-T^{458-C}W, CENP-SX, MBP^{CENP-T/W}, CENP-A/H4, H2A/H2B, CDK1, Cyclin B, CKS1, PLK1, Aurora B⁴⁵⁻³⁴⁴/INCENP⁸³⁵⁻⁹⁰³ and for the production of DNA sequences were generated as previously described (13, 14, 20, 26, 64, 65, 79, 81, 107, 108). Plasmid expressing human CK2 α ¹⁻³³⁵ and CK2 β ¹⁻¹⁹³ were a kind gift of K. Niefind (University of Cologne, Germany). SPT16 and SSRP1 with a N-terminal His-tag and a TEV cleavage site were cloned into a pFL-derived MultiBac vector (109). Sequences of MBP^{TEV}SPT16, His^{TEV}SSRP1, His^{TEV}SPT16⁶⁴⁴⁻⁹⁸⁸ (Mid-AID), His^{TEV}SPT16⁶⁴⁴⁻⁹³⁰ (Mid), His^{TEV}SSRP1¹⁹⁷⁻⁵¹⁴ (Mid-AID), His^{TEV}SSRP1¹⁹⁷⁻⁴³³ (Mid), His^{TEV}SPT16⁵⁰⁸⁻⁹⁸⁸, MBP^{TEV}SPT16⁵⁰⁸⁻⁹⁸⁸, SSRP1¹⁻⁵¹⁴, His^{TEV}SSRP1¹⁻⁵¹⁴, His-MBP^{CENP-C^{EGFP}} and GST^{CENP-U} were inserted into pLIB vectors. These were used to combine MBP^{TEV}SPT16+His^{TEV}SSRP1 (MBP^{FACT}), His^{TEV}SPT16⁵⁰⁸⁻⁹⁸⁸+SSRP1¹⁻⁵¹⁴ (FACT^{trunc}), MBP^{TEV}SPT16⁵⁰⁸⁻⁹⁸⁸+His^{TEV}SSRP1¹⁻⁵¹⁴ (MBP^{FACT^{trunc}}), GST^{CENP-U} with previously described CENP-O/P/Q/R and His^{TEV}CENP-Q^{68-C} and CENP-U^{115-C} with CENP-O/P/R (64) in pBIG1a vectors for baculovirus based multigene-expression (110). His^{PreSc}SPT16¹⁻⁵⁴⁴ (Peptidase-like domain), His^{PreSc}SSRP1⁵³¹⁻⁶²⁷ (HMG domain), His^{PreSc}SSRP1^{513-C}, MBP^{CENP-C²⁻⁴⁰⁰-His}, MBP^{CENP-C⁴⁰¹⁻⁶⁰⁰-His}, MBP^{CENP-C⁴⁰¹⁻⁵⁴⁵-His}, MBP^{CENP-C⁵⁴⁶⁻⁶⁰⁰-His}, MBP^{CENP-C⁶⁰¹⁻⁷²⁰-His}, MBP^{CENP-C⁷²¹⁻⁷⁵⁹-His}, His-MBP^{TEV}CENP-C^{760-C}, MBP^{TEV}HJURP¹⁻⁸⁰ and His^{PreSc}CENP-A co-expressed with H4 and MBP^{TEV}HJURP¹⁻⁸⁰ were cloned into a pETDuet vector using Gibson cloning (111). For mammalian expression of N-terminally tagged EGFP-SSRP1, SSRP1 sequence was obtained by PCR and subcloned in-frame with the sequence encoding the EGFP tag in pCDNA5/FRT/TO-EGFP-IRES, a previously modified version (112) of the pCDNA5/FRT/TO vector (Invitrogen).

Purification of DNA fragments

Generation of the Widom 601 145-bp (ATCAGAATCCCGGTGCCGAGGCCGCTCAATTG GTCGTAGACAGCTCTAGCACCGCTTAAACGCACGTACGCGCTGTCCCCGCGTTTT AACCGCCAAGGGGATTACTCCCTAGTCTCCAGGCACGTGTCAGATATATACATCGA T) and the CEN1 (centromere 1)-like 75-bp (ATCCGTGGTAGAATAGGAAAT ATCTTCCTATAGAACTAGACAGAATGATTCTCAGAACTCCTTTGTGATGGAT), 165-bp (GTGGTAGAATAGGAAATATCTTCCTATAGAACTAGACAGAATGATTCT

CAGAAACTCCTTTGTGATGTGTGCGTTCAACTCACAGAGTTTAACCTTTCTTTTCAT
AGAGCAGTTAGGAAACACTCTGTTTGTAAATGTCTGCAAGTGGATATTCAGACGCC
TTG, 183-bp (AGGCCTTCGTTGGAAACGGGATTTCTTCATATTCTGCTAGA
CAGAAGAATTCTCAGTAACCTTCTGTTGTGTGTAATTCAACTCACAGAGTTGAAC
GATCCTTTACACAGAGCAGACTTGAAACACTCTTTTTGTGGAATTTGCAGGCCTAG
ATTCAGCCGCTTTGAGGTCAATCACCCC) and 199-bp (ATCGCCCTTGAG
GCCITCGTTGGAAACGGGATTTCTTCATATTCTGCTAGACAGAAGAATTCTCAGTA
ACTTCTTGTGTTGTGTGTAATTCAACTCACAGAGTTGAACGATCCTTTACACAGAGC
AGACTTGAAACACTCTTTTTGTGGAATTTGCAGGCCTAGATTTAGCCGCTTTGAG
GTCAATCACCCCGTGGAT) was performed as previously described (20, 26).

Reconstitution of nucleosomes

CENP-A nucleosome core particles on 145-bp 601 or CEN1-like 183-bp DNA were produced as previously reported (403113).

To generate CENP-A nucleosomes on 348 bp DNA, 165-bp DNA was ligated to the front of 183-bp DNA on pre-reconstituted CENP-A nucleosome. The two species were mixed in equimolar amounts based on the concentration of the DNA fragments in a buffer consisting of 10 mM Tris pH 7.4, 100 mM NaCl and 1 mM EDTA. Two times the amount of MBP-T4 DNA ligase^{His} (produced in house) relative to the DNA fragment was added with 10X T4 DNA Ligase Buffer and the reaction was incubated for approximately 16 h at 4 °C. The reaction was passed through two consecutive 1 ml HisTrap FF columns (Cytiva), equilibrated in 10 mM Tris pH 7.4 and 100 mM NaCl to remove His-tagged T4 DNA ligase. The flow-through and wash was collected and EDTA was added to a final concentration of 2 mM.

Field Code Changed

Protein expression and purification

CENP-C²⁻⁵⁴⁵HIKM, CENP-C²⁻⁵⁴⁵, CENP-C¹⁸⁹⁻⁴⁰⁰, CENP-HIKM, CENP-LN, CENP-OPQR, CENP-TWSX, CENP-TW, CENP-T^{458-C}W, CENP-SX, MBP-CENP-T/W, CDK1/Cyclin B/CKS1, PLK1 and Aurora B⁴⁵⁻³⁴⁴/INCENP⁸³⁵⁻⁹⁰³ were expressed and purified as previously reported (13, 14, 20, 26, 64, 65, 79, 81, 107, 108). GST-CENP-OPQR and CENP-OPQ^{68-C}U^{115-C}R were purified identically to the wild type (64) by using either glutathione affinity or Nickel affinity as a first step.

FACT, MBP-FACT, FACT^{trunc}, MBP-FACT^{trunc}, SPT16^{Mid-AID}, Mid domain, SSRP1^{Mid-AID} and Mid domain were expressed by infecting Tnao38 cells with a virus:culture ratio of 1:20 and

incubating the cells at 27 °C for 72 h. For the expression of ^{MBP}CENP-C^{EGFP} a virus:culture ratio of 1:40 was used. SPT16 peptidase-like domain, SSRP1^{HMG}, SSRP1⁵¹³⁻⁷⁰⁹, ^{MBP}HJURP^{1-80-His} and ^{His}CENP-A/H4/^{MBP}HJURP¹⁻⁸⁰ were expressed in *E. coli* BL21(DE3)-Codon-plus-RIL cells by growing transformed cells to an OD₆₀₀ of 0.7 in TB medium supplemented with ampicillin and chloramphenicol at 25 °C. Expression was induced by adding 0.1 mM IPTG and cells were cultured for 16 hours at 18 °C.

All purification steps were performed at 4 °C or samples were kept on ice. If not otherwise indicated, cells were resuspended in lysis buffer supplemented with Protease-Inhibitor Mix HP Plus (Serva), 1 mM PMSF and 10 µg/ml DNaseI and lysed by sonication. The lysate was subsequently clarified by centrifugation for 45 min at 10,000 xg at 4 °C and filtration. After the final purification step, proteins of interest (POI) were concentrated, flash-frozen in liquid nitrogen and stored at -72 or -80 °C.

Cells expressing FACT, ^{MBP}FACT, FACT^{trunc}, ^{MBP}FACT^{trunc}, SPT16^{Mid-AID}, Mid domain, SSRP1^{Mid-AID}, Mid domain and SPT16 peptidase-like domain were resuspended in a buffer containing 20 mM Tris-HCl pH 8.0, 300 mM NaCl, 5% glycerol and 1 mM TCEP (Buffer A). The lysate was applied to a 5 ml HisTrap FF (Cytiva). The column was first extensively washed with Buffer A and then with Buffer A including 30 mM imidazole. Full-length FACT was eluted by a linear gradient to 400 mM imidazole, others were eluted in Buffer A with 250 mM imidazole. The fractions containing protein were pooled and diluted 1:4 with Buffer A containing 150 mM NaCl. This was loaded on two sequential 1 ml HiTrap Q HP (Cytiva) anion exchange columns. The columns were washed and protein was eluted by a gradient to 1 M NaCl. Peak fractions were analyzed in SDS-PAGE and fractions containing protein or a stoichiometric complex were pooled and concentrated. To obtain dephosphorylated protein FACT and ^{MBP}FACT were treated with λ-phosphatase (produced in house) at 4 °C in presence of 1 mM MnCl₂ for approximately 16 h. Full-length FACT, ^{MBP}FACT, FACT^{trunc} and ^{MBP}FACT^{trunc} were finally applied to a HiLoad 16/600 Superose 6 pg column, the others were purified on a HiLoad 16/600 Superdex 200 pg (Cytiva).

Expressions of SSRP1^{HMG} and SSRP1⁵¹³⁻⁷⁰⁹ were resuspended in a buffer composed of 20 mM HEPES pH 6.8, 300 mM NaCl, 5% glycerol, 10 mM imidazole and 1 mM TCEP. Nickel affinity purification was performed as explained above. Protein was diluted 1:4 in the same buffer with 100 mM NaCl and loaded on a 5 ml HiTrap Heparin HP column (Cytiva) and protein was eluted in a gradient to 1 M NaCl. Fractions containing the protein were pooled and concentrated to be applied to a HiLoad 16/600 Superdex 75 pg (Cytiva).

A pellet of ^{MBP}CENP-C^{EGFP} expressing Tnap38 cells was resuspended in approximately 10 volumes of TALON buffer (50 mM Hepes pH 7.0, 500 mM NaCl, 5 mM MgCl₂, 5% glycerol, 5 mM imidazole, 2 mM TCEP) supplemented with 2 mM PMSF and DNaseI. Affinity purification was performed on a 5 ml HisTALON Cartridge pre-packed with TALON Superflow Resin (Cytiva) and the column was washed with 10 CV buffer. The protein was eluted in TALON buffer A with 200 mM imidazole and subsequently diluted to 300 mM NaCl in Heparin buffer (20 mM Hepes pH 7.0, 5% glycerol, 2 mM TCEP). A 5 ml HiTrap Heparin HP column (Cytiva) was equilibrated in Heparin buffer including 300 mM NaCl and the diluted protein was bound to it. The column was washed with Heparin buffer with 300 mM NaCl and the protein was eluted in a linear gradient to 1 M NaCl in 150 ml. Peak fractions containing the POI were concentrated and subjected to SEC on a HiLoad 16/600 Superose 6 pg (Cytiva) in SEC buffer (20 mM Hepes pH 7.0, 500 mM NaCl, 5% glycerol, 1 mM TCEP).

^{MBP}CENP-C²⁻⁴⁰⁰ and ^{MBP}CENP-C^{721-C} were purified in a buffer consisting of 50 mM Hepes, pH 7.5, 500 mM NaCl, 10% glycerol and 1 mM TCEP. Proteins were purified on a 5 ml HisTrap FF (Cytiva) as indicated above and eluted in 250 mM imidazole. The eluate was diluted 3 times in Heparin buffer (20 mM Hepes pH 7.5, 150 mM NaCl, 5% glycerol, 1 mM TCEP), bound to a 5 ml HiTrap Heparin HP column (Cytiva) and eluted by a linear gradient to 1 M NaCl. Subsequently to SDS-PAGE, relevant fractions of ^{MBP}CENP-C²⁻⁴⁰⁰ were further purified by SEC on a HiLoad 16/600 Superdex 200 pg, while ^{MBP}CENP-C^{721-C}, which dimerizes, was applied to a HiLoad 16/600 Superose 6 pg (Cytiva).

^{MBP}CENP-C⁴⁰¹⁻⁶⁰⁰, ^{MBP}CENP-C⁴⁰¹⁻⁵⁴⁵, ^{MBP}CENP-C⁵⁴⁶⁻⁶⁰⁰, ^{MBP}CENP-C⁷²¹⁻⁷⁵⁹ and ^{MBP}CENP-C^{760-C} were obtained by Nickel affinity purification in 20 mM Hepes pH 7.5, 500 mM NaCl, 10% glycerol, 10 mM imidazole and 1 mM TCEP. Proteins were eluted in 250 mM imidazole, concentrated and applied to a HiLoad 16/600 Superdex 200 pg (Cytiva) in 20 mM Hepes pH 7.5, 300 mM NaCl, 5% glycerol and 1 mM TCEP.

^{MBP}CENP-C⁶⁰¹⁻⁷²⁰ was purified on a 5 ml HisTrap FF (Cytiva) in 20 mM Tris pH 8.0, 300 mM NaCl, 5% glycerol and 1 mM TCEP and eluted in 250 mM imidazole. The eluate was diluted 5 times in 20 mM Tris pH 8.0, 200 mM NaCl, 5% glycerol and 1 mM TCEP and applied to two consecutive 1 ml HiTrap Q HP (Cytiva) columns. The POI was collected in the flow-through, while DNA bound to the column. The flow-through was concentrated and purified on a HiLoad 16/600 Superdex 200 pg (Cytiva).

Cells expressing ^{His}CENP-A/H4/^{MBP}HJURP¹⁻⁸⁰ were resuspended in a buffer consisting of 20 mM Tris pH 8.0, 1 M NaCl and 1 mM TCEP. The lysate was applied to a 5 ml HisTrap FF

(Cytiva) column, which was first washed with buffer and subsequently with buffer including 10 mM imidazole. The protein was eluted in 250 mM imidazole and diluted 1:4 with IEX buffer (20 mM HEPES pH 6.8, 600 mM NaCl, 1 mM TCEP) and loaded on a 1 ml HiTrap SP HP (Cytiva). The protein was eluted by a linear gradient to 2 M NaCl and afterwards concentrated and purified on a HiLoad 16/600 Superdex 200 pg (Cytiva) in the initial buffer including 1 M NaCl. To obtain untagged protein, the eluate of Nickel affinity purification was treated with PreScission and TEV protease for approximately 16 h at 4 °C while it was dialyzed to a buffer containing 750 mM NaCl. Purification on a 1 ml HiTrap SP HP and SEC on a HiLoad 16/600 Superdex 200 pg (Cytiva) was performed as explained above.

^{MBP}HJURP¹⁻⁸⁰ expressing *E. coli* were resuspended in a buffer consisting of 20 mM Hepes pH 7.5, 300 mM NaCl, 5% glycerol and 1 mM TCEP. The cells were lysed by high pressure in a microfluidizer and afterwards clarified by centrifugation. Nickel affinity purification was performed as explained above and the POI was eluted in a linear gradient to 400 mM imidazole. Pure fractions were pooled and concentrated and applied to HiLoad 16/600 Superdex 200 pg (Cytiva) in a buffer with 2.5% glycerol.

Generation of the CK2 holoenzyme was loosely based on previous literature (114, 115). The CK2 α ¹⁻³³⁵ expression plasmid carried a resistance against Kanamycin, while the CK2 β ¹⁻¹⁹³ expression plasmid was resistant against Ampicillin. The expression was performed in *E. coli* BL21(DE3)-Codon-plus-RIL in TB medium with the specific antibiotic and additional Chloramphenicol. The cells were grown to an OD₆₀₀ of 0.6 at 37 °C. The expressions were induced by the addition of 0.5 mM IPTG and incubated at 30 °C for 4 h. The two isolated cultures were harvested and mixed together. The cells were resuspended in lysis buffer (50 mM Tris pH 8.5, 500 mM NaCl, 30 mM imidazole). The centrifuged lysate was incubated at 4 °C for 16 h to ensure the efficient formation of the holoenzyme. The next day, the lysate was filtered and purified on a 5 ml HisTrap FF (Cytiva). The column was washed after the application of the lysate and protein was eluted in lysis buffer with 250 mM imidazole. The eluate was concentrated and applied to SEC on a HiLoad 16/600 Superdex 200 pg (Cytiva) in SEC buffer (25 mM Tris pH 8.5, 500 mM NaCl).

Analytical SEC

Proteins were mixed in SEC buffer (20 mM HEPES pH 6.8, 300 mM NaCl, 2.5% glycerol, 1 mM TCEP), diluted to 5 μ M in 55 μ l. Complexes were incubated for at least 1 h at 4 °C. Samples were centrifuged and 5 μ l of sample were taken for SDS-PAGE analysis prior to SEC

on a Superose 6 Increase 5/150 GL (Cytiva, Marlborough, US-MA) on an ÄKTA micro system (Cytiva). All samples were eluted under isocratic conditions at 4 °C in SEC buffer at a flow rate of 0.2 ml/min. Fractions of 100 µl were collected and analyzed by SDS-PAGE and Coomassie blue staining.

Pull-down assays

The proteins were mixed at 3 µM in binding buffer (20 mM HEPES pH 6.8, 300 mM NaCl, 2.5% glycerol, 1 mM TCEP, 0.01% Tween) to a total volume of 50 µl. The samples were incubated at 4 °C for at least 1 h. Afterwards, they were centrifuged for 15 min at 16,000 xg prior to mixing them with 25 µl of amylose beads (New England Biolabs) or glutathione beads (Serva). Then, 20 µl were taken as an input sample for SDS-PAGE. The rest of the solution was incubated at 4 °C for an additional hour on an orbital shaker set to 1000 rpm (IKA VXR basic Vibrax). The samples were centrifuged, at 800 xg for 3 min at 4 °C. The unbound protein in the supernatant was removed, and the beads were washed 4 times with 500 µl of binding buffer. At the last step, the maximum amount of buffer was carefully removed and beads were taken up in 20 µl of SDS-PAGE sample loading buffer. The samples were boiled for 5 min at 96 °C and analyzed by SDS-PAGE and Coomassie staining.

Gel densitometry

SDS-PAGES were imaged in a ChemiDoc MP imaging system (BioRAD). Subsequent densitometric analysis of protein bands was performed using ImageLab Software (BioRAD). The band intensity of Coomassie-stained proteins was determined to quantify binding in pull-down assays. The band intensity was normalized to the bait to account for differential loading. Differential molecular weight and staining by Coomassie were not accounted for. Therefore, different subunits cannot be compared.

To determine subunit stoichiometry, SDS-PAGE gels were supplemented with 2,2,2-trichloroethanol (TCE) and proteins were visualized by fluorescence upon UV irradiation. The band intensity was normalized to the number of tryptophan residues to establish the relative quantity of proteins (116, 117).

***In vitro* phosphorylation**

Proteins were diluted to 2.5 µM in 20 mM HEPES pH 6.8, 300 mM NaCl, 2.5% glycerol and 1 mM TCEP. Otherwise, concentrations and buffer were used according to analytical SEC or pull-down assays. 1 mM Sodium orthovanadate and 5 µM okadaic acid were added to inhibit residual

Lambda-phosphatase activity and 10 mM MgCl₂ and 2 mM ATP were added for kinase activity. The samples were incubated at 25 °C for 90 minutes with CK2 at a 1:20 ratio. Pro-Q Diamond phosphoprotein stain (Invitrogen, Carlsbad, California, United States) was performed according to the manufacturer's manual.

Identification of phosphorylation sites by mass spectrometry

Liquid chromatography coupled to mass spectrometry was used to analyze phosphorylation sites on FACT after phosphorylation *in vitro* by CK2 as described above. We compared phosphorylated FACT with dephosphorylated FACT and untreated FACT, each expressed in insect cells. Samples were reduced, alkylated and digested with LysC/Trypsin and prepared for mass spectrometry as previously described (118). The obtained peptides were subjected to a desalting cartridge in water with 0.1% formic acid for 5 min. Subsequently, they were separated on an U3000 nanoHPLC system (Thermo Fisher Scientific) using a gradient from 5-30% acetonitrile in 9 µl with 0.1% formic acid on a PepMap C18 nanoHPLC column (Thermo Fisher Scientific). The samples were directly introduced via a nano-electrospray source into a quadrupole Orbitrap mass spectrometer (Q-Exactive Plus, Thermo Fisher Scientific). The Q-Exactive was operated in a data-dependent mode acquiring one survey scan followed by up to ten MS/MS scans. To identify phospho-sites, the resulting raw files were processed with MaxQuant (version 2.2.0.0), searching against the sequences of SPT16 and SSRP1 and a contamination database including N-terminal acetylation, oxidation (M) and phosphorylation (STY) as variable modifications and carbamidomethylation (C) as fixed modification. A false discovery rate cut off of 1% was applied at the peptide and protein levels and on the site decoy fraction (119), phosphorylated *in vitro* by CK2 as described above. Insect cell-expressed and dephosphorylated FACT were subjected to the analysis as controls. Samples were reduced, alkylated and digested with LysC/Trypsin and prepared for mass spectrometry as previously described.

In silico prediction of phosphorylation sites

NetPhos – 3.1 with a score higher than 0.5 and Scansite 4.0 at Medium setting searches were performed to predict CK2 phosphorylation sites on SPT16 and SSRP1 (97–99). Only sites that were predicted by both algorithms are displayed in Fig. 5D.

Cell culture

All Cells were grown at 37 °C in the presence of 5% CO₂. Parental Flp-In T-Rex DLD-1 or TTR1 and DLD-1^{YFP-AID} cells were a kind gift from D. C. Cleveland (University of California, San Diego, USA). DLD-1 cells were grown in Dulbecco's modified Eagle's medium (DMEM; PAN Biotech) and parental Flp-In T-Rex hTERT RPE-1 cells were a kind gift from J. Pines (Institute of Cancer Research: London, London, GB) and hTERT RPE-1 cells expressing endogenously tagged CENP-U-FKBP^{F36V} were grown in DMEM F12 media (PAN Biotech); both supplemented with 10% tetracycline-free fetal bovine serum (Sigma) and l-glutamine (PAN Biotech). K562-SSRP1-dtag cells were a kind gift from P. Cramer, M. Oudelaar and K. Žumer (Max Planck Institute for Multidisciplinary Sciences, Göttingen, Germany) and were grown in RPMI (Gibco) media supplemented with 1x GlutaMAX (Gibco).

Cell ~~synchronisation~~synchronization and drug treatments

Degradation of the endogenous CENP-C^{YFP-AID} was achieved through the addition of 500 µM Indole-Acetic-Acid (IAA, Sigma-Aldrich) and degradation of endogenous CENP-U-FKBP^{F36V} was achieved by addition of 500 nM dTAG^V-1 (Tocris). Cells were synchronized using S-trityl-L-cysteine (STLC) at 5 µM for 16 hours or Nocodazole 3.3 µM for 4 hours. Mitotic degradation of FACT was achieved by treating STLC arrested K562 cells with 500 nM dTAG^V-1 (Tocris).

RNA interference

Depletion of endogenous proteins was achieved through transfection of small interfering RNA (siRNA) with RNAiMAX (Invitrogen) according to manufacturer's instructions. Following siRNAs treatments were performed in this study: 30 nM of each oligo, siCENP-T (Dharmacon, 5'-GACGAUAGCCAGAGGGCGU-3', 5'-AAGUAGAGCCCUUACACGA-3') for 60 hours (81), siCENP-H (Sigma, 5'-CUAGUGUGCUCUAUGGAUAA-3') (65), siCENP-I (Sigma, 5'-AAGCAACTCGAAGAACATCTC-3') (120), siCENP-K (Dharmacon, On-target plus smartpool-XX) (80), siCENP-M (Sigma, 5'-ACAAAAGGUCUGUGGCUAA-3', 5'-UUAAGCAGCUGGCGUGUUA-3', 5'-GUGCUGACUCCAUAACAU-3') (~~7481~~) for 72 hours,

Field Code Changed

Transient transfection of Flp-In T-Rex hTERT RPE-1

1 µg of pcDNA5-EGFP-SSRP1-IRES was transiently transfected in Flp-In T-Rex hTERT-RPE-1 cells using FuGENE® HD Transfection Reagent (Promega) following the standard manufacturer's protocol. Expression was induced with 200 ng/ml of doxycycline 6 hours post transfection. Cells were arrested with RO3306 for 16 hours and released for 1 hour to enrich mitotic population for immunofluorescence analysis.

Generation of stable cell lines

Stable Flp-In T-Rex DLD-1 osTIR1 cell lines were generated using FRT/Flp recombination. CENP-Uⁿ and 115-C constructs were cloned into pcDNA5 plasmids (79) and were co-transfected with pOG44 (Invitrogen), encoding the Flp recombinase, into DLD-1 cells using XtremeGENE (Roche) according to the manufacturer's instructions. After selection for 2 weeks in DMEM supplemented with hygromycin B (250 µg/ml; Carl Roth) and blasticidin (4 µg/ml; Thermo Fisher Scientific), single-cell colonies were isolated, expanded and the expression of the transgenes was checked by immunofluorescence microscopy and immunoblotting analysis. The gene expression was induced by the addition of 0.3 µg/ml doxycycline (Sigma-Aldrich).

hTERT RPE-1 CENP-U-FKBP^{F36V}-NeoR knock-in cell line was generated via electroporation of gRNA-Cas9 ribonucleoproteins (RNPs) as previously described (121). Briefly, 3×10^5 cells were electroporated using P3 Primary Cell Nucleofector[®] 4D Kit and Nucleofector 4D system (Lonza) with 400 ng donor DNA, 120 pmol of Cas9, 1.5 µl Alt-R[™]-CRISPR-Cas9 crRNA (TTAGAGAAGCTCCTTGACCA (GGG)), 1.5 µl Alt-R[®] CRISPR-Cas9 tracrRNA (100 µM, IDT) and 1.2 µl of Alt-R[®] Cas9 Electroporation enhancer (100 µM, IDT). After electroporation, cells were seeded into DMEM F12 media supplemented with 1 µM NU7441 for 48 h before selection with 400 µg/ml of G418 for 2 weeks. The pool of cells was subjected to single cell dilution to obtain monoclonal lines. Genomic DNA was isolated from clones and in-frame knock-in was confirmed by Sanger sequencing using primers spanning the locus of insertion (Primer fwd: 5'-CATGTGTGTGGTAGTCACAGCATG-3', Primer rev: 5'-TCTGGGATAATGGCATTGATGATGC-3').

Immunofluorescence

Cells were grown on coverslips pre-coated with poly-L-lysine (Sigma-Aldrich). Cells were permeabilized with 0.5% Triton X-100 solution in PHEM (Pipes, HEPES, EGTA, MgCl₂) buffer supplemented with 100 nM microcystin for 5 minutes before fixation with 4% paraformaldehyde

(PFA) in PHEM for 15 minutes. After blocking with 5% boiled goat serum (BGS) in PHEM buffer for 30 min, cells were incubated for 2 hours at room temperature with the following primary antibodies; CENP-C (Guinea pig, MBL, #PD030, 1:1,000), CENP-HK, CENP-TW (Rabbit, made in-house, 1:800), SSRP1 (Mouse, BioLegend Europe #609702, 1:200), CENP-A (Mouse, GeneTex GTX13939, 1:500), CENP-O (Gift from McAinsh Lab, 1:200), PLK1 (Mouse, Abcam #ab17057, 1:500), CENP-R (Rabbit, Proteintech #107431-AP, 1:200), CREST (Anticentromere Anti immune serum) (Human, Antibodies Inc. (via antibodies-online), #15-234, 1:200) diluted in 2.5% BGS-PHEM with exception of CENP-U (Sigma (Atlas, HPA022048, 1:100) which was diluted in 5% BGS-PHEM and incubated at 37 °C for 3 hours.

Subsequently, cells were incubated for 1 hour at room temperature with the following secondary antibodies: (all 1:200 in 2.5% BGS-PHEM): Goat anti-mouse Alexa Fluor 488 (Invitrogen A A11001), goat anti-mouse Rhodamine Red (Jackson Immuno Research 115-295-003), donkey anti-rabbit Alexa Fluor 488 (Invitrogen A21206), donkey anti-rabbit Rhodamine Red (Jackson Immuno Research 711-295-152), goat anti-human Alexa Fluor 647 (Jackson Immuno Research 109-603-003), goat anti-guinea pig Alexa Fluor 647 (Invitrogen A-21450). All washing steps were performed with PHEM supplemented with 0.1% Triton-X-100 (PHEM-T) buffer. DNA was stained with 0.5 µg/ml DAPI (Serva) and Mowiol (Calbiochem) was used as mounting media.

Cells were imaged at room temperature using a spinning disk confocal device on the 3i Marianas system equipped with an Axio Observer Z1 microscope (Zeiss, Jena, Germany), a CSU-X1 confocal scanner unit (Yokogawa Electric Corporation, Tokyo, Japan), 100x / 1.4NA oil objectives (Zeiss), and Orca Flash 4.0 V2 sCMOS Camera (Hamamatsu) and Orca Fusion BT sCMOS camera (Hamamatsu). Images were acquired as z sections at 0.27 µm using Slidebook Software 2023.3 and 2024.2 (Intelligent Imaging Innovations, Denver, USA). Images were converted into maximum intensity projections and exported as 16-bit TIFF files. Alternatively, cells were imaged using a UPLSAPO 100x / 1.4NA oil objective on a DeltaVision deconvolution microscope (GE Healthcare, UK) equipped with an IX71 inverted microscope (Olympus, Japan), and a pco.edge sCMOS camera (PCO-TECH Inc., USA). Images were acquired as z sections at 0.2 µm

Quantification of kinetochore signals were performed on 16-bit maximum intensity projections using a semi-automatic quantification MACRO in FIJI (122) ~~with background subtraction, with background subtraction. When CREST was used as kinetochore reference, Otsu-thresholding of the DAPI signal was applied for generating a KT's segmentation mask per cell. Integrated intensities per cell were calculated for each fluorescence channel based on the DAPI reference~~

mask. Background-corrected mean fluorescence intensities were multiplied by the ROI area to obtain the total fluorescence signal for proteins of interest and reference in each cell. Total fluorescence signal for each channel of interests was plotted using GraphPad Prism 9. When CENP-C was used as kinetochore reference, the mask was created using CENP-C for individual kinetochore picking. Data was exported to Microsoft Excel for normalization and plotted using GraphPad Prism 9 software. Statistical analysis was performed with a non-parametric t-test comparing two unpaired groups (Mann-Whitney test). Symbols indicate n.s. $p > 0.05$, * $p \leq 0.05$, ** $p \leq 0.01$, *** $p \leq 0.001$, **** $p \leq 0.0001$. Images were assembled in Adobe Illustrator 2024.

Co-immunoprecipitation

For co-immunoprecipitation, DLD-1 cells expressing EGFP, ^{EGFP}CENP-U^{fl} and ^{EGFP}CENP-U^{115-C} were harvested by mitotic shake off and lysed using lysis buffer (75 mM HEPES pH 7.5, 150 mM KCl, 10% glycerol, 1 mM EGTA, 1.5 mM MgCl₂, 1 mM DTT, 0.075% NP-40) supplemented with 1 mM PMSF, Protease-Inhibitor Mix HP Plus (Serva), phosphatase inhibitor PhosSTOP (Sigma Aldrich), and Benzoase (EMD Millipore Corp, USA). Lysates were clarified by centrifugation at 22,000 g for 30 min at 4 °C and supernatant was collected for immunoprecipitation analysis. Lysates at a concentration of 7 mg/ml were incubated with 20 µl GFP-Trap magnetic agarose (ChromoTEK) for 3 hours at 4 °C in a total volume of 500 µl in Lysis Buffer. Beads were washed 3 times with Lysis Buffer. The dry beads were resuspended in SDS-PAGE sample loading buffer and boiled for 5 minutes at 95 °C. The samples were analyzed by SDS-PAGE and subsequent Western blotting analysis. The following antibodies were used: GFP (rabbit, made in-house, 1:3,000), SSRP1 (Mouse, BioLegend, 1:1000) PLK1 (1:1000), anti-mouse or anti-rabbit (1:10,000; Amersham, NXA931 and NA934) conjugated to horseradish peroxidase were used. After incubation with ECL Western blotting reagent (GE Healthcare), images were acquired with the ChemiDoc MP System (Bio-Rad) using Image Lab 6.0.1 software.

Supplemental material

Fig. S1 supports the biochemical part of Fig. 1 and includes the previously established reconstitution of the CCAN, additional SEC runs suggesting a 1:1 of FACT and CCAN, and additional SEC runs and summarizing table related to the pull-down in Fig. 1E. Fig. S2 supports Fig. 1 and presents the localization of GFP-SSRP1 and additional IF experiments after a 4-hour depletion of CENP-C. Fig. S3 also supports Fig.1, displaying additional IF experiments after a 24-hour depletion of CENP-C. Fig. S4 supports Fig. 2 with additional pull-downs and a co-

immunoprecipitation illustrating the interaction between CENP-C and FACT, as well as CENP-OPQUR and FACT. Fig. S5 supports Fig. 3, which contains additional information on the CENP-HIKM and CENP-T RNAi experiments. Fig. S6 supports Fig. 3 and presents additional controls for the rapid depletion of CENP-U-FKBP^{F36V}. Fig. S7 supports Fig.3 and includes controls for the rapid depletion of SSRP1^{dTAG}. Fig. S8 supports Fig. 5 with additional SEC experiments, a pull-down and phospho-stainings related to experiments in Fig. 5. Fig. S9 supports Fig. 6 and displays various SEC experiments and pull-downs. Table S1 summarizes the phospho-sites on SPT16 and SSRP1 that were phosphorylated by CK2 *in vitro* and could be detected by mass spectrometry.

Data availability

All vectors, reagents, and data described in this manuscript are available from Andrea Musacchio upon reasonable request.

References

Acknowledgments

We thank Franziska Müller and Petra Janning for help with mass spectrometry experiments, Dongqing Pan for the generation of initial constructs for the expression of recombinant FACT, Karsten Niefind for providing expression constructs for CK2, Duccio Conti for the help in initial experiments, Sabine Wohlgemuth for the purification of CENP-LN, Carolin Körner for the preparation of recombinant kinases, Lia Nitz for the production of a subset of ^{MBP}CENP-C fusion proteins, Nico Schmidt for help with microscopy experiments and data analysis, and Patrick Cramer, Kristina Žumer, A. Marieke Oudelaar, Daniele Fachinetti ~~and~~, Don C. Cleveland, and Jonathon Pines for sharing cell lines.

Funding: A.M. acknowledges funding from the Max Planck Society, the European Research Council (ERC) Synergy Grant 951430 (BIOMECHANET), the DFG's Collaborative Research Centre 1430 "Molecular Mechanisms of Cell State Transitions", and the CANTAR network under the Netzwerke-NRW program.

The authors declare that they have no competing interests.

Author contributions: (following CRediT model): Conceptualization: J.S., BM., AM.; J. Schweighofer, B. Mulay, A. Musacchio. Funding acquisition: A. Musacchio. Investigation: J.S., BM., HH., DV., MEP.; J. Schweighofer, B. Mulay, I. Hoffmann, D. Vogt, M. E. Pesenti. Visualization: J.S., BM., AM.; J. Schweighofer, B. Mulay, A. Musacchio. Supervision: AM.; A. Musacchio. Writing—

original draft: ~~JS, AM, J. Schweighofer, A. Musacchio~~. Writing—review & editing: ~~all authors~~.

~~Competing interests: The authors declare that they have no competing interests. Data and materials availability: All authors.~~

References

1. A. Musacchio, A. Desai, A Molecular View of Kinetochores Assembly and Function. *Biology* **6**, 5–5 (2017).
2. P. B. Talbert, S. Henikoff, What makes a centromere? *Experimental Cell Research* **389**, 111895–111895 (2020).
3. A. Stirpe, P. Heun, The ins and outs of CENP-A: Chromatin dynamics of the centromere-specific histone. *Seminars in Cell & Developmental Biology*, doi: 10.1016/j.semcdb.2022.04.003 (2022).
4. B. R. Brinkley, E. Stubblefield, The fine structure of the kinetochore of a mammalian cell in vitro. *Chromosoma* **19**, 28–43 (1966).
5. I. M. Cheeseman, J. S. Chappie, E. M. Wilson-Kubalek, A. Desai, The Conserved KMN Network Constitutes the Core Microtubule-Binding Site of the Kinetochore. *Cell* **127**, 983–997 (2006).
6. D. R. Foltz, L. E. T. Jansen, B. E. Black, A. O. Bailey, J. R. Yates, D. W. Cleveland, The human CENP-A centromeric nucleosome-associated complex. *Nature cell biology* **8**, 458–69 (2006).
7. M. Perpelescu, T. Fukagawa, The ABCs of CENPs. *Chromosoma* **120**, 425 (2011).
8. A. D. McAinsh, P. Meraldi, The CCAN complex: Linking centromere specification to control of kinetochore–microtubule dynamics. *Seminars in Cell & Developmental Biology* **22**, 946–952 (2011).
9. K. Song, B. Gronemeyer, W. Lu, E. Eugster, J. E. Tomkiel, Mutational analysis of the central centromere targeting domain of human centromere protein C, (CENP-C). *Experimental Cell Research* **275**, 81–91 (2002).
10. S. Trazzi, G. Perini, R. Bernardoni, M. Zoli, J. C. Reese, A. Musacchio, G. D. Valle, The C-Terminal Domain of CENP-C Displays Multiple and Critical Functions for Mammalian Centromere Formation. *PLOS ONE* **4**, e5832 (2009).
11. C. W. Carroll, M. C. C. Silva, K. M. Godek, L. E. T. Jansen, A. F. Straight, Centromere assembly requires the direct recognition of CENP-A nucleosomes by CENP-N. *Nature Cell Biology* **11**, 896–902 (2009).
12. C. W. Carroll, K. J. Milks, A. F. Straight, Dual recognition of CENP-A nucleosomes is required for centromere assembly. *Journal of Cell Biology* **189**, 1143–1155 (2010).
13. K. Klare, J. R. Weir, F. Basilico, T. Zimniak, L. Massimiliano, N. Ludwigs, F. Herzog, A. Musacchio, CENP-C is a blueprint for constitutive centromere-associated network assembly within human kinetochores. *The Journal of Cell Biology* **210**, 11–22 (2015).
14. S. Pentakota, K. Zhou, C. Smith, S. Maffini, A. Petrovic, G. P. Morgan, J. R. Weir, I. R. Vetter, A. Musacchio, K. Luger, Decoding the centromeric nucleosome through CENP-N. *eLife* **6**, 1–25 (2017).
15. H. Kato, J. Jiang, B.-R. Zhou, M. Rozendaal, H. Feng, R. Ghirlando, T. S. Xiao, A. F. Straight, Y. Bai, A Conserved Mechanism for Centromeric Nucleosome Recognition by Centromere Protein CENP-C. *Science* **340**, 1110–1113 (2013).

16. R. L. Cohen, C. W. Espelin, P. De Wulf, P. K. Sorger, S. C. Harrison, K. T. Simons, Structural and Functional Dissection of Mif2p, a Conserved DNA-binding Kinetochores Protein. *Mol Biol Cell* **19**, 4480–4491 (2008).
17. E. Screpanti, A. De Antoni, G. M. Alushin, A. Petrovic, T. Melis, E. Nogales, A. Musacchio, Direct binding of Cenp-C to the Mis12 complex joins the inner and outer kinetochores. *Current Biology* **21**, 391–398 (2011).
18. K. L. McKinley, N. Sekulic, L. Y. Guo, T. Tsinman, B. E. Black, I. M. Cheeseman, The CENP-L-N complex forms a critical node in an integrated meshwork of interactions at the centromere-kinetochores interface. *Mol Cell* **60**, 886–898 (2015).
19. H. Nagpal, T. Hori, A. Furukawa, K. Sugase, A. Osakabe, H. Kurumizaka, T. Fukagawa, Dynamic changes in CCAN organization through CENP-C during cell-cycle progression. *Mol Biol Cell* **26**, 3768–3776 (2015).
20. K. Walstein, A. Petrovic, D. Pan, B. Hagemeyer, D. Vogt, I. R. Vetter, A. Musacchio, Assembly principles and stoichiometry of a complete human kinetochores module. *Science Advances* **7**, 1–23 (2021).
21. F. Rago, K. E. Gascoigne, I. M. Cheeseman, Distinct Organization and Regulation of the Outer Kinetochores KMN Network Downstream of CENP-C and CENP-T. *Current Biology* **25**, 671–677 (2015).
22. P. J. Huis in 't Veld, S. Jeganathan, A. Petrovic, P. Singh, J. John, V. Krenn, F. Weissmann, T. Bange, A. Musacchio, Molecular basis of outer kinetochores assembly on CENP-T. *eLife*, doi: 10.7554/eLife.21007 (2016).
23. T. Hori, M. Amano, A. Suzuki, C. B. Backer, J. P. Welburn, Y. Dong, B. F. McEwen, W.-H. Shang, E. Suzuki, K. Okawa, I. M. Cheeseman, T. Fukagawa, CCAN Makes Multiple Contacts with Centromeric DNA to Provide Distinct Pathways to the Outer Kinetochores. *Cell* **135**, 1039–1052 (2008).
24. T. Nishino, K. Takeuchi, K. E. Gascoigne, A. Suzuki, T. Hori, T. Oyama, K. Morikawa, I. M. Cheeseman, T. Fukagawa, CENP-T-W-S-X forms a unique centromeric chromatin structure with a histone-like fold. *Cell* **148**, 487–501 (2012).
25. T. Fukagawa, W. C. Earnshaw, The Centromere: Chromatin Foundation for the Kinetochores Machinery. *Developmental Cell* **30**, 496–508 (2014).
26. M. E. Pesenti, T. Raisch, D. Conti, K. Walstein, I. Hoffmann, D. Vogt, D. Prumbaum, I. R. Vetter, S. Raunser, A. Musacchio, Structure of the human inner kinetochores CCAN complex and its significance for human centromere organization. *Molecular Cell* **82**, 2113–2131.e8 (2022).
27. S. Yatskevich, K. W. Muir, D. Bellini, Z. Zhang, J. Yang, T. Tischer, M. Predin, T. Dendooven, S. H. McLaughlin, D. Barford, Structure of the human inner kinetochores bound to a centromeric CENP-A nucleosome. *Science* **3810**, 2022.01.07.475394–2022.01.07.475394 (2022).
28. T. Tian, L. Chen, Z. Dou, Z. Yang, X. Gao, X. Yuan, C. Wang, R. Liu, Z. Shen, P. Gui, M. Teng, X. Meng, D. L. Hill, L. Li, X. Zhang, X. Liu, L. Sun, J. Zang, X. Yao, Structural insights into human CCAN complex assembled onto DNA. *Cell Discov* **8**, 1–15 (2022).
29. C. Obuse, H. Yang, N. Nozaki, S. Goto, T. Okazaki, K. Yoda, Proteomics analysis of the centromere complex from HeLa interphase cells: UV-damaged DNA binding protein 1 (DDB-1) is a component of the CEN-complex, while BMI-1 is transiently co-localized with the centromeric region in interphase. *Genes to Cells* **9**, 105–120 (2004).
30. H. Izuta, M. Ikeno, N. Suzuki, T. Tomonaga, N. Nozaki, C. Obuse, Y. Kisu, N. Goshima, F. Nomura, N. Nomura, K. Yoda, Comprehensive analysis of the ICEN (Interphase Centromere Complex) components enriched in the CENP-A chromatin of human cells. *Genes to Cells* **11**, 673–684 (2006).

31. Y. Roulland, K. Ouararhni, M. Naidenov, L. Ramos, M. Shuaib, S. H. Syed, I. N. Lone, R. Boopathi, E. Fontaine, G. Papai, H. Tachiwana, T. Gautier, D. Skoufias, K. Padmanabhan, J. Bednar, H. Kurumizaka, P. Schultz, D. Angelov, A. Hamiche, S. Dimitrov, The Flexible Ends of CENP-A Nucleosome Are Required for Mitotic Fidelity. *Molecular Cell* **63**, 674–685 (2016).
32. C. P. Seath, A. J. Burton, X. Sun, G. Lee, R. E. Kleiner, D. W. C. MacMillan, T. W. Muir, Tracking chromatin state changes using nanoscale photo-proximity labelling. *Nature* **616**, 574–580 (2023).
33. G. Orphanides, G. LeRoy, C. H. Chang, D. S. Luse, D. Reinberg, FACT, a factor that facilitates transcript elongation through nucleosomes. *Cell* **92**, 105–116 (1998).
34. R. Belotserkovskaya, S. Oh, V. A. Bondarenko, G. Orphanides, V. M. Studitsky, D. Reinberg, FACT facilitates transcription-dependent nucleosome alteration. *Science (New York, N.Y.)* **301**, 1090–3 (2003).
35. A. Saunders, J. Werner, E. D. Andrulis, T. Nakayama, S. Hirose, D. Reinberg, J. T. Lis, Tracking FACT and the RNA Polymerase II Elongation Complex Through Chromatin in Vivo. *Science* **301**, 1094–1096 (2003).
36. F.-K. Hsieh, O. I. Kulaeva, S. S. Patel, P. N. Dyer, K. Luger, D. Reinberg, V. M. Studitsky, Histone chaperone FACT action during transcription through chromatin by RNA polymerase II. *Proceedings of the National Academy of Sciences* **110**, 7654–7659 (2013).
37. M. B. Schlesinger, T. Formosa, POB3 is required for both transcription and replication in the yeast *Saccharomyces cerevisiae*. *Genetics* **155**, 1593–606 (2000).
38. D. M. Keller, H. Lu, p53 serine 392 phosphorylation increases after UV through induction of the assembly of the CK2.hSPT16.SSRP1 complex. *The Journal of biological chemistry* **277**, 50206–13 (2002).
39. N. M. Krohn, C. Stemmer, P. Fojan, R. Grimm, K. D. Grasser, Protein Kinase CK2 Phosphorylates the High Mobility Group Domain Protein SSRP1, Inducing the Recognition of UV-damaged DNA*. *Journal of Biological Chemistry* **278**, 12710–12715 (2003).
40. B. C. Tan, C. Chien, S. Hirose, S. Lee, Functional cooperation between FACT and MCM helicase facilitates initiation of chromatin DNA replication. *The EMBO Journal* **25**, 3975–3985 (2006).
41. A. Kumari, O. M. Mazina, U. Shinde, A. V. Mazin, H. Lu, A role for SSRP1 in recombination-mediated DNA damage response. *Journal of Cellular Biochemistry* **108**, 508–518 (2009).
42. H. Xin, S. Takahata, M. Blanksma, L. McCullough, D. J. Stillman, T. Formosa, yFACT induces global accessibility of nucleosomal DNA without H2A-H2B displacement. *Molecular cell* **35**, 365–76 (2009).
43. J. Han, Q. Li, L. McCullough, C. Kettelkamp, T. Formosa, Z. Zhang, Ubiquitylation of FACT by the Cullin-E3 ligase Rtt101 connects FACT to DNA replication. *Genes Dev.* **24**, 1485–1490 (2010).
44. J. L. C. Richard, M. S. Shukla, H. Menoni, K. Ouararhni, I. N. Lone, Y. Roulland, C. Papin, E. B. Simon, T. Kundu, A. Hamiche, D. Angelov, S. Dimitrov, FACT Assists Base Excision Repair by Boosting the Remodeling Activity of RSC. *PLOS Genetics* **12**, e1006221 (2016).
45. J. Yang, X. Zhang, J. Feng, H. Leng, S. Li, J. Xiao, S. Liu, Z. Xu, J. Xu, D. Li, Z. Wang, J. Wang, Q. Li, The Histone Chaperone FACT Contributes to DNA Replication-Coupled Nucleosome Assembly. *Cell Reports* **16**, 3414 (2016).
46. G. Yang, Y. Chen, J. Wu, S.-H. Chen, X. Liu, A. K. Singh, X. Yu, Poly(ADP-ribosyl)ation mediates early phase histone eviction at DNA lesions. *Nucleic Acids Research* **48**, 3001–3013 (2020).
47. G. Orphanides, W.-H. Wu, W. S. Lane, M. Hampsey, D. Reinberg, The chromatin-specific transcription elongation factor FACT comprises human SPT16 and SSRP1 proteins. *Nature* **400**, 284–288 (1999).

48. D. D. Winkler, K. Luger, The histone chaperone FACT: Structural insights and mechanisms for nucleosome reorganization. *Journal of Biological Chemistry* **286**, 18369–18374 (2011).
49. D. D. Winkler, U. M. Muthurajan, A. R. Hieb, K. Luger, Histone Chaperone FACT Coordinates Nucleosome Interaction through Multiple Synergistic Binding Events*. *Journal of Biological Chemistry* **286**, 41883–41892 (2011).
50. X. Wang, Y. Tang, J. Xu, H. Leng, G. Shi, Z. Hu, J. Wu, Y. Xiu, J. Feng, Q. Li, The N-terminus of Spt16 anchors FACT to MCM2–7 for parental histone recycling. *Nucleic Acids Research* **51**, 11549–11567 (2023).
51. B. Safaric, E. Chacin, M. J. Scherr, L. Rajappa, C. Gebhardt, C. F. Kurat, T. Cordes, K. E. Duderstadt, The fork protection complex recruits FACT to reorganize nucleosomes during replication. *Nucleic Acids Research* **50**, 1317–1334 (2022).
52. H. Leng, S. Liu, Y. Lei, Y. Tang, S. Gu, J. Hu, S. Chen, J. Feng, Q. Li, FACT interacts with Set3 HDAC and fine-tunes GAL1 transcription in response to environmental stimulation. *Nucleic Acids Research* **49**, 5502–5519 (2021).
53. D. J. Kemble, F. G. Whitby, H. Robinson, L. L. McCullough, T. Formosa, C. P. Hill, Structure of the spt16 middle domain reveals functional features of the Histone chaperone FACT. *Journal of Biological Chemistry* **288**, 10188–10194 (2013).
54. D. J. Kemble, L. L. McCullough, F. G. Whitby, T. Formosa, C. P. Hill, FACT Disrupts Nucleosome Structure by Binding H2A-H2B with Conserved Peptide Motifs. *Molecular Cell* **60**, 294–306 (2015).
55. Y. Tsunaka, Y. Fujiwara, T. Oyama, S. Hirose, K. Morikawa, Integrated molecular mechanism directing nucleosome reorganization by human FACT. *Genes & development* **30**, 673–86 (2016).
56. A. T. Yarnell, S. Oh, D. Reinberg, S. J. Lippard, Interaction of FACT, SSRP1, and the high mobility group (HMG) domain of SSRP1 with DNA damaged by the anticancer drug cisplatin. *J Biol Chem* **276**, 25736–25741 (2001).
57. M. Štros, D. Launholt, K. D. Grasser, The HMG-box: a versatile protein domain occurring in a wide variety of DNA-binding proteins. *Cell. Mol. Life Sci.* **64**, 2590 (2007).
58. M. Okada, K. Okawa, T. Isobe, T. Fukagawa, CENP-H-containing complex facilitates centromere deposition of CENP-A in cooperation with FACT and CHD1. *Molecular biology of the cell* **20**, 3986–95 (2009).
59. C. C. Chen, S. Bowers, Z. Lipinski, J. Palladino, S. Trusiak, E. Bettini, L. Rosin, M. R. Przewlaka, D. M. Glover, R. J. O'Neill, B. G. Mellone, Establishment of Centromeric Chromatin by the CENP-A Assembly Factor CAL1 Requires FACT-Mediated Transcription. *Developmental Cell* **34**, 73–84 (2015).
60. G. M. R. Deyter, S. Biggins, The FACT complex interacts with the E3 ubiquitin ligase Psh1 to prevent ectopic localization of CENP-A. *Genes and Development* **28**, 1815–1826 (2014).
61. E. S. Choi, A. Strålfors, S. Catania, A. G. Castillo, J. P. Svensson, A. L. Pidoux, K. Ekwall, R. C. Allshire, Factors That Promote H3 Chromatin Integrity during Transcription Prevent Promiscuous Deposition of CENP-ACnp1 in Fission Yeast. *PLoS Genetics* **8** (2012).
62. E. Lejeune, M. Bortfeld, S. A. White, A. L. Pidoux, K. Ekwall, R. C. Allshire, A. G. Ladurner, The Chromatin-Remodeling Factor FACT Contributes to Centromeric Heterochromatin Independently of RNAi. *Current Biology* **17**, 1219–1224 (2007).
63. L. Prendergast, S. Müller, Y. Liu, H. Huang, F. Dingli, D. Loew, I. Vassias, D. J. Patel, K. F. Sullivan, G. Almouzni, The CENP-T/-W complex is a binding partner of the histone chaperone FACT. *Genes & Development* **30**, 1313–1326 (2016).

64. M. E. Pesenti, D. Prumbaum, P. Auckland, C. M. Smith, A. C. Faesen, A. Petrovic, M. Erent, S. Maffini, S. Pentakota, J. R. Weir, Y. C. Lin, S. Raunser, A. D. McAinsh, A. Musacchio, Reconstitution of a 26-Subunit Human Kinetochores Reveals Cooperative Microtubule Binding by CENP-OPQUR and NDC80. *Molecular Cell* **71**, 923-939.e10 (2018).
65. J. R. Weir, A. C. Faesen, K. Klare, A. Petrovic, F. Basilico, J. Fischböck, S. Pentakota, J. Keller, M. E. Pesenti, D. Pan, D. Vogt, S. Wohlgenuth, F. Herzog, A. Musacchio, Insights from biochemical reconstitution into the architecture of human kinetochores. *Nature* **537**, 249–253 (2016).
66. J. L. Birch, B. C.-M. Tan, K. I. Panov, T. B. Panova, J. S. Andersen, T. A. Owen-Hughes, J. Russell, S.-C. Lee, J. C. B. M. Zomerdijk, FACT facilitates chromatin transcription by RNA polymerases I and III. *The EMBO Journal* **28**, 854–865 (2009).
67. E. Jeong, J. A. Martina, P. S. Contreras, J. Lee, R. Puertollano, The FACT complex facilitates expression of lysosomal and antioxidant genes through binding to TFEB and TFE3. *Autophagy* **18**, 2333–2349 (2022).
68. D. Fachinetti, J. S. Han, M. A. McMahon, P. Ly, A. Abdullah, A. J. Wong, D. W. Cleveland, DNA sequence-specific binding of CENP-B enhances the fidelity of human centromere function. *Dev Cell* **33**, 314–327 (2015).
69. K. Nishimura, T. Fukagawa, H. Takisawa, T. Kakimoto, M. Kanemaki, An auxin-based degron system for the rapid depletion of proteins in nonplant cells. *Nat Methods* **6**, 917–922 (2009).
70. R. W. Dirks, S. Snaar, Dynamics of RNA polymerase II localization during the cell cycle. *Histochem Cell Biol* **111**, 405–410 (1999).
71. F. L. Chan, O. J. Marshall, R. Saffery, B. W. Kim, E. Earle, K. H. A. Choo, L. H. Wong, Active transcription and essential role of RNA polymerase II at the centromere during mitosis. *Proc Natl Acad Sci U S A* **109**, 1979–1984 (2012).
72. S. Rošić, F. Köhler, S. Erhardt, Repetitive centromeric satellite RNA is essential for kinetochore formation and cell division. *Journal of Cell Biology* **207**, 335–349 (2014).
73. H. Liu, Q. Qu, R. Warrington, A. Rice, N. Cheng, H. Yu, Mitotic Transcription Installs Sgo1 at Centromeres to Coordinate Chromosome Segregation. *Mol Cell* **59**, 426–436 (2015).
74. O. Molina, G. Vargiu, M. A. Abad, A. Zhiteneva, A. A. Jeyapakash, H. Masumoto, N. Kouprina, V. Larionov, W. C. Earnshaw, Epigenetic engineering reveals a balance between histone modifications and transcription in kinetochore maintenance. *Nat Commun* **7**, 13334 (2016).
75. G. O. M. Bobkov, N. Gilbert, P. Heun, Centromere transcription allows CENP-A to transit from chromatin association to stable incorporation. *Journal of Cell Biology* **217**, 1957–1972 (2018).
76. Y. H. Kang, J.-E. Park, L.-R. Yu, N.-K. Soung, S.-M. Yun, J. K. Bang, Y.-S. Seong, H. Yu, S. Garfield, T. D. Veenstra, K. S. Lee, Self-Regulated Plk1 Recruitment to Kinetochores by the Plk1-PBIP1 Interaction Is Critical for Proper Chromosome Segregation. *Molecular Cell* **24**, 409–422 (2006).
77. A. C. Amaro, C. P. Samora, R. Holtackers, E. Wang, I. J. Kingston, M. Alonso, M. Lampson, A. D. McAinsh, P. Meraldi, Molecular control of kinetochore-microtubule dynamics and chromosome oscillations. *Nat Cell Biol* **12**, 319–329 (2010).
78. S. Hua, Z. Wang, K. Jiang, Y. Huang, T. Ward, L. Zhao, Z. Dou, X. Yao, CENP-U Cooperates with Hec1 to Orchestrate Kinetochore-Microtubule Attachment. *J Biol Chem* **286**, 1627–1638 (2011).
79. P. Singh, M. E. Pesenti, S. Maffini, A. Srinivasamani, P. Singh, M. E. Pesenti, S. Maffini, S. Carmignani, M. Hedtfeld, A. Petrovic, Article BUB1 and CENP-U , Primed by CDK1 , Are the Main PLK1 Kinetochore Receptors in Mitosis BUB1 and CENP-U , Primed by CDK1 , Are the Main PLK1 Kinetochore Receptors in Mitosis. *Molecular Cell*, 1–21 (2021).

80. M. Okada, I. M. Cheeseman, T. Hori, K. Okawa, I. X. McLeod, J. R. Yates, A. Desai, T. Fukagawa, The CENP-H-I complex is required for the efficient incorporation of newly synthesized CENP-A into centromeres. *Nat Cell Biol* **8**, 446–457 (2006).
81. F. Basilico, S. Maffini, J. R. Weir, D. Prumbaum, A. M. Rojas, T. Zimniak, A. De Antoni, S. Jeganathan, B. Voss, S. van Gerwen, V. Krenn, L. Massimiliano, A. Valencia, I. R. Vetter, F. Herzog, S. Raunser, S. Pasqualato, A. Musacchio, The pseudo GTPase CENP-M drives human kinetochore assembly. *eLife* **3**, 1–28 (2014).
82. B. Nabet, J. M. Roberts, D. L. Buckley, J. Paulk, S. Dastjerdi, A. Yang, A. L. Leggett, M. A. Erb, M. A. Lawlor, A. Souza, T. G. Scott, S. Vittori, J. A. Perry, J. Qi, G. E. Winter, K.-K. Wong, N. S. Gray, J. E. Bradner, The dTAG system for immediate and target-specific protein degradation. *Nat Chem Biol* **14**, 431–441 (2018).
83. K. Žumer, M. Ochmann, A. Aljahani, A. Zheenbekova, A. Devadas, K. C. Maier, P. Rus, U. Neef, A. M. Oudelaar, P. Cramer, FACT maintains chromatin architecture and thereby stimulates RNA polymerase II pausing during transcription in vivo. *Mol Cell*, S1097-2765(24)00396–4 (2024).
84. K. Zhou, Y. Liu, K. Luger, Histone chaperone FACT FACilitates Chromatin Transcription: mechanistic and structural insights. *Current Opinion in Structural Biology* **65**, 26–32 (2020).
85. D. M. Keller, X. Zeng, Y. Wang, Q. H. Zhang, M. Kapoor, H. Shu, R. Goodman, G. Lozano, Y. Zhao, H. Lu, A DNA damage-induced p53 serine 392 kinase complex contains CK2, hSpt16, and SSRP1. *Mol Cell* **7**, 283–292 (2001).
86. Y. Li, D. M. Keller, J. D. Scott, H. Lu, CK2 Phosphorylates SSRP1 and Inhibits Its DNA-binding Activity*. *Journal of Biological Chemistry* **280**, 11869–11875 (2005).
87. K. Mayanagi, K. Saikusa, N. Miyazaki, S. Akashi, K. Iwasaki, Y. Nishimura, K. Morikawa, Y. Tsunaka, Structural visualization of key steps in nucleosome reorganization by human FACT. *Scientific Reports* **9**, 10183–10183 (2019).
88. S. F. Rusin, M. E. Adamo, A. N. Kettenbach, Identification of Candidate Casein Kinase 2 Substrates in Mitosis by Quantitative Phosphoproteomics. *Front Cell Dev Biol* **5**, 97 (2017).
89. K. C. Graham, D. W. Litchfield, The Regulatory β Subunit of Protein Kinase CK2 Mediates Formation of Tetrameric CK2 Complexes*. *Journal of Biological Chemistry* **275**, 5003–5010 (2000).
90. C. Borgo, C. D'Amore, S. Sarno, M. Salvi, M. Ruzzene, Protein kinase CK2: a potential therapeutic target for diverse human diseases. *Sig Transduct Target Ther* **6**, 1–20 (2021).
91. S. E. Roffey, D. W. Litchfield, CK2 Regulation: Perspectives in 2021. *Biomedicine* **9**, 1361 (2021).
92. M. Faust, M. Montenarh, Subcellular localization of protein kinase CK2. A key to its function? *Cell Tissue Res* **301**, 329–340 (2000).
93. V. Martel, O. Filhol, A. Nueda, D. Gerber, M. J. Benitez, C. Cochet, Visualization and molecular analysis of nuclear import of protein kinase CK2 subunits in living cells. *Mol Cell Biochem* **227**, 81–90 (2001).
94. L. Schwind, A. D. Zimmer, C. Götz, M. Montenarh, CK2 phosphorylation of C/EBP δ regulates its transcription factor activity. *Int J Biochem Cell Biol* **61**, 81–89 (2015).
95. I. M. Johnston, S. J. Allison, J. P. Morton, L. Schramm, P. H. Scott, R. J. White, CK2 Forms a Stable Complex with TFIIB and Activates RNA Polymerase III Transcription in Human Cells. *Mol Cell Biol* **22**, 3757–3768 (2002).

96. E. A. Kuenzel, J. A. Mulligan, J. Sommercorn, E. G. Krebs, Substrate specificity determinants for casein kinase II as deduced from studies with synthetic peptides. *Journal of Biological Chemistry* **262**, 9136–9140 (1987).
97. N. Blom, S. Gammeltoft, S. Brunak, Sequence and structure-based prediction of eukaryotic protein phosphorylation sites. *J Mol Biol* **294**, 1351–1362 (1999).
98. N. Blom, T. Sicheritz-Pontén, R. Gupta, S. Gammeltoft, S. Brunak, Prediction of post-translational glycosylation and phosphorylation of proteins from the amino acid sequence. *Proteomics* **4**, 1633–1649 (2004).
99. J. C. Obenauer, L. C. Cantley, M. B. Yaffe, Scansite 2.0: Proteome-wide prediction of cell signaling interactions using short sequence motifs. *Nucleic Acids Res* **31**, 3635–3641 (2003).
100. T. Wang, Y. Liu, G. Edwards, D. Krzizike, H. Scherman, K. Luger, The histone chaperone FACT modulates nucleosome structure by tethering its components. *Life Science Alliance* **1**, 1–13 (2018).
101. L. Farnung, M. Ochmann, M. Engholm, P. Cramer, Structural basis of nucleosome transcription mediated by Chd1 and FACT. *Nature Structural and Molecular Biology* **28**, 382–387 (2021).
102. C. Jeronimo, A. Angel, V. Q. Nguyen, J. M. Kim, C. Poitras, E. Lambert, P. Collin, J. Mellor, C. Wu, F. Robert, FACT is recruited to the +1 nucleosome of transcribed genes and spreads in a Chd1-dependent manner. *Molecular Cell* **81**, 3542–3559.e11 (2021).
103. T. Dendooven, Z. Zhang, J. Yang, S. H. McLaughlin, J. Schwab, S. H. W. Scheres, S. Yatskevich, D. Barford, Cryo-EM structure of the complete inner kinetochore of the budding yeast point centromere. *Science Advances* **9**, eadg7480 (2023).
104. J. M. Gottesfeld, D. J. Forbes, Mitotic repression of the transcriptional machinery. *Trends in Biochemical Sciences* **22**, 197–202 (1997).
105. R. Belotserkovskaya, S. Oh, V. A. Bondarenko, G. Orphanides, V. M. Studitsky, D. Reinberg, FACT Facilitates Transcription-Dependent Nucleosome Alteration. *Science* **301**, 1090–1093 (2003).
106. Y. Tsunaka, J. Toga, H. Yamaguchi, S. Tate, S. Hirose, K. Morikawa, Phosphorylated Intrinsically Disordered Region of FACT Masks Its Nucleosomal DNA Binding Elements. *J Biol Chem* **284**, 24610–24621 (2009).
107. P. J. Huis in 't Veld, S. Wohlge-muth, C. Koerner, F. Müller, P. Janning, A. Musacchio, Reconstitution and use of highly active human CDK1:Cyclin-B:CKS1 complexes. *Protein Sci* **31**, 528–537 (2022).
108. F. Girdler, F. Sessa, S. Patercoli, F. Villa, A. Musacchio, S. Taylor, Molecular Basis of Drug Resistance in Aurora Kinases. *Chemistry & Biology* **15**, 552–562 (2008).
109. D. J. Fitzgerald, P. Berger, C. Schaffitzel, K. Yamada, T. J. Richmond, I. Berger, Protein complex expression by using multigene baculoviral vectors. *Nat Methods* **3**, 1021–1032 (2006).
110. F. Weissmann, G. Petzold, R. VanderLinden, P. J. Huis in 't Veld, N. G. Brown, F. Lampert, S. Westermann, H. Stark, B. A. Schulman, J.-M. Peters, biGBac enables rapid gene assembly for the expression of large multisubunit protein complexes. *Proc. Natl. Acad. Sci. U.S.A.* **113** (2016).
111. D. G. Gibson, L. Young, R.-Y. Chuang, J. C. Venter, C. A. Hutchison, H. O. Smith, Enzymatic assembly of DNA molecules up to several hundred kilobases. *Nat Methods* **6**, 343–345 (2009).
112. V. Krenn, A. Wehenkel, X. Li, S. Santaguida, A. Musacchio, Structural analysis reveals features of the spindle checkpoint kinase Bub1–kinetochore subunit Knl1 interaction. *Journal of Cell Biology* **196**, 451–467 (2012).

113. A. Guse, C. J. Fuller, A. F. Straight, A cell-free system for functional centromere and kinetochore assembly. *Nature Protocols* **7**, 1847–1869 (2012).
114. J. Raaf, E. Brunstein, O. Issinger, K. Niefind, The interaction of CK2 α and CK2 β , the subunits of protein kinase CK2, requires CK2 β in a preformed conformation and is enthalpically driven. *Protein Science* **17**, 2180–2186 (2008).
115. C. Werner, A. Gast, D. Lindenblatt, A. Nickelsen, K. Niefind, J. Jose, J. Hochscherf, Structural and Enzymological Evidence for an Altered Substrate Specificity in Okur-Chung Neurodevelopmental Syndrome Mutant CK2 α Lys198Arg. *Frontiers in Molecular Biosciences* **9** (2022).
116. C. L. Ladner, J. Yang, R. J. Turner, R. A. Edwards, Visible fluorescent detection of proteins in polyacrylamide gels without staining. *Anal Biochem* **326**, 13–20 (2004).
117. W. Holzmüller, U. Kulozik, Protein quantification by means of a stain-free SDS-PAGE technology without the need for analytical standards: Verification and validation of the method. *Journal of Food Composition and Analysis* **48**, 128–134 (2016).
118. J. Rappsilber, M. Mann, Y. Ishihama, Protocol for micro-purification, enrichment, pre-fractionation and storage of peptides for proteomics using StageTips. *Nat Protoc* **2**, 1896–1906 (2007).
119. J. Cox, M. Mann, MaxQuant enables high peptide identification rates, individualized p.p.b.-range mass accuracies and proteome-wide protein quantification. *Nat Biotechnol* **26**, 1367–1372 (2008).
120. S.-T. Liu, J. C. Hittle, S. A. Jablonski, M. S. Campbell, K. Yoda, T. J. Yen, Human CENP-I specifies localization of CENP-F, MAD1 and MAD2 to kinetochores and is essential for mitosis. *Nat Cell Biol* **5**, 341–345 (2003).
121. S. Ghetti, M. Burigotto, A. Mattivi, G. Magnani, A. Casini, A. Bianchi, A. Cereseto, L. L. Fava, CRISPR/Cas9 ribonucleoprotein-mediated knockin generation in hTERT-RPE1 cells. *STAR Protoc* **2**, 100407 (2021).
122. J. Schindelin, I. Arganda-Carreras, E. Frise, V. Kaynig, M. Longair, T. Pietzsch, S. Preibisch, C. Rueden, S. Saalfeld, B. Schmid, J.-Y. Tinevez, D. J. White, V. Hartenstein, K. Eliceiri, P. Tomancak, A. Cardona, Fiji: an open-source platform for biological-image analysis. *Nat Methods* **9**, 676–682 (2012).

~~data needed to evaluate the conclusions in the paper are present in the paper and the Supplementary Materials.~~

January 24, 2025

RE: JCB Manuscript #202412042T

Andrea Musacchio
Max Planck Society

Dear Prof. Musacchio:

Thank you for submitting your revised manuscript entitled "Interactions with multiple inner kinetochore proteins determine mitotic localization of FACT". We would be happy to publish your paper in JCB after remaining reviewer concerns are resolved, and final revisions necessary to meet our formatting guidelines (see details below).

As you will see, reviewers all commended the significant efforts to improve this manuscript. While Reviewer 1 noted the work would be further strengthened by full validation of proposed phospho-sites, the other reviewers strongly support publication. However, we concur with Reviewer 1 that the complexity of the data may pose an undue burden on readers who may not be familiar with these complexes. Please include one or multiple models to guide the reader through the text as suggested by Reviewer 1. Further to the goal of readability, please add references to specific lanes in the gels that support each major claim, not simply references to the gel itself. We feel strongly that this effort will reward the authors with a paper that is more accessible to the broad readership at JCB and strengthen its impact accordingly. Finally, please attend to the specific points noted by Reviewers 1 and 2 below.

A. MANUSCRIPT ORGANIZATION AND FORMATTING:

Full guidelines are available on our Instructions for Authors page, <http://jcb.rupress.org/submission-guidelines#revised>. Submission of a paper that does not conform to JCB guidelines will delay the acceptance of your manuscript.

1) Text limits: Character count for Articles is < 40,000, not including spaces. Count includes abstract, introduction, results, discussion, and acknowledgments. Count does not include title page, figure legends, materials and methods, references, tables, or supplemental legends.

2) Figures limits: Articles may have up to 10 main figures and 5 supplemental figures/tables.

** Please endeavor to reduce the number of supplemental figures, perhaps by making some of them into main figures as well as merging them as appropriate.

3) Figure formatting: Scale bars must be present on all microscopy images, including inset magnifications. Molecular weight or nucleic acid size markers must be included on all gel electrophoresis. Please avoid pairing red and green for images and graphs to ensure legibility for color-blind readers. If red and green are paired for images, please ensure that the particular red and green hues used in micrographs are distinctive with any of the colorblind types. If not, please modify colors accordingly or provide separate images of the individual channels.

** The insets in Fig1F are exceedingly small. Please improve these, perhaps by showing only the merged inset image, in a 5th column to the right of the other images.

4) Statistical analysis: Error bars on graphic representations of numerical data must be clearly described in the figure legend. The number of independent data points (n) represented in a graph must be indicated in the legend. Statistical methods should be explained in full in the materials and methods. For figures presenting pooled data the statistical measure should be defined in the figure legends. Please also be sure to indicate the statistical tests used in each of your experiments (either in the figure legend itself or in a separate methods section) as well as the parameters of the test (for example, if you ran a t-test, please indicate if it was one- or two-sided, etc.). Also, if you used parametric tests, please indicate if the data distribution was tested for normality (and if so, how). If not, you must state something to the effect that "Data distribution was assumed to be normal but this was not formally tested."

** Please define error bars shown in Fig 2B-E.

5) Abstract and title: The abstract should be no longer than 160 words and should communicate the significance of the paper for a general audience. The title should be less than 100 characters including spaces. Make the title concise but accessible to a general readership.

6) Materials and methods: Should be comprehensive and not simply reference a previous publication for details on how an experiment was performed. Please provide full descriptions in the text for readers who may not have access to referenced manuscripts. We also provide a report from SciScore and an associate score, which we encourage you to use as a means of

evaluating and improving the methods section.

7) Please be sure to provide the sequences for all of your primers/oligos, plasmids, and RNAi constructs in the materials and methods. You must also indicate in the methods the source, species, and catalog numbers (where appropriate) for all of your antibodies. Please also indicate the acquisition and quantification methods for immunoblotting/western blots.

8) Microscope image acquisition: The following information must be provided about the acquisition and processing of images:

- a. Make and model of microscope
- b. Type, magnification, and numerical aperture of the objective lenses
- c. Temperature
- d. Imaging medium
- e. Fluorochromes
- f. Camera make and model
- g. Acquisition software
- h. Any software used for image processing subsequent to data acquisition. Please include details and types of operations involved (e.g., type of deconvolution, 3D reconstitutions, surface or volume rendering, gamma adjustments, etc.).

10) Supplemental materials: There are strict limits on the allowable amount of supplemental data. Articles may have up to 5 supplemental figures. Please also note that tables, like figures, should be provided as individual, editable files. A summary of all supplemental material should appear at the end of the Materials and methods section.

13) ORCID IDs: ORCID IDs are unique identifiers allowing researchers to create a record of their various scholarly contributions in a single place. At resubmission of your final files, please provide an ORCID ID for all authors.

15) A data availability statement is required for all research article submissions. The statement should address all data underlying the research presented in the manuscript. Please visit the JCB instructions for authors for guidelines and examples of statements at (<https://rupress.org/jcb/pages/editorial-policies#data-availability-statement>).

Please note that JCB requires authors to submit Source Data used to generate figures containing gels and Western blots with all revised manuscripts. This Source Data consists of fully uncropped and unprocessed images for each gel/blot displayed in the main and supplemental figures. Since your paper includes cropped gel and/or blot images, please be sure to provide one Source Data file for each figure that contains gels and/or blots along with your revised manuscript files. File names for Source Data figures should be alphanumeric without any spaces or special characters (i.e., SourceDataF#, where F# refers to the associated main figure number or SourceDataFS# for those associated with Supplementary figures). The lanes of the gels/blots should be labeled as they are in the associated figure, the place where cropping was applied should be marked (with a box), and molecular weight/size standards should be labeled wherever possible. Source Data files will be directly linked to specific figures in the published article.

Journal of Cell Biology now requires a data availability statement for all research article submissions. These statements will be published in the article directly above the Acknowledgments. The statement should address all data underlying the research presented in the manuscript. Please visit the JCB instructions for authors for guidelines and examples of statements at (<https://rupress.org/jcb/pages/editorial-policies#data-availability-statement>).

B. FINAL FILES:

Thank you for your attention to these final processing requirements. Please revise and format the manuscript and upload materials within 7 days. If you need an extension for whatever reason, please let us know and we can work with you to determine a suitable revision period.

Thank you for this interesting contribution, we look forward to publishing your paper in Journal of Cell Biology.

Sincerely,

William Earnshaw
Monitoring Editor
Journal of Cell Biology

Tim Fessenden
Scientific Editor
Journal of Cell Biology

Reviewer #1 (Comments to the Authors (Required)):

The authors investigated molecular interaction between CCAN and FACT complexes. They revealed contact domains in FACT and the cognate subcomplexes of CCAN by in vitro reconstitution from recombinant proteins followed by SEC and pull down assay. They also investigated a couple of potential means to control interactions between FACT and CCAN. They proposed that phosphorylation of FACT by CK2 is essential for binding to the CCAN; and CENP-A nucleosomes or DNA prevent CCAN from interacting with FACT.

It is disappointing that the authors did not address the concerns raised by Reviewer 1 on identity of phosphorylation. I consider data as preliminary without exact phosphorylation sites and/or cognate protein kinases in vivo. The in vitro study on phosphorylation is conducted on the premise that CK2 regulates FACT-CCAN interaction in vivo. Is there any actual evidence for that? Please see also my question on Figure S8D below.

I acknowledge that the authors made substantial efforts to dissect interactions between FACT and CCAN, unearthing a network involving multiple CCAN subunits. The author also revealed crosstalk among CCAN super-subunits and FACT, and made

hypothesis to interpret each phenotype, but stop short of further investigation to validate the hypothesis.
The data is rather complicated for readers to comprehend the network between CCAN subunits and FACT, without a model that explains the unexpected observations reported in this manuscript.

Specific points

Figure 1B

Is this the best angle of view to show geometrical distribution of subcomplexes TW, C, HIKM and OPQR? Are they reasonably close enough to be contacted by a single SPT16 MID domain?

How big is the size of SPT16 MID domain relative a kinetochore?

Figure S8D

I don't see any difference between lanes 5 and 6. If CK2a is present in lane 6, it should look more like lane 4, unless the three kinases counteract CK2.

Figures 3K and S7

Why is the CCAN intensity not normalized by CENPC or CREST intensity as in other Figures?

Figure S7 E and G, S5C

p-value and median are not shown.

Reviewer #2 (Comments to the Authors (Required)):

This MS by Musacchio and his colleagues is transferred from Review commons. This focuses on FACT-CCAN interaction from structural viewpoints and includes many new findings.

This reviewer reviewed original MS and found that authors addressed many concerns from original reviewers including many of new experimental data. It is especially nice to show CENP-TW is reduced by acute FACT degradation in K562 cells.

Overall, I think that this version is suitable for publication in JCB.

Minor point:

In introduction "A second subunit, CENP-T..."

In this para, authors start to mention about CENP-C and CENP-N and they explain about CENP-C first. Why is "a second subunit" CENP-T? It is a little odd.

Reviewer #3 (Comments to the Authors (Required)):

The authors have addressed my concerns. The manuscript is suitable for publication.

Response to Reviewers

Reviewer 1

We thank the reviewer for their support and their constructive criticism.

Specific points

Figure 1B

Is this the best angle of view to show geometrical distribution of subcomplexes TW, C, HIKM and OPQR? Are they reasonably close enough to be contacted by a single SPT16 MID domain?

How big is the size of SPT16 MID domain relative a kinetochore?

We feel that this is the best angle because it is the only angle that allows visualization of all subunits in a single view. The depiction of the MID domain is complex due to the presence of a 60-residue intrinsically disordered region (IDR). IDRs are also present at the C-terminus. A rapid calculation for an extended IDR of 60 residues would conservatively predict a potential extension of 12 nm (0.2 nm/residue), which is sufficient to span most of the CCAN.

Figure S8D

I don't see any difference between lanes 5 and 6. If CK2a is present in lane 6, it should look more like lane 4, unless the three kinases counteract CK2.

We agree with the reviewer that lane 6 was expected to appear more like lane 4, and we agree that it might signify that one or more of the three kinases counteract CK2 in vitro. Regretfully, we cannot provide more than a handwaving explanation for this interesting phenomenon at the moment.

Figures 3K and S7

Why is the CCAN intensity not normalized by CENPC or CREST intensity as in other Figures?

The main problem with the previous automated analysis is that the non-circular shape of the CREST-staining led to inconsistencies with the statistical analysis and the statement. In contrast, the same analysis works well when the CENP-C signal was used for KT identification (e.g. in Figure 4), as CENP-C staining yields well separated circular signals ideally suited for our automated identification of individual KTs and subsequent retrieval of fluorescence intensities. We have therefore modified our analysis macro for all experiments where CREST was used as a reference. We used Otsu-thresholding of the DAPI signal for generating a segmentation mask per each cell. Then, integrated cell intensities were calculated for each fluorescence channel based on the DAPI reference mask. We have therefore plotted total signal intensities for all experiments where CREST was used as kinetochore reference and updated the Methods and Results sections to reflect the revised analysis.

Figure S7 E and G, S5C

p-value and median are not shown.

We have added the missing values.

Reviewer 2 and Reviewer 3

We are very grateful to the reviewer for their positive assessment and for supporting publication of this manuscript. We have slightly amended the text to overcome the minor apparent contradiction mentioned by reviewer 2.